

# Precipitation of Calcium Carbonate Mineral Induced by Viral Lysis of Cyanobacteria

Hengchao Xu[1, 2], Xiaotong Peng[2*], Shijie Bai[2, 3], Kaiwen Ta[2], Shouye Yang[1], Shuangquan Liu[2], Ho Bin Jang[3], Zixiao Guo[2]

[1]School of Ocean and Earth Science, Tongji University, Shanghai, China
[2]Deep-sea Science division, Institute of Deep-sea Science and Engineering, Chinese Academy of Science, Sanya, Hainan, China
[3]Department of Microbiology, The Ohio State University, Columbus, OH, USA

*Correspondence to*: X. Peng (xtpeng@idsse.ac.cn)

**Abstract.** Viruses have been acknowledged as important components of the marine system for the past two decades, but the understanding of their role in the functioning of the geochemical cycle remains poor. Viral induced rupturing of cyanobacteria are theoretically capable of releasing intracellular bicarbonate and inducing the homogeneous nucleation of calcium carbonate, but experimental-based support for viral induced calcification is lacking. In this laboratory study, both water carbonate chemistry and precipitates were monitored during the cyanophage infection and lysis of host cells. Our results show that viral lysis of cyanobacteria can influence the carbonate equilibrium system remarkably and promotes the nucleation and stabilization of carbonate minerals. Amorphous calcium carbonate (ACC) and aragonite were evident in the lysate compared to the brucite precipitate in non-infection cultures implying that a different precipitation process had occurred. Based on the carbonate chemistry change and microstructure of the precipitation, the viral induced calcification may initiate by rapid intracellular calcification because of the unfettered intracellular access of $Ca^{2+}$ and react with cytoplasmic alkalinity after the virus attacks and breaks down the cell wall. The experimental results raised here first demonstrate the pathway and result regarding how viruses influence the mineralization of carbonate minerals. Furthermore, our results also imply that viruses play a crucial role in seawater carbonate chemistry and may balance the geochemical element budget within the earth system.

## 1 Introduction

Over the past few years, several studies have highlighted the urgent need to better understand the changes in sea water carbonate chemistry, which is one of the fundamental processes in the carbon cycle on both the global and regional scale, mainly because it is the primary buffer for the acidity of water in the earth-surface environment (Ridgwell and Zeebe, 2005; Martin, 2017; Zeebe, 2012). The ocean is recognized as a large carbon reservoir that contains about sixty times more carbon in the form of dissolved inorganic carbon than that in the pre-anthropogenic atmosphere (Zeebe and Wolf-Gladrow, 2009). Dissolved carbon dioxide in the typical surface seawater pH of 8.2 is mainly in the form of $[HCO_3^-]$ compared to the speciation



of [$CO_3^{2-}$] and [$CO_2$] with a ratio at 89 : 10.5 : 0.5 (Zeebe and Wolf-Gladrow, 2009). Formation and dissolution of calcium carbonate is the most important process that can change the carbonate chemistry in sea water (Equation 1).

$$Ca^{2+} (aq) + 2HCO_3^- (aq) \rightleftharpoons CaCO_3 (s) + CO_2 (aq) + H_2O \tag{1}$$

Carbonate in modern sea water is considered supersaturated with saturation index ranging from 2 to 4. But there is no persistent
precipitation of calcium carbonate in sea water because of the inability to overcome the energetic threshold for homogeneous precipitation. Marine calcium carbonate mineral is traditionally considered to calcification by coccolithophores, foraminifera or pteropods. When unaccounted fraction of marine calcium carbonate particle in seawater was identified recently, additional environmental factors affecting the formation of calcium carbonate should be further investigated (Heldal et al., 2012).

Cyanobacteria, which are the ubiquitous abundant organisms and play important roles in most aquatic environments, are
usually known to influence the $CaCO_3$ precipitation by taking up inorganic carbon via photosynthesis (Obst et al., 2009b;Planavsky et al., 2009;Yang et al., 2016;Kamennaya et al., 2012;Kranz et al., 2010;Semesi et al., 2009;Riding, 2011;Merz-Preiß, 2000) and are recognized as major players in the formation of carbonate sedimentary deposits such as stromatolites (Riding, 2012;Altermann et al., 2006), the oldest one of which was formed by cyanobacteria and is possibly as old as 3.5 billion years (Schopf, 1993). Although there is great importance in the calcium carbonate formation mediated by
cyanobacteria in the sedimentary deposits, the involved mechanisms are still poorly understood. In many cases, the precipitation of $CaCO_3$ by cyanobacteria have been invariably considered a non-controlled process promoted either by photosynthetic uptake of inorganic carbon, raising the pH adjacent to cyanobacterial cells (Riding, 2006) or induced by cell-surface properties for the nucleation of $CaCO_3$ minerals (Obst et al., 2009b). Furthermore, diverse cyanobacterial taxa have been shown recently to form amorphous calcium carbonates intracellularly with the diameter of several hundreds of
nanometers (Couradeau et al., 2012;Benzerara et al., 2014). The intracellular carbonate makes the mechanisms involved in calcification more confusing (Cam et al., 2015;Li et al., 2016;Cam et al., 2016). Thus, new pathway that some biological processes alter the carbonate system are thus important to evaluate.

There are an estimated $10^{30}$ viruses in the ocean, the majority of which infect microbes, including bacteria, archaea, and microeukaryotes, and all of which are vital players in global geochemical cycle of key nutrient elements (Suttle, 2005,
2007;Brussaard et al., 2008). It has been established that viruses infecting specific hosts can be extremely abundant. For example, 3-31% of free-living bacteria are infected by viruses and can occur in excess of $10^5$ infectious units ml[-1] (Suttle and Chan, 1994). Hence, virus lysis of microbe is certainly thought to have direct effects on both ecosystem function involving the release of nutrients back into the environment and some host involved geochemical reaction (Brussaard et al., 2008;Rohwer and Thurber, 2009;Weitz and Wilhelm, 2012;Jover et al., 2014).
virus induced cyanobacteria lysate theoretically can elevate the saturation index of carbonate minerals by releasing of cytoplasmic-associated bicarbonate (Lisle and Robbins, 2016). This thermodynamics calculation proposed by Lisle and Robbins (2016) also highlights that the released cytoplasmic-associated bicarbonate can be as much as ~23-fold greater than in the surrounding seawater, which can shift the carbonate chemistry toward the homogenous nucleation of calcium carbonate.



However, it does not take into account in theoretically calculation that magnesium in seawater may influence the properties and behaviours of calcite in seawater (Morse et al., 2007). Furthermore, viral particles could act as nucleation sites for different mineral precipitation. In the past few years, researchers have investigated that the capsid of viruses can interact directly with elements in solutions and thus potentially mediate the formation and precipitation of different minerals (Daughney et al.,
2004;Kyle et al., 2008;Peng et al., 2013;Pacton et al., 2014;De Wit et al., 2015;Laidler and Stedman, 2010). It has been widely studied that viruses can interact with iron minerals under both marine system (including experimental conditions) (Daughney et al., 2004;Bonnain et al., 2016) and in the natural low pH acid mine drainage environment (Kyle et al., 2008;Kyle and Ferris, 2013). Thus, the iron-based flocculation of viruses has already been accepted as a simple and effective method to concentrate marine viruses for ecological study (John et al., 2011). Long-term and short-term experimental silicification was also studied
for the simulation of hot spring mineralization and checking the preservation of viruses in the mineralized structure (Laidler and Stedman, 2010;Orange et al., 2011). The study of the hot spring biofilm by Peng et al. (2013) proved that viruses can be preserved in the natural environment. Recent studies have shown that hypersaline carbonate minerals can precipitate at the surface of viral particles and have implication to the nano-sized calcium carbonate structures in various geological settings (Pacton et al., 2014;Lisle and Robbins, 2016;Perri et al., 2017). All of the above viral mineralization studies are focused on
heterogeneous nucleation, which occurs at nucleation sites on surfaces and is assumed to be much more common than homogeneous nucleation (Sear, 2014). When combined with the release of cytoplasmic-associated bicarbonate, which induces the homogeneous nucleation of calcium carbonate and available nucleation site for heterogenous nucleation, the comprehending of viral influence on the precipitation of carbonate is extremely poor.

Laboratory study of viral calcification was adopted here by culturing *Synechococcus* spp. and cyanophage. Such modeling
experiments do not intend to mimic the processes occurring within the cells, which remain unknown, and generally do not provide an ultimate and direct answer as to which geobiological processes are involved in biomineralization. However, these experiments constrain to some extent the chemical conditions necessary to predict the geochemical process similar to those in the aquatic environment. Carbonate parameters and cultural status were monitored to calculate the carbonate equilibrium system and saturation index. Precipitates of the culture were also characterized to identify the microstructure of the minerals.
Our results provide gross support of the importance of carbonate nucleation and stabilization during viral induced cyanobacteria mortality in the marine system. The extension of the viral role in mediating sea water carbonate systems will also provide an important, but ignored, carbon cycle in earth system.

## 2 Methods and Materials

### 2.1 Cyanobacteria and Cyanophage

Cyanobacteria (Fig. 1) and virus infected cyanobacteria (known as cyanophage) (Fig. 2) were isolated from the surface seawater from Sanya Bay. The isolation, purification and identification of cyanobacteria *Synechococcus* spp. is followed methods from Waterbury (2006). Cells of this strain are coccoid-shaped, and ~1.3 μm in diameter. Cyanophages were isolated





using the double-layer agar technique with the method modified from Millard (2009). Isolated virus particles, which are ~53 nm in diameter, are classified as podovirus base on the morphology and metagenomic analysis (Supplement 1).

## 2.2 Experimental Setup

Cultures of cyanobacteria were grown at 25℃, in 0.2-μm-filtered artificial medium (based on F/2 media, Table 1). Light
intensity was provided in a 12:12-h light: dark cycle. Experiments were carried out in sterile 4-L borosilicate bottles. For each treatment, cells from the precultures were inoculated into 2 L of culture media. In one treatment, viruses were inoculated at the fifth day. Both treatments were run in duplicate incubations.

Subsamples from these incubations were taken to measure the microbial density and chemical composition of the medium after filtering and fixing. Cyanobacteria were counted under a *Leica* fluorescence microscope with autofluorescence.
Magnesium and calcium cations were determined by ion chromatography (*Dionex* ICS-900) after 0.02 μm filtering and acid by 1 M HCl. The precision of the IC method used is 2 ppm for $Ca^{2+}$ and 5 ppm for $Mg^{2+}$. The TA samples (30 ml) were sterile filtered through 0.2 μm filters and stored in borosilicate bottles at room temperature and analyzed by potentiometric titration (*Metrohm* 916 Ti-Touch) within one week. DIC samples (5 ml) were sterile filtered by 0.2 μm filters and stored in borosilicate flasks without headspace at 4 ℃. Subsamples for DIC and TA were poisoned with $HgCl_2$ solution. Salinity was determined
by measuring the apparent electrical conductivity. To determine the activity of carbonate species and the degree of saturation in the solutions sampled, the geochemical computer program PHREEQC Interactive [version 3.3; Wateq4f database; United States Geological Survey (USGS), Reston, VA, USA] was used.

## 2.3 Electronic microscope and XRD

At the end of the stable phase, the particulate fraction of the residual medium (~1 L Fig. 1) was harvested via centrifugation
(13000 g, 5 min) for electronic microscopy and mineral study with the methods adopted from Peng et al. (2013). In brief, subsamples for TEM study were fixed by the addition of glutaraldehyde (to 4% final concentration), rinsed in distilled water to remove salt, mounted on copper grids and air-dried. The TEM analysis was conducted on a JEM-2100F field emission electron microscope operated at an accelerating voltage of 200 kV. Elemental analysis was conducted at 200 kV using an Oxford INCA Energy TEM X-ray energy dispersive spectrometer. Elemental maps were acquired in a STEM DF mode
operating at 200 kV with a focused electron beam (1 nm). The mineralogy of the structures in the areas of interest were determined using SAED. For SEM analysis, dried precipitation was fixed onto aluminum stubs with two-way adherent abs and allowed to dry overnight before being sputter coated with carbon for 2–3 min. Samples were examined with an Apreo scanning electron microscope (Thermofisher Scientific). XRD was employed to characterize the bulk mineralogy of the precipitates. The subsamples were thoroughly ground, followed by analyses using a LabX XRD-6100 X-ray Diffractometer
with Cu Kα radiation (λ = 1.54056 nm) and a 2θ angle in the range of 10° to 80° at a speed of 1° min$^{-1}$.



## 3 Results

### 3.1 The growth of cyanobacteria culture

Cell growth was monitored over the course of 20 days. After the inoculation, cells did not exhibit an exponential phase for 4~5 days (Fig. 4a). For one treatment, cyanophage was inoculated to allow the adsorption and infection of cyanobacteria. The cell abundance of the viral group is slightly lower than the non-treated group (Fig. 4a). Although the cell abundance increases at the first few days, the growth rates of viral treatment are slightly lower and the cell number is reduced to $1.7 \times 10^7$ cell/ml compared to the $1.3 \times 10^8$ cell/ml of the non-virus treatment. The color of the culture medium varied daily between the two treatments after viral induced lysis. On the 13$^{th}$ day, viral treatment began to coagulate. The supernatant of the viral treatment became clarified on the 14$^{th}$ day and seemed completely clear on the 17$^{th}$ day. The non-virus treatment, in contrast, has some turbidity and higher cell density. By the time cells were deposited, the white precipitation phase emerged in the virus treatment (Fig. 3c, d).

### 3.2 Change of carbonate parameters

The carbonate chemistry showed similar patterns during the early and mid-exponential growth phases (< 10days) but started to deviate strongly in term of the total alkalinity (TA) and dissolved inorganic carbon (DIC) after the cell lysis rates were greater than cell replication rates (Fig. 4b, c). The change of DIC was well-coordinated with the cell growth. When cultures were at the end of exponential phase (days 14), the DIC declined to the lowest. However, when cells lysed by the phage, the DIC rose again to the initial level because of reequilibration with atmosphere and increase of PCO$_2$. TA also dropped from the initial concentration of 3866 μmol/kg to the 1252 μmol/kg at the exponential phase. During the lytic phase, TA increased again to values of 2936 μmol/kg in three days and kept balance during the lytic cycle. Compared to the TA and cell lysis, there is an early TA increase. Both calcium and magnesium cations were removed from the solution at the early-exponential growth phases (Fig. 5). It is interesting to find that precipitated cations re-dissolve to the solution in non-infection treatment. In striking contrast, there is a persistent calcium removal within the viral lysate, indicating that robust viral induced calcification had happened (Fig. 5 a, b).

### 3.3 Microstructure of viral induced carbonate precipitation

SEM and TEM images of the white precipitates from the viral lysate show numerous calcium nanoparticles scattered or aggregated. These particles are in a spherical morphology, having diameters ranging from dozens of nanometers to hundreds of nanometers (Fig. 6, 7, 8). STEM mappings and XEDS analysis showed that there is evidence of calcium accumulation all around the particle surface (Fig. 7) and selected-area electron diffraction (SAED) patterns with their diffuse halos (Fig. 6b), confirming that they are amorphous calcium carbonate (ACC). Although Mg is not dominant in the particles, there were signs of enrichment of Mg around the particles (Fig. 7c). SEM images of the nano-particle are attached to the surface of the infected cells and usually have an encrusted structure (Fig. 8). However, occasionally, the inner core of the nanoparticles may lose (Fig.



8 d). The bulk mineralogy of the cultural deposits, based on XRD analyses, were dominated by brucite in the non-infection treatment compared to aragonite in the lysate (Fig. 9).

## 4 Discussion

### 4.1 Carbonate chemistry influenced by viral lysis of cyanobacteria

Various studies in recent years have demonstrated the direct effects of carbonate chemistry shift over the course of cyanobacteria growth (Dittrich et al., 2003;Kranz et al., 2010;Millo et al., 2012;Obst et al., 2009b;Yang et al., 2016). In cases where photosynthesis occurred, it results in the stimulation of cell division and DIC uptake, but no alkalinity changes because no other sources of base are added as follows:

$$HCO_3^- + H_2O \rightarrow CH_2O + OH^- + O_2 \hspace{3cm} (2)$$

Studies of cyanobacteria calcification always attribute the increase of pH to the growth of cyanobacteria which construct a favorable calcification environment where carbonate is the dominant inorganic carbon species and induces calcification by the incorporation of carbonate ions into a growing $CaCO_3$ crystal (Lee et al., 2004;Obst et al., 2009a;Kranz et al., 2010). It is interpreted by the majority of research that cyanobacteria calcification is restricted to certain species (Merz-Preiß, 2000;Lee et al., 2004). The mineralization induced by photosynthetic acid-base equilibrium by *Synechococcus* spp. in the

present study seems to be transitional and unable to calcify to the extent that a stable $CaCO_3$ precipitate was formed. It is inferred based on the observation that cations are released to the solution in non-infection treatment. Fixed $Ca^{2+}$ redissolves to the concentration equivalent to the former concentration and $Mg^{2+}$ partial release (Fig. 5a, b).

Magnesium, which is actually precipitated from the solution, is responsible eventually for removing TA from the cyanobacteria culture:

$$Mg^{2+} + 2OH^- \rightarrow Mg(OH)_2 \hspace{3cm} (3)$$

As DIC transport by cyanobacteria proceeded, the pH of the growth medium increased, leading to the formation of $Mg(OH)_2$ under the supersaturate state (Equation 2 and 3). Saturation indices (SI), which are determined using the software PHREEQC, yielded values > 0 for brucite during the first 8 days (0.34 ~ 1.15) and values < 0 after the 10th day (-1.46). There is no surprise that biologically mediated brucite crystal formation emerges under high pH and low $pCO_2$

microenvironments created by cyanobacteria, which are similar to coral microbial biofilms (Nothdurft et al., 2005) and cultures of the diatom (Tesson et al., 2008).

In regard to calcium, despite the SI $_{aragonite}$ >0, the formation of stable carbonate (aragonite or calcite) was documented as improbable because of saturation states that could not develop or persist to overcoming the activation energy barriers for nucleation (Morse and He, 1993). It seems that the two treatments of the tested culture grew at similar rates and reached

similar cell densities during the first 10 days despite the inoculation of cyanophage (Fig. 4a). However, when lytic rates run



over the bacterial replication, $Mg^{2+}$ has been recovered to the initial level, but the further removed $Ca^{2+}$ was simultaneously present with distinct variations compared to the non-infection treatment. There is a strong positive correlation between $Mg^{2+}$ and DIC recovered after the $12^{th}$ day, which is the time point when the lytic rate begins to dominate cell replication. In present open system, atmospheric $CO_2$ is dissolved in water and changes the $PCO_2$ level and acid-base balance of the system.

Hence, some of unstable mineral phases can dissolve with acidification during the culture of cyanobacteria:

$$Mg(OH)_2 + H_2CO_3 \rightarrow Mg^{2+} + HCO_3^- + 2H_2O \quad\quad\quad (4)$$

$$CaCO_3 + H_2CO_3 \rightarrow Ca^{2+} + 2HCO_3^- \quad\quad\quad (5)$$

However, it should be noted that, with the aid of the virus, calcification is consistent and more stable aragonite minerals are formed. Thus, a microenvironment favorable for calcification is available after the viral lysis of cyanobacteria and stable

carbonate minerals are deposited.

## 4.2 Precipitation of carbonate mineral induced by virus lysis of cyanobacteria

The precipitate was investigated by means of XRD showing particles that can be described best as aggregates of aragonite in viral lysate and brucite in bacterial culture (Fig. 9). XRD results combined with chemical parameters change of the non-infected culture revealing that, despite the chemical environment, it may be favorable for calcium nucleation during

the growth of cyanobacteria, which is unable to calcify to the extent that a stable $CaCO_3$ precipitate was formed. Only partial $Mg^{2+}$ may precipitate under the supersaturated state with the fact that both bivalent cations ($Mg^{2+}$, $Ca^{2+}$) cycled to the dissolved phase when the growth curves are in stable phase (Fig. 4a, b).

It has been extensively investigated that various physicochemical factors control the formation of the $CaCO_3$ polymorph and aragonite tends to precipitate under a high molar ratio of Mg/Ca (Folk, 1974;Berner, 1975). The microenvironment

maintained by the growth of *Synechococcus* spp. could not overcome the activation energy barriers for the nucleation of aragonite. TEM images revealed no order in the majority of the detected particles, which was confirmed by the diffuse rings in the selected area electron diffraction patterns but the appearance of nanodomains within the ACC particles (Rodriguez-Blanco et al., 2008). ACC, which received relatively little attention as one of metastable $CaCO_3$ phases, was increasingly recognized as prevalent during calcification (Blue et al., 2017). ACC may precipitate when the conditions promote high local

supersaturations for short periods of time (Blue et al., 2017). Although the saturation index (SI) of the ACC <0 (Table 2), implying the nonspontaneous ACC formation within the solution, the growth of cyanobacteria in the present experiment created an ACC favorable microenvironment at the $10^{th}$ day reflected by the removing of $Ca^{2+}$. Consequently, with the growth of cyanobacteria, carbonate alkalinity limitation leads to the redissolve of ACC. However, with the aid of the viral cycle and the lysis of the host, the dissolution of carbonate seemed not to happen, and a more stable mineral formed.

Although the theoretical calculation proposed by Lisle and Robbins (2016) indicates viral induced rupturing of cells released by cytoplasmic-associated bicarbonate, thereby dramatically reducing the activation energy for nucleation of carbonate polymorphs, experimental-based processes for the nonclassical multistep calcification is lacking. The lysogenic cycle of



*Synechococcus* spp. does not result in immediate lysing of the host cell, but dormancy by integrating their genome with host DNA and replicating along with it relatively harmless. When the reproductive cycle initiates, the virus attacks and breaks down the cell wall peptidoglycan, which is an essential structure that protects the cell protoplast from mechanical damage and from osmotic rupture (Middelboe and Jørgensen, 2006). Thus, $Ca^{2+}$ has unfettered access to the intracellular space and

reacts with cytoplasmic alkalinity (Fig. 10).

Preliminary investigations have also demonstrated that viruses from the hypersaline lake occur and are incorporated in biogenic carbonate, suggesting that virus may be mistaken for nanobacteria and play a role in initiating mineralization (De Wit et al., 2015;Pacton et al., 2014;Perri et al., 2017). The viral drive during the biogenic carbonate precipitation in hypersaline lakes is attributed to either an indirect route, involving silicified viruses as an intermediate phase during diagenesis (Pacton et

al., 2014) or a direct incorporation of amino acids polymerized with viral proteins into growing high-Mg calcite crystals (De Wit et al., 2015). The encrusted structure indicated by SEM images may support the heterogeneous nucleation of carbonate formation (Fig. 8). When $Ca^{2+}$ access to intracellular space, the existence of capsid synthetized by viral DNA provides nucleation site for initial calcification (Fig 10).  This heterogeneous calcification involves addition of material to the preexisting viral surface, which may similar to mineralized virus in microbial mat from hot spring (Peng et al., 2013) or  hypersaline lakes

(Pacton et al., 2014;De Wit et al., 2015;Perri et al., 2017). The pathway that viral induced calcification during the lysis of the host cells retrieved by experimental study expands roles of viruses in marine carbonate precipitation.

### 4.3 Significance of viral induced carbonate precipitation

Owing to the fact that biologically mediated $CaCO_3$ precipitation is one of the fundamental processes in the carbon cycle (Ridgwell and Zeebe, 2005;Planavsky et al., 2009;Riding, 2011;Kamennaya et al., 2012), the study of viral impact is important

toward understanding carbon cycling on both the global and regional scales. The so-called "Whiting events", which refers to an event of high levels of suspended, fine-grained $CaCO_3$ precipitation, have long been a spectacular and extensively investigated $CaCO_3$ precipitation event. Although whiting events were recognized to be of abiotic origin, mechanisms of formation and maintenance of "Whiting" are still debated (Wright and Oren, 2005;Morse et al., 2003). Regarding the fact that cyanobacteria are the most abundant primary product and play important roles in the global carbon cycle, biologically- induced

$CaCO_3$ formation is much more frequent than the abiotic processes under different conditions of seawater chemistry in the geological past (Obst et al., 2009b;Riding, 2011). The influence of photosynthesis on $CaCO_3$ precipitation in general is based either on the uptake mechanism of inorganic carbon (Riding, 2006;Riding, 2011) or surface-induced mechanisms involving the cell surface or extracellular polymers as preferential sites for mineral nucleation (Obst et al., 2009b). With regard to viruses numbering, usually ten times more than their hosts (Suttle, 2005, 2007), viruses have been recognized as new agents of the

nucleation site for the carbonate mineral (De Wit et al., 2015;Perri et al., 2017) and can influence carbonate chemistry notably (Lisle and Robbins, 2016). However, the viral influence on the "whiting event" is hardly know. Clear evidence of net carbonate precipitation from the waters culturing cyanobacteria and cyanophage suggest that release of the virus during plankton bloom may stimulate viral induced $CaCO_3$ precipitation, representing one potential whiting mechanism for $CO_2$ sequestration.





Plankton blooms are frequently known in few recent years as a result of extensive accumulation of algal and cyanobacteria population impulses by nutrient enrichment. Cyanophages are vital parasites of unicellular marine cyanobacteria modulating microbial production and, in some cases, terminating plankton blooms (McDaniel et al., 2002;Bratbak et al., 1996;Bratbak et al., 1993). Experimental studies proposed in the present work give a glimpse of carbonate precipitation during plankton blooms
influenced by viruses and reinforced the great importance of viruses in global geochemical cycles.

Mg/Ca in benthic foraminiferal calcite have often been used as proxies to reconstruct the paleoenvironment. The basis for the Mg/Ca paleothermometry method lies in the determination that past seawater Mg/Ca (Lear et al., 2000). The reconstruction of seawater chemistry reveals that the Mg/Ca ratios were lower from the mid-Jurassic to the Oligocene than at present (Coggon et al., 2010). The higher Mg/Ca is also reflected by the greater abundance of aragonitic non-skeletal and biogenic marine
carbonate since the early Cenozoic (Stanley and Hardie, 1998). The processes responsible for the increase in the seawater Mg/Ca ratio during the Oligocene were concluded to river discharge, sediment burial and hydrothermal exchange (Coggon et al., 2010). Cyanobacteria, which were recognized as the most ancient group, were responsible for the original oxidization of the Earth's atmosphere and dominated the elemental cycles over geological time scales (Des Marais, 2000;Hohmann-Marriott and Blankenship, 2011). Virologists have placed the viruses in the earliest phases of life's evolution and associated them with
the primitive precursors of the cellular systems (Holmes, 2011). In view of the marine viral induced carbonate deposit and increased Mg/Ca ratio of the medium (Fig. 5c), the possibility that viral processes alter the paleoenvironment is thus important to evaluate.

## 5 Conclusion

First, we provide here a detailed view of carbonate chemistry change, mineral composition during viral infection and lysis
of cyanobacteria. Amorphous calcium carbonate and aragonite were evident in the lysate, which differed substantially from no calcification in the non-infection culture. We also inferred that not only photosynthetic acid-base equilibrium influence the carbonate chemistry; but that viruses may also play important roles in either releasing cytoplasmic alkalinity or acting as nucleation sites to balance the carbonate system. Altogether, our results expand the viral role in mediating sea water carbonate systems and provide new insights in certain aspects of the global geochemical process.


## Acknowledgments

Authors would like to acknowledge Dr. Dezhang Ren at Shanghai Jiaotong University for his help with the XRD analysis. The authors are also indebted to Dr. Weijia Zhang at the Institute of Deep-sea Science and engineering for her help with the cyanobacteria identification. Financial support for this research came from "Strategic Priority Research Program" of the
Chinese Academy of Science (Grant No. XDB06020000), "National Key Basic Research Program of China" (2015CB755905), and the "Natural Science Foundation of Hainan Province, China" (20164175).




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




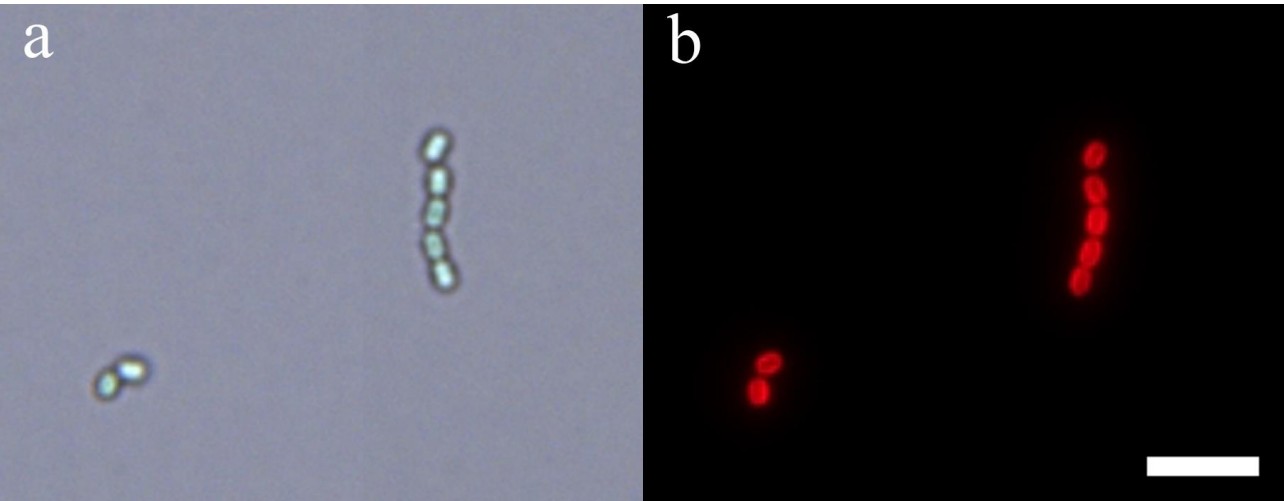

**Figure 1: Strain of *Synechococcus* spp. isolated from Sanya Bay. Scale bar = 10 µm.**

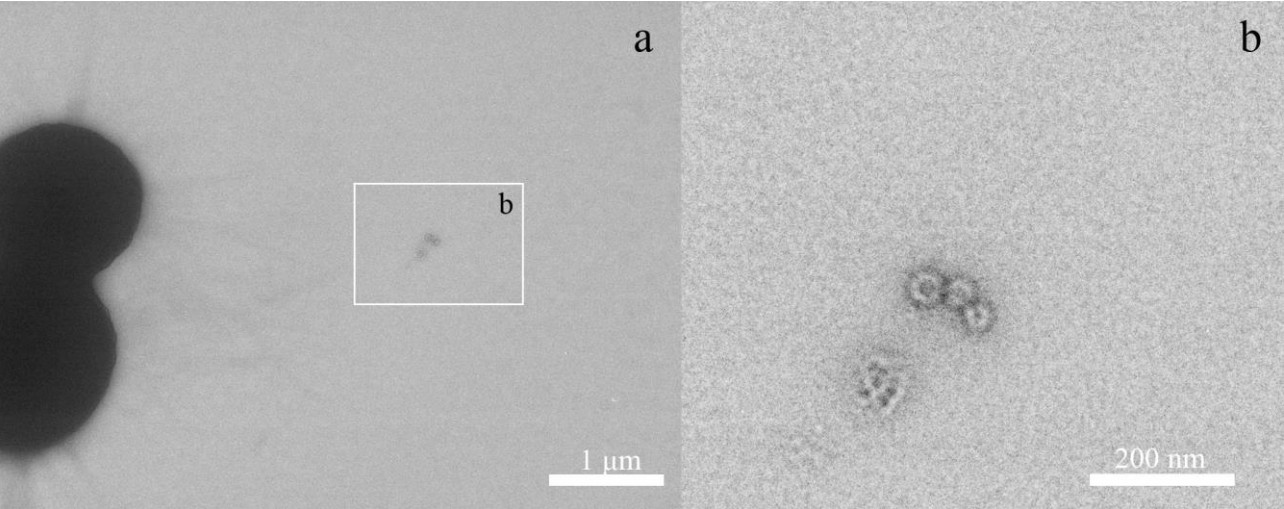

**Figure 2: Cyanophage isolated from Sanya Bay. (a) Cyanophage will infect the host cell of Synechococcus spp. (b) enlarged view of**
5 **the cyanophage particles.**



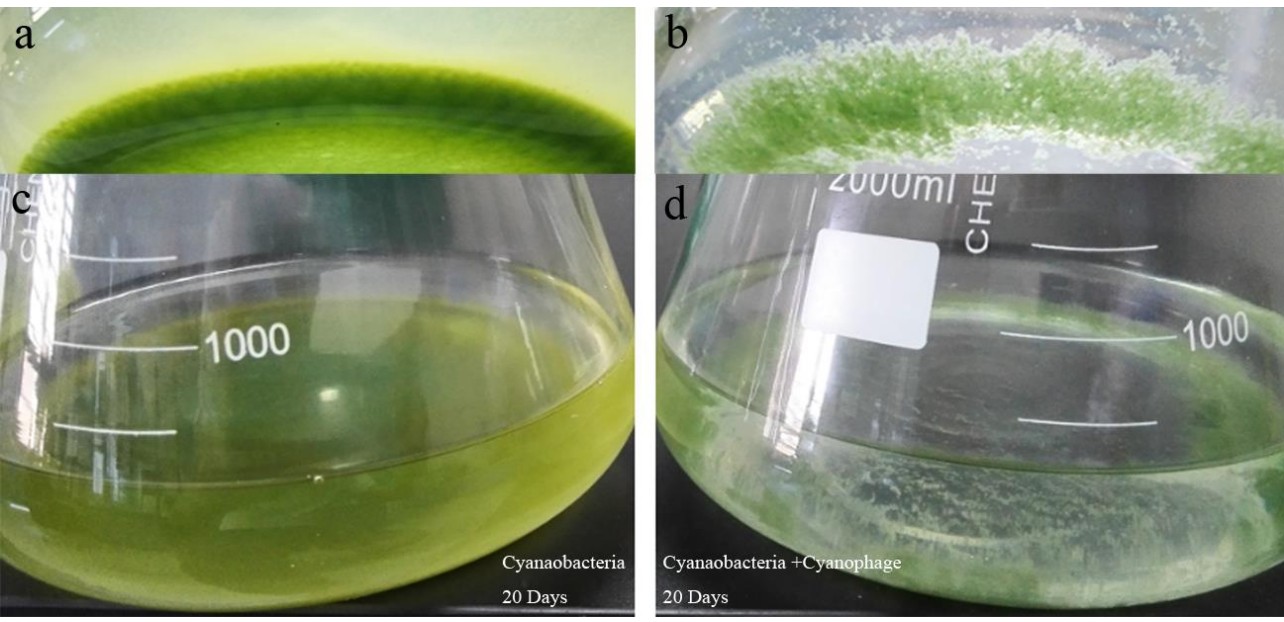

**Figure 3: Photos of the culture medium at the end of the experiments. (a and c) without inoculation of the cyanophage. (b and d) white precipitates are evident in viral lysis of cyanobacteria.**

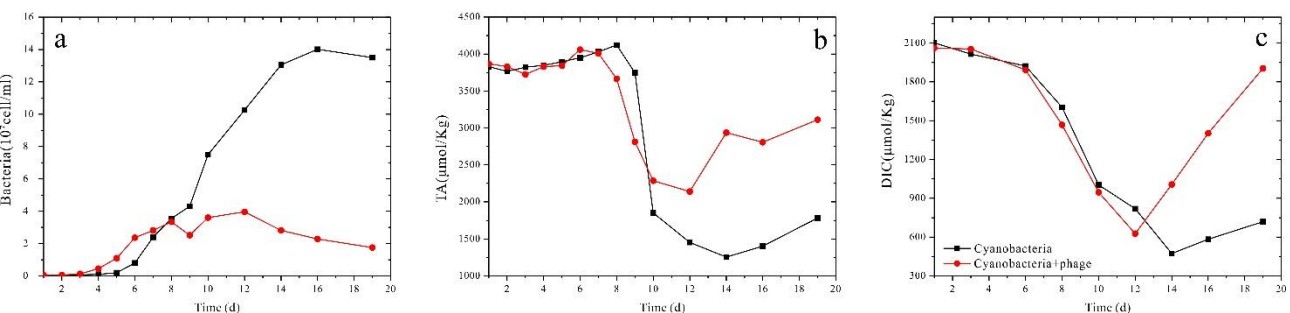

5 **Figure 4: Changes in the solution bacteria concentration (a), total alkalinity (b) and dissolved inorganic carbon (c)**

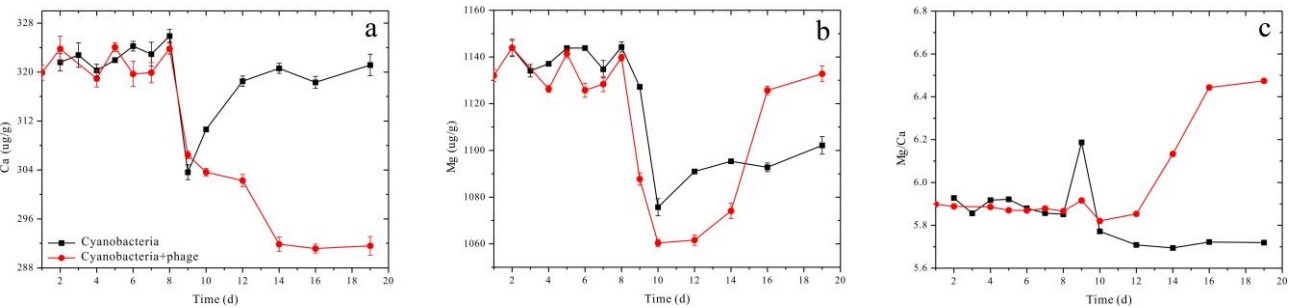

**Figure 5: Changes in the solution calcium concentration (a), magnesium concentration (b) and Ma/Ca atomic ratio (c).**

**Figure 6:** **TEM images show the crystallization of ACC nanoparticles in the viral lysate. (a) Nanoparticles with a diameter approximately 50-200 nm are scattered in the viral lysate. The insert in the bottom right image (b) shows a selected area electron diffraction pattern of ACC, revealing only diffuse rings related to poorly ordered materials (c) and (d) Enlarged view of ACC nanoparticles showing that some of them aggregate together.**





**Figure 7: Chemical composition of ACC nanoparticles. (a) STEM images showing ACC nanoparticles. (b, c, d) XEDS maps of Ca, Mg and Si respectively showing that ACC is composed mainly of Ca.**





**Figure 8:** SEM-BSE photomicrographs of nano-particles at the surface of the host cell of Synechococcus spp. (a, b) host cells infected by cyanophage and mineral particles are evident at the surface of the cell. Scale bar = 1 µm. (c, d, e) nano-particles with an encrusted structure. The inner core occasionally has been discarded leaving the broken, extracellular mineral crust (d). Scale bar = 200 nm



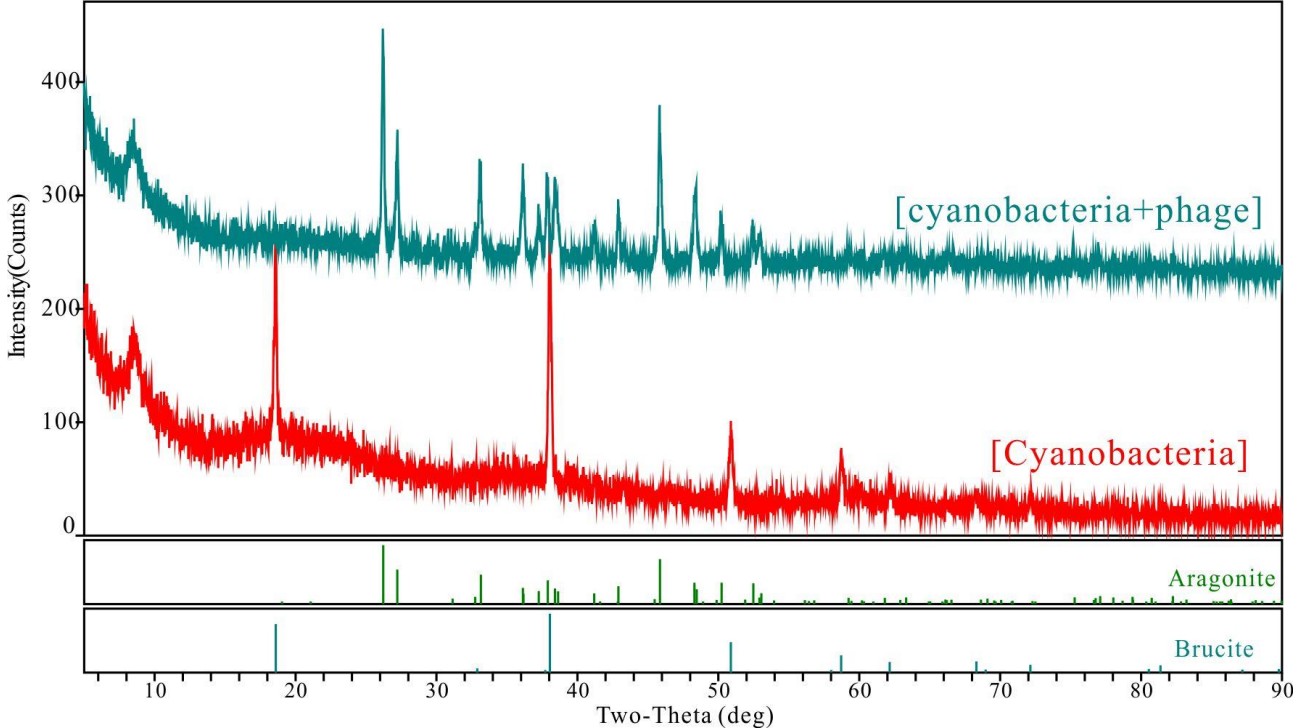

Figure 9: Typical X-ray diffraction patterns collected for each polymorph that formed in this study.





**Figure 10:   Formation model of the carbonate mineral induced by viral lysis of the cyanobacteria. Although the chemical environment may be favorable for mineral nucleation during photosynthesis, it is unable to mineralize to the extent that stable CaCO3 precipitates are formed. During the viral lysis of cyanobacteria, calcification was initiated by rapid intracellular calcification because of the unfettered intracellular access of Ca2+ reacts with cytoplasmic alkalinity after virus attacks and breaks down the cell wall. Meanwhile, more available nucleation sites (e.g., viral capsid) also stimulate the calcification.**





**Table 1 Composition of the artificial seawater (modified F/2 media).**

| Chemical | Amount (g/L) | Chemical | Amount ($10^{-5}$ g/L) |
|---|---|---|---|
| NaCl | 21.19 | $CuSO_4 \cdot 5H_2O$ | 0.98 |
| $Na_2SO_4$ | 3.55 | $ZnSO_4 \cdot 7H_2O$ | 2.20 |
| KCl | 0.60 | $CoCl \cdot 6H_2O$ | 1.00 |
| $NaHCO_3$ | 0.29 | $MnCl_2 \cdot 4H_2O$ | 18.0 |
| KBr | 0.09 | $Na_2MoO_42H_2O$ | 0.63 |
| $H_3BO_3$ | 0.02 | $Na_2MoO_4 \cdot 2H_2O$ | 436 |
| NaF | 0.003 | $FeCl_3 \cdot 6H_2O$ | 315 |
| $MgCl_2\ 6H_2O$ | 9.59 | vitamin B1 | 0.01 |
| $CaCl_2$ | 1.01 | Vitamin Biotin | 0.00005 |
| $SrCl_2\ 6H_2O$ | 0.02 | Vitamin B12 | 0.00005 |
| $NaNO_3$ | 0.075 | | |
| $NaH_2PO_4\ H_2O$ | 0.005 | | |
| $Na_2SiO_3\ 9H_2O$ | 0.03 | | |

**Table 2 Saturation indices calculated from culture system.**

| | Time (d) saturation index | 6 | 8 | 10 | 12 | 14 | 16 |
|---|---|---|---|---|---|---|---|
| | pH | 9.56 | 9.96 | 8.66 | 8.95 | 9.32 | 9.29 |
| | Aragonite | 1.34 | 1.35 | 0.56 | 0.69 | 0.65 | 0.73 |
| Cyanbacteria | Brucite | 0.34 | 1.15 | -1.47 | -0.89 | -0.15 | -0.2 |
| | ACC | -0.72 | -0.71 | -1.50 | -1.37 | -1.41 | -1.33 |
| | vaterite | 0.92 | 0.93 | 0.14 | 0.27 | 0.23 | 0.31 |
| | pH | 9.65 | 9.85 | 8.87 | 9.89 | 9.93 | 9.74 |
| | Aragonite | 1.36 | 1.3 | 0.68 | 0.92 | 1.12 | 1.1 |
| Cyanbacteria + Phage | Brucite | 0.52 | 0.92 | -1.06 | 0.93 | 1.06 | -0.02 |
| | ACC | -0.70 | -0.76 | -1.38 | -1.14 | -0.94 | -0.96 |
| | vaterite | 0.94 | 0.88 | 0.26 | 0.50 | 0.70 | 0.68 |