# Peer review of "Precipitation of Calcium Carbonate Mineral Induced by Viral Lysis of Cyanobacteria"

_Biogeosciences, 2018_

## Referee Comment (RC1) · Anonymous Referee #1 · 15 May 2018

This study deals with an interesting topic, providing an experimental test of the hypothesis that lysis of cyanobacterial cells by virus may trigger Ca-carbonate precipitation by helping overcoming the energy barrier of nucleation. If true, this means that such biological events may change the apparent solubility of carbonates in eg seawater. While the study overall provides some results which are convincing to me, there are several flaws which need to be corrected first before acceptance for publication. The most important ones are: 1) there is a strong incoherence between XRD results showing crystalline aragonite and TEM data suggesting ACC. TEM data should be revised. 2) It is not convincingly explained what happens around day 8 and how DIC and total alkalinity follow such different paths. 3) The Mg story is really not convincing. How can you explain that the solution goes from supersaturated to undersaturated with brucite. I de-

tail thereafter these comments and add several other ones which should be addressed

Introduction: L29: sentence is awkwardly written. It should read "Dissolved inorganic carbon in the typical..." (dissolved CO2 is not HCO3-)

Page 2: formation and dissolution of carbonate is one of the most instead of the most. I guess photosynthesis is at least as important L4: Needs rewritting "seawater is considered supersaturated with several calcium carbonate phases such as xx (calcite?), with saturation index..." a solution is supersaturated with. A carbonate is not

L13 : you do not need to get into that debate about whether cyanobacteria formed the first stromatolites or not. Many studies now argue against this idea and I agree this is beyond the scope of your paper to debate about that. You should rephrase

L18: Instead of furthermore, I would write "In contrast", since it has been suggested that intracellular precipitation might be controlled in opposition to non-control as mentioned on line 16. And I would remind that this is true for some species of cyanobacteria, not all. Last, it was recently showed that it can occurs in undersaturated solutions (Cam et al., 2018 Geobiology). I am wondering if the same could be imagined in some environments with viruses lysing cyanobacteria. Maybe as a perspective?

L30: you should specify that this increase would be very local

Page 3, L19: Synechococcus spp.This is such as broad name encompassing so different bacteria. Could you specify at this point the name of the strain? Or specify that the strain was isolated in the present study?

P4: Is the strain axenic?

L7: I guess the other not-mentioned treatment is a control where no virus has been added? Please specify this

Are culture bottles closed to air exchange or are they open? This is important to understand the evolution of DIC in your system I guess

L8-9: please rephrase. I guess you do not measure OD on a fixed and filtered suspension. And Why do you fix the cells before measuring the chemical composition of the solution. Could fixation modify the chemical composition of the solution?

L11: what is a TA sample?

2.3: Electron microscopy instead of electronic microscope

L19: what is the stable phase? Do you mean stationary phase (after exponential)? But what do you mean by the end of it?

L26 what is abs?

L27: Technically you do not sputter coat with carbon. This is achieved by evaporation. You can rewrite as: before being carbon coated

Page 5: L 3-4: the lag phase lasted 4-5 days

L5: you say slightly lower but I see on Fig 4a a DO of 3 vs 14. This sounds like a very big difference to me

Fig. 4a: you said you ran duplicates. Where are the error bars then in your curves?. Could you show the pH curve?

Why TA does not match with alkalinity? Which other species contribute to alkalinity here? Do you think that they als might vary differently? I think of N species? Or P species? TA sharply decreases at day 8 while alkalinity starts decreasing at day 6. How do you explain that?

Why is there such a sharp decrease of Ca in all conditions at day 8? Why does dissolved Mg increase again in viral treatment after 14 days? L21: you do not mention that Mg also redissolve in the viral treatment.

Fig 6, caption: by definition ACC is not crystalline so you should not write about crystallization of ACC but precipitation or formation

Fig 7: please show a spectrum. Maps are not enough. Why is these are phosphates? Only spectra could show that there is no P peak

Fig 9: what is the peak at 10°? If XRD sees aragonite (ie crystalline phase) how does it come that you see amorphous Ca-carbonates? Is aragonite amorphizing under the TEM beam?

Discussion: P25: the authors claim that this is no surprise that brucite forms but to my knowledge this has never been really shown by previous studies on cyanobacteria cultures; How do they explain that they produced brucite and not the other groups. Moreover, they detect brucite by XRD but they mention that the solution became undersaturated with brucite after 8 days. How is it possible. I would expect in the worst case a SI of 0. Not below. I see that Mg is released after 8 days even in the viral treatment. This could be consistent with the undersaturation of the solution with brucite. But how do we switch from supersaturated to undersaturated? Precipitation of brucite should take it to saturated. And since there is no Mg going to aragonite and there is no carbonate in brucite, this cannot be explained by aragonite precipitation and changes in DIC.

End of discussion is too long. The paragraph on page 9 from L 6 to 17 could be skipped or at least significantly reduced since this is quite faraway from the main scope of the paper

---

## Referee Comment (RC2) · Anonymous Referee #2 · 16 May 2018

The manuscript by Xu, et al. describes the isolation of a cyanobacterium and cyanophage and the use of this host/bacteriophage system to induce mineral precipitation in artificial medium under laboratory conditions. Using observational data, mineral characterization technologies, microscopic cyanobacterial counts and chemical analyses the authors conclude the presence of cyanophage in a culture of the host cyanobacterium lyses the host and releases cellular constituents into the culture medium. The author propose this release of dissolved and particulate cellular constituents promotes the precipitation of specific polymorphs of calcium carbonate and magnesium hydroxide.

General Comments: 1. The manuscript is not well organized, with some sections lacking adequate methodological information. Collectively, these issues makes the

manuscript a bit difficult to read and interpret. Examples will be specifically described in the Specific Comments section.

2. The use of "calcium carbonate" is used throughout the manuscript but in most instances, this is too general a descriptor within the context of this study. The use of the specific calcium carbonate polymorph names [e.g., amorphous calcium carbonate (ACC), vaterite, aragonite, calcite] will be more appropriate when applicable.

3. The mineral precipitation mechanisms of homogenous and heterogeneous nucleation are not clearly delineated throughout the manuscript and appear to be used interchangeably in some instances. Additionally, mineral nucleation and precipitation are also used interchangeably. This conflation of terms, phrases and concepts makes it more difficult for the reader to read and interpret the manuscript. The experimental design for this study does not allow the detection or characterization of nucleation events, only gross precipitation that can only be indirectly assumed the result of one or both mechanisms of nucleation.

4. Throughout the manuscript total alkalinity (TA) and alkalinity are used interchangeably. When working with marine carbonate chemistry there is a significant difference between the ways these two alkalinities are calculated. Make it clear to the reader which one is being used or referred to.

5. There is repeated mention of carbonate chemistry in the text and tables. However, there is no listing of the geochemical data used for or the output from the geochemical modeling analyses. These data sets need to be included.

6. Saturation indices (SI) are a central component of this manuscript but there is no description of how these were calculated. When working with carbonate chemistry in marine waters most calculations on saturation states are per polymorph, like aragonite and calcite. These calculations are not normally performed within commonly used geochemical modeling programs but rely on CO2SYS or a recently developed application, CO2calc. The calculated SI values from the geochemical modeling software and

CO2SYS/CO2calc are not always equivalent.

7. The Discussion section is too long and not focused on placing the data and interpretations from this study in context of previously published papers. There is a considerable amount of text dedicated to introducing and developing concepts that are on the periphery of the stated objectives and generated data of this study. This section needs to be edited to remove these passages and re-written to focus the discussion in a focused and concise style.

Specific Comments:

Abstract

Pg 1: Ln 10-22. (1) The data do not support the statement that the presence of viruses stabilizes the carbonate minerals detected in this study. (2) There's no evidence in this study of rapid intracellular calcification due to intracellular calcium concentrations.

Introduction

Pg 2: Ln 1-2. The precipitation and dissolution of calcium carbonate does later sea water chemistry but do these processes have a more significant influence on the carbonate chemistry than carbon dioxide flux from the atmosphere?

Pg 2: Ln 11-15. This passage describes sedimentary processes and stromatolites. Consideration should be given to removing this passage from the manuscript. This concept is not relevant to the stated theme of this manuscript.

Pg 2: Ln 16. The precipitation of calcium carbonate, regardless of the mechanism, is always a controlled geochemical or biogeochemical process.

Pg 2: Ln 23-29. All of the information in this passage is true. However, it's not clear how this information is relevant to the stated objectives and experimental design of this study. Considering should be given to removing this passage from the manuscript.

Pg 2: Ln 33. Which of the calcium carbonate polymorphs was capable of homogenous

nucleation in the cited study?

Pg 3: Ln 1-2. The authors state the cited study does not consider the role of magnesium in their calculations. However, they don't tell the reader why it's important in this study. Since the role of magnesium in mineral precipitation is one of the objectives, this would be the place to provide the reader with some background information that will put the data and interpretations in the proper context.

Pg 3: Ln 5-14. The information in this passage is not relevant to this study based on the stated objectives and experimental design. Its inclusion is a distraction from the Consideration should be given to removing this passage from the manuscript.

Pg 3: Ln 17-18. (1) The authors state the understanding of viral influences on the precipitation of carbonate is poorly understood but they in the previous 16 lines of text they list several published studies that do characterize this process. (2) Here is an example of the confusing use of nucleation and precipitation in the same sentence.

Pg 3: Ln 30. (1) A virus infected cyanobacterium is not a cyanophage. (2) Interpreting Figure 2 as showing a cyanophage and its host bacterium is a bit of a reach. These images could be almost anything.

Pg 3: Ln 31-32. There needs to be, at a minimum, an abbreviated description of the cited method used to isolate, purify and identify the cyanobacterial specie. It should not be incumbent on the reader to run the most basic description of a method.

Pg 4: Ln 1-2. (1) As noted in the previous comment, at a minimum, an abbreviated description of the cited method for the cyanophage isolation needs to be provided. (2) There is a reference to metagenomics analysis, including a supplemental data file. The method of sample collection, processing, sequencing and sequence analysis (including the bioinformatics) is not mentioned or described anywhere in this manuscript. This is a significant deficiency in this version of the manuscript.

Pg 4: Ln 3-17. (1) This section is very poorly organized and developed with respect to

the different methods mentioned. There is no reasonable way a person could replicate this research effort or interpret the data from these methods using this section for guidance. (2) In this reviewer's opinion there are seven distinct methods: culture growth conditions; cyanobacteria counts; ion chromatography; salinity, total alkalinity and DIC measurements; geochemical modeling of carbonate chemistry; metagenomics analysis. Consider giving each of these their own sub-section within Experimental Setup and develop each section so the reader will understand how the methods were performed. (3) List the incubation or experimental times for each experiment type. (4) What is meant by a "pre-culture"? (5) A "one treatment" is mentioned. Are there other treatments? If so, describe those treatments and their respective differences. (6) List the volumes for each experimental container, the volumes of the sub-samples and the times at which the sub-samples were collected. (7) For the cyanobacterial growth cultures provide the light wavelength and dose.

Pg 4: Ln 15-17. There is reference to geochemical modeling using a specific program. However, there is no mention or listing of the data on which the geochemical analyses were performed or the outputs from those analyses (e.g., activities of carbonate species). Both of these data sets need to be included in this manuscript. Additionally, the PHREEQC code used for these analyses needs to be included, most likely as part of the Supplemental Data files.

Pg 4: Ln 19. Here and in several later passages, there are references to "phases" of the cyanobacterial cultures. Based on the references it's assumed these phases are similar to those commonly measured during the growth of bacteria in the laboratory (i.e., lag, exponential, stationary). However, the methods used to determine these growth phases and the data from those experiments are not provided. This is another method that should be included in the Experimental Setup section.

Results

Pg 5: Ln 4-5. How many cyanophage were inoculated into the cyanobacterium culture?

Were the cyanophage titered? If so, the titer data need to be included. Without these data you cannot know the ratio of host cells-to-cyanophage (i.e., MOI), which has a significant influence on infection rates.

Pg 5: Ln 5-7. (1) Based on the brief description of the counting method and Figure 1b, it appears the cyanobacterial host abundances are based solely on the auto-fluorescence of the photosynthesizing microorganism. How can you be sure that non-fluorescent bacteria are not present in these cultures? Without knowing this, the possibility that some or all of the observed responses are due to a non-host bacterium or bacteria cannot be ruled out. (2) From this passage, it appears the cyanobacterial host abundances in the two culture types were measured using fluorescent microscopy but the cyanophage abundances were not determined in the co-culture. How can you conclude the cyanophage were responsible for any of the observations if there is no indication of how many were added to the host culture at time zero and their abundances at the different time points are not known? These cyanophage abundance counts from the different sub-samples should show increases as the host abundances decrease. Without the cyanophage abundance data you can't support the reductions in host abundances as being solely due to cyanophage induced lysis.

Pg 5: Ln 13-23. (1) This section needs to be revised into a more organized and concise presentation of the geochemical data. It's very difficult to interpret in its current format. (2) Throughout this section there are repeated references to different sub-samples and geochemical data associated with those sub-samples. This suggests sub-samples were collected at specific time points during the incubations and those sub-samples were then analyzed for the presence and concentrations of analytes required for the geochemical modeling of changes in the carbonate chemistry. The times of the sub-samples are not provided, the analytical data from those sub-samples is not presented and the outputs from the geochemical modeling analyses are not listed. Collectively, this information and data have to be included for the proper interpretation of the documented observations. For example, reference is made to seemingly unrealistic changes in total alkalinity, DIC and calcium and magnesium removal over the different growth phases of the host culture. The geochemical modeling data need to be presented to support these observations. (3) Was pCO2 measured during these experiments? If so, this method needs to be included and described.

Discussion

Pg 6: Ln 9. Its not clear what this balanced equation is representing? Its not at all clear from the text how formaldehyde (CH2O) is formed from bicarbonate and water or what the significance of CH2O is to the study described by this manuscript. Was the intention to let CH2O be a general reference to a carbohydrate? Or the dissociation of bicarbonate to carbon dioxide?

Pg 6: Ln 14-17, Ln 21-26; Pg 7: Ln 9-10, Ln 16-17, Ln 19-21, Ln 25-29. All of these passages require the reader to have access to the geochemical modeling input data and output results for each sample before an independent interpretation and evaluation of the observations can be made.

Pg 6: Ln 1 and Ln 32. These are the first time brucite is mentioned, other than the abstract. As this mineral seems to a significant product of the processes described in this manuscript, consideration should be given to bring this mineral and its relative importance into the manuscript in the Introduction and the Materials and Methods sections, including the geochemical modeling sub-section.

Pg 6: Ln 22-23. Here and within other passages later in the manuscript, there is reference to the saturation index (SI) as being the metric for determining the saturation, supersaturation and under-saturation state of the culture medium. Though true, the SI alone will not tell you if a mineral will precipitate or dissolve. For example, if this were unconditionally true then average seawater, which has an SI between 2-3, one or more of the calcium carbonate polymorphs would be precipitating out of solution all the time. Instead, average seawater is metastable with no precipitation, indicating there are more geochemical factors other than SI that dictate if a precipitation even will proceed or not.

Here is another example of where the geochemical modeling data would assist in the discussion of the results from this study.

Pg 8: Ln 1-16 and Ln 24-31; Pg 9: Ln 1-17. These passages do not add anything supportive to the manuscript, based on the stated objectives, experimental design and presented data. Consider deleting these from the manuscript.

Pg 8: Ln 20-23, Ln 28-33. The precipitation of one of the calcium carbonate polymorphs in tropical marine waters, or whitings, has been shown via peer reviewed publications to be driven by biological processes and not the simple physical re-suspension of established carbonate sediments.

Conclusion

Pg 9: Ln 19-20. Without the geochemical data and model outputs a detailed view of carbonate chemistry changes have not been provided. Also, without cyanophage abundance counts the cyanophage infection and lysis of the host cells cannot be definitively stated.

References

Pg 10-13. There are 65 references listed. Based on there being issues with citations being incomplete, having punctuation errors, odd symbols inserted (which may have been a conversion issue) and inconsistency with the DOI information format, there are 36 references that need to be reviewed and corrected if needed. There were too many to list individually.

Table 1. (1) This are several modified f/2 medium formulations, but the recipe listed in this table is not included in those published formulation. If there is a citation for this formulation of f/2 please include it. Also if the mineral and vitamin solutions were purchased then this needs to be noted as well. If these solutions were made in the laboratory, then there are several ingredients that are missing (e.g., EDTA). (2) The final pH (which should be between 7.8-8.2) of the media needs to be included as well.

Table 2: (1) The pH values for all of the samples are relatively high compared to the initial pH of f/2. Were these measured or the products of the geochemical modeling? (2) The higher pH values are unrealistically high for open marine waters but, interestingly, the SI values are all less than those for the two dominant polymorphs (aragonite and calcite) in typical marine water. At these pH values, higher SI values, relative to typical marine water, would be predicted. This is another example of where having the geochemical modeling data is critical for the proper interpretation and assessment of the research data.

Figures 1 and 2. These images do not contribute additional information to the reader. Consider deleting these from the manuscript.

Figure 5. Panel 5c is not cited in the manuscript.

Figure 8. The simple shape of an object is not enough to proclaim it being a microbial cell, virus or encrusted microbial structure. Unless more definitive proof can be provided that will support the declaration that this images are what's written in the legend, consideration should be given to deleting this figure from the manuscript.

Figure 10. This is a nice graphic but it's not explained well in the manuscript and there is a lot of information which is not even mentioned in the manuscript. This figure would be a nice summary graphic but the information in the image will have to be presented in the manuscript.

Supplementary Figure. The information in this figure stands alone because nothing about or within this figure is mentioned or described in the manuscript. In the manuscript's current format, this figure does not support anything presented in the Results or Discussion sections. For this reason, consideration should be given to removing this figure from the manuscript.

---

## Author Comment (AC1) · 5 Jul 2018

This study deals with an interesting topic, providing an experimental test of the hypothesis that lysis of cyanobacterial cells by virus may trigger Ca-carbonate precipitation by helping overcoming the energy barrier of nucleation. If true, this means that such biological events may change the apparent solubility of carbonates in eg seawater. While the study overall provides some results which are convincing to me, there are several flaws which need to be corrected first before acceptance for publication.

Reply: Thank you very much for your appreciation on the overall performance of the research work. Improvements have been done following the suggestion from the reviewer.

[Figure]

The most important ones are: 1) there is a strong incoherence between XRD results showing crystalline aragonite and TEM data suggesting ACC. TEM data should be revised.

Reply: We understand the concern pointed out by the reviewer regarding to mineral composition of the precipitation. For ACC, which is metastable hydrated phase, it is thought to be poorly ordered and lack of distinct peaks within XRD pattern (Rodriguez-Blanco et al., 2008). Thus, bulk samples for XRD analysis are composed by aragonite. ACC in TEM images are confirmed by the diffuse rings in the selected area electron diffraction patterns (Fig. 6b). ACC and aragonite also can be seen from the SEM images. We can add the SEM images as a supplementary figure in revised version.

2) It is not convincingly explained what happens around day 8 and how DIC and total alkalinity follow such different paths.

Reply: Growth of cyanobacteria directly results in the DIC uptake. With regards to the total alkalinity, its changes depend on the other sources of base added to the medium. On day 6-8, two treatments grow at similar rates and reach similar cell density. Moreover, there are no cations removed from the medium. Thus, only DIC decreased and TA values remained relatively constant. As DIC transport by cyanobacterial replication, the pH of the medium increase, leading to the immobilization of calcium and magnesium and total alkalinity decrease delaying.

3) The Mg story is really not convincing. How can you explain that the solution goes from supersaturated to undersaturated with brucite. I detail thereafter these comments and add several other ones which should be addressed

Reply: The geochemical condition of the medium is largely influenced by the growth of cyanobacteria. Photosynthetic bicarbonate uptake and its conversion within the cell to $CO_2$ by carbonic anhydrase lead to increase of pH in the immediate vicinity of the cell (Reviewed by Ridding 2011). In present study, when lytic rates run over the bacterial replication, photosynthesis ceases and atmospheric $CO_2$ dissolve in the water and this

will chang the acid-base balance of the system.

Introduction: L29: sentence is awkwardly written. It should read "Dissolved inorganic carbon in the typical..." (dissolved CO2 is not HCO3-)

Reply: Thanks for your correction. It has been rephrased.

Page 2: formation and dissolution of carbonate is one of the most instead of the most. I guess photosynthesis is at least as important

Reply: Thanks for your correction. It has been rephrased.

L4: Needs rewritting "seawater is considered supersaturated with several calcium carbonate phases such as xx (calcite?), with saturation index..." a solution is supersaturated with. A carbonate is not

Reply: It has been rephrased. "Modern sea water is considered supersaturated with several calcium carbonate phrases such as calcite, with $\Omega$ values ranging from 2 to 4."

L13 : you do not need to get into that debate about whether cyanobacteria formed the first stromatolites or not. Many studies now argue against this idea and I agree this is beyond the scope of your paper to debate about that. You should rephrase

Reply: It has been deleted.

L18: Instead of furthermore, I would write "In contrast", since it has been suggested that intracellular precipitation might be controlled in opposition to non-control as mentioned on line 16. And I would remind that this is true for some species of cyanobacteria, not all. Last, it was recently showed that it can occurs in undersaturated solutions (Cam et al., 2018 Geobiology). I am wondering if the same could be imagined in some environments with viruses lysing cyanobacteria. Maybe as a perspective?

Reply: We agree with the suggestion. We highly appreciate the latest reference shared by the reviewer. Cam et al., (2018) is included as a reference for the introduction of intracellular calcification. We can design some viral calcification experiments in undersaturated medium in the near future to discuss the viral influence in intracellular precipitation.

L30: you should specify that this increase would be very local

Reply: We agree with the suggestion. It has been rephrased. The theoretically saturation index change is at the cellular level.

Page 3, L19: Synechococcus spp.This is such as broad name encompassing so different bacteria. Could you specify at this point the name of the strain? Or specify that the strain was isolated in the present study?

Reply: Cyanobacteria are identified to the group of Synechococcus spp. based on the morphology of cell. DNA work will be carried out to precise the statement in the revised manuscript version.

P4: Is the strain axenic?

Reply: During all the treatment, operations are asepsis strictly.

L7: I guess the other not-mentioned treatment is a control where no virus has been added? Please specify this

Reply: Yes. The other treatment is no inoculated with virus. We re-write completely the "Experimental setup" paragraph to make our statements more accurate.

Are culture bottles closed to air exchange or are they open? This is important to understand the evolution of DIC in your system I guess

Reply: They are air exchangeable. So the DIC increase after the viral lysis of cyanobacteria and inhibition on photosynthesis.

L8-9: please rephrase. I guess you do not measure OD on a fixed and filtered suspension.

Reply: We did not measure OD but enumeration the abundance of cyanobacteria by

autofluorescence.

And Why do you fix the cells before measuring the chemical composition of the solution. Could fixation modify the chemical composition of the solution?

Reply: Although we collect the DIC and total alkalinity subsample in bottles without headspace, carbonate chemistry parameters of solution may change during the storage. So, the subsamples were poisoned with HgCl2. The same methods had been used for seawater sampling and analyses elsewhere (Cao and Dai 2011)

L11: what is a TA sample?

Reply: It is total alkalinity.

2.3: Electron microscopy instead of electronic microscope

Reply: It has been correct.

L19: what is the stable phase? Do you mean stationary phase (after exponential)? But what do you mean by the end of it?

Reply: Yes. Stationary phase is used more common than stable phase in literature. The particulate of

L26 what is abs?

Reply: It refers to "selected-area electron diffraction"

L27: Technically you do not sputter coat with carbon. This is achieved by evaporation. You can rewrite as: before being carbon coated

Reply: The change is done accordingly "The samples were carbon coated and before being sputter coated with carbon for 2–3 min. Samples were examined with an Apreo scanning electron microscope (Thermofisher Scientific)."

Page 5: L 3-4: the lag phase lasted 4-5 days

Reply: The sentence has been rephrased.

L5: you say slightly lower but I see on Fig 4a a DO of 3 vs 14. This sounds like a very big difference to me

Reply: The sentence has been rephrased. "The cell abundance of the viral group is slightly lower than the non-treated group on the fifth day to eighth day."

Fig. 4a: you said you ran duplicates. Where are the error bars then in your curves?. Could you show the pH curve?

Reply: Error bars were added in panel Fig 4 (a, b). DIC samples were measured in one set in other lab, so we cannot show the error bars of DIC but the accuracy of measurements of measuring are given. pH values are calculated from the geochemical program and can showed in revised manuscript.

Why TA does not match with alkalinity? Which other species contribute to alkalinity here? Do you think that they als might vary differently? I think of N species? Or P species? TA sharply decreases at day 8 while alkalinity starts decreasing at day 6. How do you explain that?

Reply: I think the reviewer want to know the reason why TA does not match with DIC during the day 6-8. During growth of cyanobacteria, the DIC drawdown by photosynthetic carbon uptake exceeds the slow reequilibration with the atmosphere. Thus, chemical speciations of the DIC pool shift toward higher $CO_3^{2-}$ concentration and pH. TA will not decrease until the precipitation of Ca and Mg, which is responsible for the base remove from the medium. That explain the DIC decrease first (day 6) and TA decrease later (day 8) by the precipitation of $CaCO_3$ and $Mg(OH)_2$.

Why is there such a sharp decrease of Ca in all conditions at day 8?

Reply: Despite the inoculation of cyanophage, culture with and without cyanophage grow at similar rates and reach the similar cell density during the first 8 days. The DIC drawdown by photosynthetic carbon uptake exceeds the slow reequilibration with

the atmosphere, causing chemical speciation of the DIC pool to shift toward higher $CO_3^{2-}$ concentration and pH. At day 8-10, the geochemical condition influenced by photosynthesis promote the Ca and Mg precipitation.

Why does dissolved Mg increase again in viral treatment after 14 days? L21: you do not mention that Mg also redissolve in the viral treatment.

Reply: There is a strong positive correlation between $Mg^{2+}$ and DIC recovered after the 12th day, which is the time point when the lytic rate begins to dominate cell replication. In present open system, atmospheric $CO_2$ is dissolved in water and changes acid-base balance of the system. Hence, unstable mineral phases (like bructie) can dissolve with acidification during the culture of cyanobacteria. The dissolution of brucite in seawater is also discussed elsewhere, which is responsible for the mineral not well preserved over long time (Nothdurft et al., 2005).

Fig 6, caption: by definition ACC is not crystalline so you should not write about crystallization of ACC but precipitation or formation

Reply: The change is done accordingly

Fig 7: please show a spectrum. Maps are not enough. Why is these are phosphates? Only spectra could show that there is no P peak

Reply: EDS spectra of the nanoparticle shows that a small peak of element P. But P signal from the STEM mapping is not consistence with the nanoparticles. So it is not included in the early version of the manuscript.

Fig 9: what is the peak at 10_? If XRD sees aragonite (ie crystalline phase) how does it come that you see amorphous Ca-carbonates? Is aragonite amorphizing under the TEM beam?

Reply: Based on XRD results, the peak at 10 is loeweite, which is a sodium-magnesium sulphate hydrate [Na12Mg7(SO4)13 15H2O], found only in salt deposits of oceanic origin (Fang and Robinson 1970). Aragonite and ACC can be shown in a figure panel

by SEM. Combine with the SEM and TEM, the formation of ACC during the cultivation can be clarified.

Discussion: P25: the authors claim that this is no surprise that brucite forms but to my knowledge this has never been really shown by previous studies on cyanobacteria cultures; How do they explain that they produced brucite and not the other groups.

Reply: It has been rephrased. We agree with the reviewer that few studies show the precipitation of brucite from cyanobacteria culture. Literature review indicates that brucite precipitation may result from increased pH coupled with increased $Mg2+$ activity and low $pCO2$ (Nothdurft et al., 2006). When the cyanobacteria grow during the log phase, DIC are removed from the solution and the pH of the growth medium increased (Table 2). The chemical change leads to the supersaturate state of brucite (Table 2). Thus, Brucite can formation in our culture.

Moreover, they detect brucite by XRD but they mention that the solution became undersaturated with brucite after 8 days. How is it possible. I would expect in the worst case a SI of 0. Not below. I see that Mg is released after 8 days even in the viral treatment. This could be consistent with the undersaturation of the solution with brucite. But how do we switch from supersaturated to undersaturated? Precipitation of brucite should take it to saturated. And since there is no Mg going to aragonite and there is no carbonate in brucite, this cannot be explained by aragonite precipitation and changes in DIC.

Reply: The story of Mg and brucite have been further discussed in revised manuscript. Saturation indices (SI), which are determined using the software PHREEQC, yielded values > 0 for brucite during the first 8 days (0.34 $\sim$ 1.15) and values < 0 after the 10th day (-1.46). The culture bottle are open to the air. So when DIC uptake by photosynthesis is slow compared to the log phrase, atmospheric $CO2$ is dissolved in water and changes the $PCO2$ level and acid-base balance of the system. Hence, brucite can dissolve with acidification during the culture of cyanobacteria. It is suggested that the

brucite microbialite of Holocene corals may not be preserved over longer time, which is easy to dissolved in undersaturated solution.

End of discussion is too long. The paragraph on page 9 from L 6 to 17 could be skipped or at least significantly reduced since this is quite far away from the main scope of the paper

Reply: We appreciate the suggestion of this part of discussion. In revised manuscript version, they were reduced to a single sentence to expand the importance of viral induced calcification.

Please also note the supplement to this comment:
https://www.biogeosciences-discuss.net/bg-2018-194/bg-2018-194-AC1-supplement.zip

[Figure]

**Fig. 1.** EDS spectra of the nanoparticle shows that a small peak of element P. But P signal from the STEM mapping is not consistence with the nanoparticles.

---

## Author Response (AR1)

*Dear editor:*

*We highly appreciate the opportunity for submitting a revised version of our manuscript. We are thankful for all the valuable comments and suggestions. Here, we submit a thoroughly revised version and marked-up version of our manuscript, which has been modified according to the reviewers' suggestions.*

- *The methods have been re-organized following the reviewer's suggestion. Each of the method is given a sub-section in revised version of manuscript.*
- *The discussion of Mg and carbonate chemistry is improved. Carbonate chemistry changes influenced by growth of cyanobacteria and viral lysis of cyanobacteria are discussed in two separate chapters.*
- *The paragraph about whiting and Mg/Ca ratios are deleted following the advice from reviewers.*

*Below we have pasted in the entire review, and we have inserted our responses to the suggestions (blue font).*

*Sincerely,*

*Xiaotong Peng, on behalf of the co-authors.*

**Response to the comments by referee#1**

This study deals with an interesting topic, providing an experimental test of the hypothesis that lysis of cyanobacterial cells by virus may trigger Ca-carbonate precipitation by helping overcoming the energy barrier of nucleation. If true, this means that such biological events may change the apparent solubility of carbonates in eg seawater. While the study overall provides some results which are convincing to me, there are several flaws which need to be corrected first before acceptance for publication.

*Thank you very much for your appreciation on the overall performance of the research work. Improvements have been done following the suggestion from the reviewer.*

The most important ones are: 1) there is a strong incoherence between XRD results showing crystalline aragonite and TEM data suggesting ACC. TEM data should be revised.

*We understand the concern pointed out by the reviewer regarding to mineral composition of the precipitation. There are two carbonate mineral phases (ACC and aragonite) precipitating by viral lysis of cyanobacteria in the medium. The particulate fraction of the medium (~1 L) was harvested via centrifugation (13000 g, 5 min) for electronic microscopy and X-ray diffraction. For ACC, which is metastable hydrated phase, it is thought to be poorly ordered and lack of distinct peaks within XRD pattern (Rodriguez-Blanco et al., 2008). Thus, bulk samples for XRD analysis are composed by aragonite. ACC in TEM images are confirmed by the diffuse rings in the selected area electron diffraction patterns (Fig. 6b).*

2) It is not convincingly explained what happens around day 8 and how DIC and total alkalinity follow such different paths.

*Growth of cyanobacteria directly results in the DIC uptake.*
*With regards to the total alkalinity, its changes depend on the other sources of base added to the medium. On day 6-8, two treatments grow at similar rates and reach similar cell density. Moreover, there are no cations removed from the medium. Thus, only DIC decreased and TA values remained relatively constant. As DIC transport by cyanobacterial replication, the pH of the medium increase, leading to the immobilization of calcium and magnesium and total alkalinity decrease delaying. thus, DIC and total alkalinity changed in different paths.*

3) The Mg story is really not convincing. How can you explain that the solution goes from supersaturated to undersaturated with brucite. I detail thereafter these comments and add several other ones which should be addressed

*Brucite, which is an unstable minera, may not be preserved over long times and can be dissolved in undersaturated state (Nothdurft et al., 2005). The geochemical condition of the medium is largely influenced by the growth of cyanobacteria. Photosynthetic bicarbonate*

*uptake and its conversion within the cell to $CO_2$ by carbonic anhydrase lead to increase of pH in the immediate vicinity of the cell (Reviewed by Ridding 2011). In present study, when lytic rates run over the bacterial replication, photosynthesis ceases and atmospheric $CO_2$ dissolve in the water and this will change the acid-base balance of the system. There is a strong positive correlation between $Mg^{2+}$ and DIC recovered after the 12th day in Group A, which is the time point when the lytic rate begins to dominate cell replication. It support the deduction that atmospheric $CO_2$ is dissolved in water and changes the acid-base balance of the system. Hence, brucite can dissolve with acidification during the culture of cyanobacteria:*

Introduction: L29: sentence is awkwardly written. It should read "Dissolved inorganic carbon in the typical..." (dissolved CO2 is not HCO3-)

*Thanks for your correction. It has been rephrased.*

Page 2: formation and dissolution of carbonate is one of the most instead of the most. I guess photosynthesis is at least as important

*Thanks for your correction. It has been rephrased.*

L4: Needs rewritting "seawater is considered supersaturated with several calcium carbonate phases such as xx (calcite?), with saturation index..." a solution is supersaturated with. A carbonate is not

*It has been rephrased.*
*"Modern sea water is considered supersaturated with several calcium carbonate phases, such as calcite and aragonite."*

L13 : you do not need to get into that debate about whether cyanobacteria formed the first stromatolites or not. Many studies now argue against this idea and I agree this is beyond the scope of your paper to debate about that. You should rephrase

*It has been deleted.*

L18: Instead of furthermore, I would write "In contrast", since it has been suggested that intracellular precipitation might be controlled in opposition to non-control as mentioned on line 16. And I would remind that this is true for some species of cyanobacteria, not all. Last, it was recently showed that it can occurs in undersaturated solutions (Cam et al., 2018 Geobiology). I am wondering if the same could be imagined in some environments with viruses lysing cyanobacteria. Maybe as a perspective?

*We agree with the suggestion. We highly appreciate the latest reference shared by the reviewer. Cam et al., (2018) is included as a reference for the introduction of intracellular calcification. We can design some viral calcification experiments in undersaturated medium in the near future to discuss the viral influence in intracellular precipitation.*

L30: you should specify that this increase would be very local

*We agree with the suggestion. It has been rephrased. The theoretically saturation index change is at the cellular level.*

Page 3, L19: Synechococcus spp.This is such as broad name encompassing so different bacteria. Could you specify at this point the name of the strain? Or specify that the strain was isolated in the present study?

*Cyanobacteria are firstly identified to the group of Synechococcus spp. based on the morphology of cell. DNA work will be carried out to precise the statement in the revised manuscript version. 16S rRNA genes of the strain were amplified by PCR and shows that the strain have the close relationship with Synechococcus sp. PCC 7177.*

P4: Is the strain axenic?

*During all the treatment, operations are asepsis strictly.*

L7: I guess the other not-mentioned treatment is a control where no virus has been added? Please specify this

*Yes. The other treatment is no inoculated with virus.*
*We re-write completely the "Experimental setup" paragraph to make our statements more accurate.*

Are culture bottles closed to air exchange or are they open? This is important to understand the evolution of DIC in your system I guess

*They are air exchangeable. So the DIC increase after the viral lysis of cyanobacteria and inhibition on photosynthesis.*

L8-9: please rephrase. I guess you do not measure OD on a fixed and filtered suspension.

*We did not measure OD but enumeration the abundance of cyanobacteria by autofluorescence.*

And Why do you fix the cells before measuring the chemical composition of the solution. Could fixation modify the chemical composition of the solution?

*Although we collect the DIC and total alkalinity subsample in bottles without headspace, carbonate chemistry parameters of solution may change during the storage. So, the subsamples were poisoned with $HgCl_2$. The same methods had been used for seawater sampling and analyses elsewhere (Cao and Dai 2011)*

L11: what is a TA sample?

*It is total alkalinity.*

2.3: Electron microscopy instead of electronic microscope

*It has been correct.*

L19: what is the stable phase? Do you mean stationary phase (after exponential)? But what do you mean by the end of it?

*Yes. Stationary phase is used more common than stable phase in literature. The precipitation was harvested at the 16$^{th}$ day, which means at the end of the stable phase.*

L26 what is abs?

*It refers to "selected-area electron diffraction"*

L27: Technically you do not sputter coat with carbon. This is achieved by evaporation. You can rewrite as: before being carbon coated

*The change is done accordingly*
*"For SEM analysis, dried precipitates were fixed onto aluminum stubs with two-way adherent abs and allowed to dry overnight. The samples were carbon-coated and examined with an Apreo scanning electron microscope (Thermofisher Scientific)."*

Page 5: L 3-4: the lag phase lasted 4-5 days

*The sentence has been rephrased.*
*"After the inoculation, cells exhibited a lag phase and started to grow exponentially for 9 to 13 d before reaching maximum cell numbers."*

L5: you say slightly lower but I see on Fig 4a a DO of 3 vs 14. This sounds like a very big difference to me

*The sentence has been rephrased.*
*"The cell abundance of Group A was slightly lower than Group B on the fifth day to eighth day"*

Fig. 4a: you said you ran duplicates. Where are the error bars then in your curves?. Could you show the pH curve?

*Error bars were added in panel Fig 4 (a, b). DIC samples were measured in one set in other*

*lab, so we cannot show the error bars of DIC but the accuracy of measurements of measuring are given. pH values are calculated from the geochemical program and can showed in revised manuscript.*

Why TA does not match with alkalinity? Which other species contribute to alkalinity here? Do you think that they als might vary differently? I think of N species? Or P species? TA sharply decreases at day 8 while alkalinity starts decreasing at day 6. How do you explain that?

*I think the reviewer want to know the reason why TA does not match with DIC during the day 6-8. During growth of cyanobacteria, the DIC drawdown by photosynthetic carbon uptake exceeds the slow reequilibration with the atmosphere. Thus, chemical speciations of the DIC pool shift toward higher $CO_3^{2-}$ concentration and pH. TA will not decrease until the precipitation of Ca and Mg, which is responsible for the base remove from the medium. That explain the DIC decrease first (day 6) and TA decrease later (day 8) by the precipitation of $CaCO_3$ and $Mg(OH)_2$.*

Why is there such a sharp decrease of Ca in all conditions at day 8?

*Despite the inoculation of cyanophage, culture with and without cyanophage grow at similar rates and reach the similar cell density during the first 8 days. The DIC drawdown by photosynthetic carbon uptake exceeds the slow reequilibration with the atmosphere, causing chemical speciation of the DIC pool to shift toward higher $CO_3^{2-}$ concentration and pH. At day 8-10, the geochemical condition influenced by photosynthesis promote the Ca and Mg precipitation.*

Why does dissolved Mg increase again in viral treatment after 14 days? L21: you do not mention that Mg also redissolve in the viral treatment.

*There is a strong positive correlation between $Mg^{2+}$ and DIC recovered after the 12th day, which is the time point when the lytic rate begins to dominate cell replication. In present open system, atmospheric $CO_2$ is dissolved in water and changes acid-base balance of the system. Hence, unstable mineral phases (like bructie) can dissolve with acidification during the culture of cyanobacteria. The dissolution of brucite in seawater is also discussed elsewhere, which is responsible for the mineral not well preserved over long time (Nothdurft et al., 2005).*

Fig 6, caption: by definition ACC is not crystalline so you should not write about crystallization of ACC but precipitation or formation

*The change is done accordingly*

Fig 7: please show a spectrum. Maps are not enough. Why is these are phosphates? Only spectra could show that there is no P peak

*EDS spectra of the nanoparticle shows that a small peak of element P. But P signal from the STEM mapping is not consistence with the nanoparticles. So it is not included in the early version of the manuscript.*

Fig 9: what is the peak at 10_? If XRD sees aragonite (ie crystalline phase) how does it come that you see amorphous Ca-carbonates? Is aragonite amorphizing under the TEM beam?

*Based on XRD results, the peak at 10 is loeweite, which is a sodium-magnesium sulphate hydrate [Na12Mg7(SO4)13 15H2O], found only in salt deposits of oceanic origin (Fang and Robinson 1970). Aragonite and ACC can be shown in a figure panel by SEM. Combine with the SEM and TEM, the formation of ACC during the cultivation can be clarified.*

Discussion: P25: the authors claim that this is no surprise that brucite forms but to my knowledge this has never been really shown by previous studies on cyanobacteria cultures; How do they explain that they produced brucite and not the other groups.

*It has been rephrased. We agree with the reviewer that few studies show the precipitation of brucite from cyanobacteria culture.*
*Literature review indicates that brucite precipitation may result from increased pH coupled with increased $Mg^{2+}$ activity and low $pCO_2$ (Nothdurft et al., 2006). When the cyanobacteria grow during the log phase, DIC are removed from the solution and the pH of the growth medium increased (Table 2). The chemical change leads to the supersaturate state of brucite (Table 2).*
*Thus, Brucite can formation in our culture.*

Moreover, they detect brucite by XRD but they mention that the solution became undersaturated with brucite after 8 days. How is it possible. I would expect in the worst case a SI of 0. Not below. I see that Mg is released after 8 days even in the viral treatment. This could be consistent with the undersaturation of the solution with brucite. But how do we switch from supersaturated to undersaturated? Precipitation of brucite should take it to saturated. And since there is no Mg going to aragonite and there is no carbonate in brucite, this cannot be explained by aragonite precipitation and changes in DIC.

*The story of Mg and brucite have been further discussed in revised manuscript according the advice from the reviewer.*
*First, not all brucite in cyanobacteria culture are dissolved after 8 days, which can be seen from the concentration of Mg. Thus, Brucite is detectable after 8 days.*
*Saturation indices (SI), which are determined using the software PHREEQC, yielded values > 0 for brucite during the first 8 days (0.34 ~ 1.15) and values < 0 after the 10th day (-1.46). The culture bottle are open to the air. So when DIC uptake by photosynthesis is slow compared to the log phrase, atmospheric $CO_2$ is dissolved in water and changes the $PCO_2$ level and acid-base balance of the system. Hence, brucite can dissolve with acidification during the culture of cyanobacteria. It is suggested that the brucite microbialite of Holocene corals may not be preserved over longer time, which is easy to dissolved in undersaturated solution.*

End of discussion is too long. The paragraph on page 9 from L 6 to 17 could be skipped or at least significantly reduced since this is quite far away from the main scope of the paper

*We appreciate the suggestion of this part of discussion. In revised manuscript version, they were reduced to a single sentence.*

**Response to the comments by referee#2**

The manuscript by Xu, et al. describes the isolation of a cyanobacterium and cyanophage and the use of this host/bacteriophage system to induce mineral precipitation in artificial medium under laboratory conditions. Using observational data, mineral characterization technologies, microscopic cyanobacterial counts and chemical analyses the authors conclude the presence of cyanophage in a culture of the host cyanobacterium lyses the host and releases cellular constituents into the culture medium. The author propose this release of dissolved and particulate cellular constituents promotes the precipitation of specific polymorphs of calcium carbonate and magnesium hydroxide.

General Comments:

1. The manuscript is not well organized, with some sections lacking adequate methodological information. Collectively, these issues makes the manuscript a bit difficult to read and interpret. Examples will be specifically described in the Specific Comments section.

*The methodological have been re-organized following the reviewer's suggestion.*

2. The use of "calcium carbonate" is used throughout the manuscript but in most instances, this is too general a descriptor within the context of this study. The use of the specific calcium carbonate polymorph names [e.g., amorphous calcium carbonate (ACC), vaterite, aragonite, calcite] will be more appropriate when applicable.

*Thanks for the comments. The specific calcium polymorph names are used at the right place in revised manuscript.*

3. The mineral precipitation mechanisms of homogenous and heterogeneous nucleation are not clearly delineated throughout the manuscript and appear to be used interchangeably in some instances. Additionally, mineral nucleation and precipitation are also used interchangeably. This conflation of terms, phrases and concepts makes it more difficult for the reader to read and interpret the manuscript. The experimental design for this study does not allow the detection or characterization of nucleation events, only gross precipitation that can only be indirectly assumed the result of one or both mechanisms of nucleation.

*We highly appreciate the term names clarification and understand that is preferable avoid any confusing terminology in the carbonate nucleation and precipitation. Nucleation of calcium carbonate mineral in the manuscript are deleted from the text.*

4. Throughout the manuscript total alkalinity (TA) and alkalinity are used interchangeably. When working with marine carbonate chemistry there is a significant difference between the ways these two alkalinities are calculated. Make it clear to the reader which one is being used or referred to.

*There are three alkalinities (total alkalinity, cytoplasmic alkalinity and carbonate alkalinity) used in the manuscript. We agree with the reviewer that we should make them clear to the reader.*
*Total alkalinity used in the manuscript is the measurement of water's ability to neutralize acids.*
*Intracellular alkalinity used in the early version of manuscript refer to the alkalinity released from the cell by lysis of bacteria. We revised it to the "Cytoplasmic bicarbonate" following Lisle and Robbins (2016).*
*Carbonate alkalinity calculates the amount of negatively charged carbonate and bicarbonate atoms in the solution. We talk about the carbonate alkalinity once in the text when we discuss the dissolution of ACC (P7 L28).*
*P6, L7 has been rephrased.*

5. There is repeated mention of carbonate chemistry in the text and tables. However, there is no listing of the geochemical data used for or the output from the geochemical modeling analyses. These data sets need to be included.

*The geochemical data are provided in a supplemental table.*

6. Saturation indices (SI) are a central component of this manuscript but there is no description of how these were calculated. When working with carbonate chemistry in marine waters most calculations on saturation states are per polymorph, like aragonite and calcite. These calculations are not normally performed within commonly used geochemical modeling programs but rely on CO2SYS or a recently developed application, CO2calc. The calculated SI values from the geochemical modeling software and CO2SYS/CO2calc are not always equivalent.

*Carbonate chemistry was calculated by the program PhreeqC [version 3.3; Wateq4f database; United States Geological Survey (USGS), Reston, VA, USA], which has been used in previous cyanobacterial calcification research (e.g. Obst et al., 2009). Compared with CO2calc and CO2SYS, PhreeqC is more convenient to calculate magnesium-related mineral. We agree with reviewer that the saturation states are per polymorph and the calculated SI values from different geochemical modeling software are not always equivalent. Under this circumstances, we give the geochemical modeling analyses with CO2calc and Phreeqc separately as supplemental materials.*

7. The Discussion section is too long and not focused on placing the data and interpretations from this study in context of previously published papers. There is a considerable amount of text dedicated to introducing and developing concepts that are on the periphery of the stated objectives and generated data of this study. This section needs to be edited to remove these passages and re-written to focus the discussion in a focused and concise style.

*We understand the concern pointed out by the reviewer regarding to the discussion section. We take the utmost care to refine the part of discussion. In revised manuscript version, some*

*part of discussion are reduced to a single sentence to expand the importance of viral induced calcification.*

Specific Comments:
Abstract
Pg 1: Ln 10-22. (1) The data do not support the statement that the presence of viruses stabilizes the carbonate minerals detected in this study.

*The precipitate investigated by XRD showing particles of aragonite in viral lysate and brucite in bacterial culture. XRD results combined with chemical parameters change of the non-infected culture revealing that it is unable to calcify to the extent that a stable $CaCO_3$ precipitate was formed. As atmospheric $CO_2$ dissolved in water, acid-base balance of the system will be changed. Unstable mineral phases can dissolve with acidification. However, with the aid of the viral cycle and the lysis of the host, the dissolution of carbonate seemed not to happen in viral treatment, and a more stable mineral formed. Thus, we conclude that viral lysis of cyanobacteria stabilizes the carbonate minerals detected in this study*

  (2) There's no evidence in this study of rapid intracellular calcification due to intracellular calcium concentrations.

*We decided to tone down our statement and reformulate the discussion regarding intracellular calcification.*

Introduction
Pg 2: Ln 1-2. The precipitation and dissolution of calcium carbonate does later sea water chemistry but do these processes have a more significant influence on the carbonate chemistry than carbon dioxide flux from the atmosphere?

*It has been rephrased.*
*"Formation and dissolution of calcium carbonate is one of the most important process that can change the carbonate chemistry in sea water"*

Pg 2: Ln 11-15. This passage describes sedimentary processes and stromatolites. Consideration should be given to removing this passage from the manuscript. This concept is not relevant to the stated theme of this manuscript.

*Thanks for your advice. They are removed.*

Pg 2: Ln 16. The precipitation of calcium carbonate, regardless of the mechanism, is always a controlled geochemical or biogeochemical process.

*$CaCO_3$ biomineralization by cyanobacteria are considered as exclusively extracellular. The sheath structure of some species of cyanobacteria may play a role in the calcification process, but environmental influence are also crucial: the saturation state of the adjacent*

*water, which affects precipitation of calcium carbonate minerals, and the availability of dissolved $CO_2$, which affects photosynthesis (reviewed by Riding 2012/Science, VOL 336). Although it is the truth that calcification by coccoliths or bivalves are controlled mineralization and are significant in marine chemistry, cyanobacterial calcification processes introduced in our manuscript are non-controlled.*

Pg 2: Ln 23-29. All of the information in this passage is true. However, it's not clear how this information is relevant to the stated objectives and experimental design of this study. Considering should be given to removing this passage from the manuscript.

*Present manuscript focus on the important role of viruses in precipitation of carbonate minerals. This passage reviews the direct effects of viral lysis of microbe in marine system and illustrates why we turn to virus to discuss potential viral effects on CaCO3 biomineralization. Thus, it is important to the experimental design. We value the comments by the reviewer. We reorganize the passage and make it more reasonable and readable.*

Pg 2: Ln 33. Which of the calcium carbonate polymorphs was capable of homogenous nucleation in the cited study?

*The thermodynamics calculation proposed by Lisle and Robbins (2016) reveal that the activation energy for nuclei formation thresholds for all three polymorphs is significantly reduced but only vaterite nucleation is energetically favored. It has been described precisely in revised version.*

Pg 3: Ln 1-2. The authors state the cited study does not consider the role of magnesium in their calculations. However, they don't tell the reader why it's important in this study. Since the role of magnesium in mineral precipitation is one of the objectives, this would be the place to provide the reader with some background information that will put the data and interpretations in the proper context.

*We appreciate the suggestion from the reviewer. Mg story is included in introduction in the revised version.*
 *"Displacements of acid-base carbonic equilibrium in seawater can not only form calcium carbonate minerals, but also can lead to the precipitation of Mg(OH)2 (brucite) (Möller, 2007). It has been proposed that the dissolution of brucite in seawater is favourable for CaCO3 precipitation (Nguyen Dang et al., 2017)."*

Pg 3: Ln 5-14. The information in this passage is not relevant to this study based on the stated objectives and experimental design. Its inclusion is a distraction from the Consideration should be given to removing this passage from the manuscript

*This passage reviews the research on viral particle acting as nucleation sites for different mineral precipitation. We refine the passage to make it more concise.*

Pg 3: Ln 17-18. (1) The authors state the understanding of viral influences on the precipitation of carbonate is poorly understood but they in the previous 16 lines of text they list several published studies that do characterize this process.

*Thanks for your reminding. It has been reparsed.*
*"Virus-related carbonate minerals are also reported in recent studies of biofilms from hypersaline lakes, where hypersaline carbonate minerals can precipitate at the surface of viral particles (Pacton et al., 2014; Lisle and Robbins, 2016; Perri et al., 2017). However, the pathway of precipitation of calcium carbonate onto the surface of viruses remains poorly understood."*

(2) Here is an example of the confusing use of nucleation and precipitation in the same sentence.

*The change is done accordingly.*
*"When combined with the release of cytoplasm-associated bicarbonate, which results in the formation of carbonate mineral energetically favored, and available viral capsids for surface-induced precipitation, the comprehension of viral influence on the precipitation of carbonate is extremely limited. "*

Pg 3: Ln 30. (1) A virus infected cyanobacterium is not a cyanophage. (2) Interpreting Figure 2 as showing a cyanophage and its host bacterium is a bit of a reach. These images could be almost anything.

*"Cyanophages, which infect this ecologically important group of cyanobacteria were isolate from the surface seawater from Sanya Bay." TEM image of cyanophage by negative staining is included in the revised version of manuscript.*

Pg 3: Ln 31-32. There needs to be, at a minimum, an abbreviated description of the cited method used to isolate, purify and identify the cyanobacterial specie. It should not be incumbent on the reader to run the most basic description of a method.
Pg 4: Ln 1-2. (1) As noted in the previous comment, at a minimum, an abbreviated description of the cited method for the cyanophage isolation needs to be provided. (2) There is a reference to metagenomics analysis, including a supplemental data file. The method of sample collection, processing, sequencing and sequence analysis (including the bioinformatics) is not mentioned or described anywhere in this manuscript. This is a significant deficiency in this version of the manuscript.

*Thanks for your suggestion.*
*Although isolation and identification of cyanobacteria and cyanophage are out of scope of the main goal of the manuscript, they are used for simulation experiment. Detailed methods for isolation and identification are provided as supplemental file in revised version.*

Pg 4: Ln 3-17. (1) This section is very poorly organized and developed with respect to the

different methods mentioned. There is no reasonable way a person could replicate this research effort or interpret the data from these methods using this section for guidance. (2) In this reviewer's opinion there are seven distinct methods: culture growth conditions; cyanobacteria counts; ion chromatography; salinity, total alkalinity and DIC measurements; geochemical modeling of carbonate chemistry; metagenomics analysis. Consider giving each of these their own sub-section within Experimental Setup and develop each section so the reader will understand how the methods were performed. (3) List the incubation or experimental times for each experiment type. (4) What is meant by a "pre-culture"? (5) A "one treatment" is mentioned. Are there other treatments? If so, describe those treatments and their respective differences. (6) List the volumes for each experimental container, the volumes of the sub-samples and the times at which the sub-samples were collected. (7) For the cyanobacterial growth cultures provide the light wavelength and dose.

*We agree with the reviewer that each of the method should be given a sub-section in revised version of manuscript. Changes are done accordingly.*
*"2.1 Cyanobacteria and Viruses*
*2.2 Culture conditions and calcification experiments*
*2.3 Measuring methods*
*2.3.1 Total alkalinity and dissolved inorganic carbon*
*2.3.2 Calcium and magnesium cations*
*2.3.3 Enumeration of cells and viruses*
*2.3.4 Electronic microscopy*
*2.3.5 X-ray Diffraction*
*2.4 Saturation indices calculation"*
*"Supplement*
    *Synechococcus sp. PCC 7177*
    *Viruses that infect Synechococcus sp. PCC 7177*
    *Saturation indices calculation"*

Pg 4: Ln 15-17. There is reference to geochemical modeling using a specific program. However, there is no mention or listing of the data on which the geochemical analyses were performed or the outputs from those analyses (e.g., activities of carbonate species). Both of these data sets need to be included in this manuscript. Additionally, the PHREEQC code used for these analyses needs to be included, most likely as part of the Supplemental Data files.

*Data used for geochemical calculating are provided in revised version. The PHREEQC code used in the early version of the manuscript, as referred, is Wateq4f database. In order to make it more directly, we list it in the supplemental data files together with data calculated by CO2calc.*

Pg 4: Ln 19. Here and in several later passages, there are references to "phases" of the cyanobacterial cultures. Based on the references it's assumed these phases are similar to those commonly measured during the growth of bacteria in the laboratory (i.e., lag, exponential, stationary). However, the methods used to determine these growth phases and the data from

those experiments are not provided. This is another method that should be included in the Experimental Setup section.

*The change is done accordingly. They growth curve is determined by the counting of autofluorescence during the culture of cyanobacteria.*

Results
Pg 5: Ln 4-5. How many cyanophage were inoculated into the cyanobacterium culture? Were the cyanophage titered? If so, the titer data need to be included. Without these data you cannot know the ratio of host cells-to-cyanophage (i.e., MOI), which has a significant influence on infection rates.

*Cyanophage were enumerated from the culture by epifluorescence microscopy with SYBR Green I staining (Patel et al., 2007).*

Pg 5: Ln 5-7. (1) Based on the brief description of the counting method and Figure 1b, it appears the cyanobacterial host abundances are based solely on the autofluorescence of the photosynthesizing microorganism. How can you be sure that nonfluorescent bacteria are not present in these cultures? Without knowing this, the possibility that some or all of the observed responses are due to a non-host bacterium or bacteria cannot be ruled out.

*Cyanobacteria have been isolated and purified using standard microbiological techniques. During all the treatment, operations are asepsis strictly.*
*Even if there is non-host and non-fluorescent bacterium, the culture media inoculation of the cyanophage will be turbid, which is caused by the growth of mixed bacteria. But Fig.3 b and d show the lysate are clear (Fig 3b and d). Even though it is not stained, non-fluorescent bacterium, if there is, may be seen from the optical microscope images. But this is conflicts with the optical microscope result. Under these conditions, we believe they are pure culture.*

(2) From this passage, it appears the cyanobacterial host abundances in the two culture types were measured using fluorescent microscopy but the cyanophage abundances were not determined in the co-culture. How can you conclude the cyanophage were responsible for any of the observations if there is no indication of how many were added to the host culture at time zero and their abundances at the different time points are not known? These cyanophage abundance counts from the different sub-samples should show increases as the host abundances decrease. Without the cyanophage abundance data you can't support the reductions in host abundances as being solely due to cyanophage induced lysis

*In the earlier version of manuscript, we made the conclusion that cyanophage were responsible for the host mortality after comparing the growth curve of culture with and without inoculation of cyanophage. But even if it is, we agree with the reviewer that the abundance of virus particles are needed.*

Pg 5: Ln 13-23. (1) This section needs to be revised into a more organized and concise presentation of the geochemical data. It's very difficult to interpret in its current format.

*Most of this section has been rewritten in revised version of manuscript.*

(2) Throughout this section there are repeated references to different subsamples and geochemical data associated with those sub-samples. This suggests sub-samples were collected at specific time points during the incubations and those sub-samples were then analyzed for the presence and concentrations of analytes required for the geochemical modeling of changes in the carbonate chemistry. The times of the sub-samples are not provided, the analytical data from those sub-samples is not presented and the outputs from the geochemical modeling analyses are not listed. Collectively, this information and data have to be included for the proper interpretation of the documented observations. For example, reference is made to seemingly unrealistic changes in total alkalinity, DIC and calcium and magnesium removal over the different growth phases of the host culture. The geochemical modeling data need to be presented to support these observations.

*We understand the concern pointed out by the reviewer regarding to the data acquisition and data view. We have included in the supplemental material a table of raw data and a comparison between different geochemical model outputs.*

(3) Was pCO2 measured during these experiments? If so, this method needs to be included and described.

*We do not measure the $pCO_2$ during the experiments. Alternatively, we calculate $pCO_2$ with total alkalinity and DIC with the known temperature, salinity and pressure. This is included in the supplemental material in revised version.*

Discussion
Pg 6: Ln 9. Its not clear what this balanced equation is representing? Its not at all clear from the text how formaldehyde (CH2O) is formed from bicarbonate and water or what the significance of CH2O is to the study described by this manuscript. Was the intention to let CH2O be a general reference to a carbohydrate? Or the dissociation of bicarbonate to carbon dioxide?

*It refers to the photosynthetic bicarbonate uptake uptake and its conversion within the cell to carbohydrate.*

Pg 6: Ln 14-17, Ln 21-26; Pg 7: Ln 9-10, Ln 16-17, Ln 19-21, Ln 25-29. All of these passages require the reader to have access to the geochemical modeling input data and output results for each sample before an independent interpretation and evaluation of the observations can be made.

*Data used for geochemical calculating are provided in a supplemental data files in revised*

*version.*

Pg 6: Ln 1 and Ln 32. These are the first time brucite is mentioned, other than the abstract. As this mineral seems to a significant product of the processes described in this manuscript, consideration should be given to bring this mineral and its relative importance into the manuscript in the Introduction and the Materials and Methods sections, including the geochemical modeling sub-section.

*The change has been done accordingly.*

Pg 6: Ln 22-23. Here and within other passages later in the manuscript, there is reference to the saturation index (SI) as being the metric for determining the saturation, supersaturation and under-saturation state of the culture medium. Though true, the SI alone will not tell you if a mineral will precipitate or dissolve. For example, if this were unconditionally true then average seawater, which has an SI between 2-3, one or more of the calcium carbonate polymorphs would be precipitating out of solution all the time. Instead, average seawater is metastable with no precipitation, indicating there are more geochemical factors other than SI that dictate if a precipitation even will proceed or not. Here is another example of where the geochemical modeling data would assist in the discussion of the results from this study.

*We agree with the reviewer that more information and comments about the geochemical factors can be helpful. Data used for geochemical calculating are provided in a supplemental data files in revised version. The precipitation and dissolution of calcium and magnesium is mainly derived from the ion concentration in the medium. When we know the ion concentration change, we calculate the saturated state of the medium and show the calculated results supporting the precipitation or dissolution.*

Pg 8: Ln 1-16 and Ln 24-31; Pg 9: Ln 1-17. These passages do not add anything supportive to the manuscript, based on the stated objectives, experimental design and presented data. Consider deleting these from the manuscript.

*We decided to tone down our statement and reformulate the discussion regarding intracellular calcification.*
*The part 4.3 in the early version of manuscript is reduced following the suggestion from the reviewer.*

Pg 8: Ln 20-23, Ln 28-33. The precipitation of one of the calcium carbonate polymorphs in tropical marine waters, or whitings, has been shown via peer reviewed publications to be driven by biological processes and not the simple physical re-suspension of established carbonate sediments.

*Thanks for reminding. The discussion about whiting is deleted following the reviewer's suggestion.*

Conclusion

Pg 9: Ln 19-20. Without the geochemical data and model outputs a detailed view of carbonate chemistry changes have not been provided. Also, without cyanophage abundance counts the cyanophage infection and lysis of the host cells cannot be definitively stated.

*These two data sets are included in the revised version of manuscript.*

References

Pg 10-13. There are 65 references listed. Based on there being issues with citations being incomplete, having punctuation errors, odd symbols inserted (which may have been a conversion issue) and inconsistency with the DOI information format, there are 36 references that need to be reviewed and corrected if needed. There were too many to list individually.

*Change was done accordingly*

Table 1. (1) This are several modified f/2 medium formulations, but the recipe listed in this table is not included in those published formulation. If there is a citation for this formulation of f/2 please include it. Also if the mineral and vitamin solutions were purchased then this needs to be noted as well. If these solutions were made in the laboratory, then there are several ingredients that are missing (e.g., EDTA). (2) The final pH (which should be between 7.8-8.2) of the media needs to be included as well.

*F/2 mediums reported in literature are prepared by stocks adding to filtered seawater. Only $NaNO_3$, $NaH_2PO_4$, $Na_2SiO_3$, trace mental stock and vitamin stock are defined. Here, the artificial seawater base modified from Harrison et al (2010).*
*We apologize for the mistaking of repeating $Na_2MoO_4$ $2H_2O$ in the table and missing $Na_2EDTA·2H_2O$.*
*The final pH of the media was adjust to 8.*

*Guillard R.R.L. and Ryther J.H. (1962) Studies of marine planktonic diatoms. I. Cyclotella nana Hustedt and Detonula confervacea Cleve. Can. J. Microbiol. 8: 229-239.*
*Guillard R.R.L. (1975) Culture of phytoplankton for feeding marine invertebrates. in "Culture of Marine Invertebrate Animals." (eds: Smith W.L. and Chanley M.H.) Plenum Press, New York, USA. pp 26-60.*
*Harrison P J, Waters R E, Taylor F J R. A BROAD SPECTRUM ARTIFICIAL SEA WATER MEDIUM FOR COASTAL AND OPEN OCEAN PHYTOPLANKTON. Journal of Phycology, 2010, 16(1):28-35.*

Table 2: (1) The pH values for all of the samples are relatively high compared to the initial pH of f/2. Were these measured or the products of the geochemical modeling? (2) The higher pH values are unrealistically high for open marine waters but, interestingly, the SI values are all less than those for the two dominant polymorphs (aragonite and calcite) in typical marine water. At these pH values, higher SI values, relative to typical marine water, would be predicted. This is another example of where having the

geochemical modeling data is critical for the proper interpretation and assessment of the research data.

*The pH values were calculated from the PhreeqC. Photosynthetic bicarbonate uptake and its conversion within the cell to CO2 by carbonic anhydrase lead to increased pH in the immediate vicinity of the cell (Reviewed by Ridding 2011).*
*In present study, when lytic rates run over the bacterial replication, photosynthesis ceases making the atmospheric CO2 dissolved in the water and changing the acid-base balance of the system. In present study, SI value care and $\Omega$ were determined following:*
*$SI = log\ \Omega = log[\ IAP/K_{sp\ Mineral}]$       wherein, $IAP = [Ca^{2+}][CO_3^{2-}]$*
*Thus, SI values in table 2 are higher than typical marine water.*
*We calculate the $\Omega$ of aragonite and calcite in CO2Calc in revised Supplemental table and the results also support our conclusion.*

Figures 1 and 2. These images do not contribute additional information to the reader. Consider deleting these from the manuscript.

*We agree with the reviewer and give the descriptions of cyanobacteria and cyanophage in a supplemental material in revised version of manuscript.*

Figure 5. Panel 5c is not cited in the manuscript.

*The citation is done accordingly.*

Figure 8. The simple shape of an object is not enough to proclaim it being a microbial cell, virus or encrusted microbial structure. Unless more definitive proof can be provided that will support the declaration that this images are what's written in the legend, consideration should be given to deleting this figure from the manuscript.

*We agree with the reviewer that morphological evidence is not enough to discriminate the microbial and mineral. We attempted to take EDS or EDS-mapping of the encrusted structure. Unfortunately, SEM under EDS model could not acquire the morphology of particle with hundreds nanometers. Since we have decided to tone down our statement regarding intracellular calcification, electron microscopic photographs are revised following the reviewers suggertion.*

Figure 10. This is a nice graphic but it's not explained well in the manuscript and there is a lot of information which is not even mentioned in the manuscript. This figure would be a nice summary graphic but the information in the image will have to be presented in the manuscript.

*Fig. 10 are revised following the suggestion of reviewer on the mechanism of viral induced calcification. We try our best to explained it well in revised version of manuscript.*

Supplementary Figure. The information in this figure stands alone because nothing about or within this figure is mentioned or described in the manuscript. In the manuscript's current format, this figure does not support anything presented in the Results or Discussion sections. For this reason, consideration should be given to removing this figure from the manuscript.

*We understand the concern pointed out by the reviewer regarding to the supplementary Figure. Since the method of isolation and characterization of cyanopahge are needed in the revised manuscript, we interpret this figure together with the characterization of cyanophage in supplementary material.*

[revised manuscript text omitted]

---

## Author Response (AR2)

*Dear editor:*

*You recently returned a review of our manuscript "Precipitation of Calcium Carbonate Mineral Induced by Viral Lysis of Cyanobacteria". After having read it, my coauthors and I appreciate the work of you and two anonymous reviewers. We are thankful for all the valuable comments and suggestions.*

*The revision was carried out according to the suggestions of the RC2. We feel it necessary to thank this reviewer for his (or her) most perceptive review. In the first point of RC2, the reviewer offered his doubt on the input data and constraints for the geochemical modeling using PHREEQC. In this response, we described the input to perform the geochemical calculations. Proper references that support our calculations were also cited to support our methods.*

*In the second point of RC2, the reviewer suspected the contribution of cyanophage in our system. We understand the reviewer's proposition because the growth curve was different from the traditional virus one-step growth curve. Whether viruses were responsible for the lysis of cyanobacteria was important for our discussion and conclusion. There were several evidences support our conclusion that cells were lysed by viruses:*
*(i) Different Growth curves of cyanobacteria in two treatments.*
*(ii) Estimation of burst size using two data achieved a considerable level.*
*(iii) The discussion that coprecipitation of viruses and calcium carbonate would result in underestimating the viral abundance.*

*In the third point of RC2, the reviewer concluded that the experimental study in present study could not be extended to natural seawater system. The system in our experiment (medium, Cyanobacteria, Cyanophage) was comparable to seawater environment. Similar experimental study were widely used in other published papers. Such modeling experiments generally do not provide an ultimate and direct answer as to which geobiological processes are involved. However, these experiments constrain, to some extent, the chemical conditions necessary to predict the geochemical processes similar to those in the aquatic environment. For this reason, we have changed our title of manuscript to "Precipitation of Calcium Carbonate Mineral Induced by Viral Lysis of Cyanobacteria: Evidence from Laboratory Experiments". The revised manuscript will focus on the interpretation of new pathway of $CaCO_3$ biomineralization that virus involved. The introduction and discussion of seawater carbonate will be deleted.*

*Below we have pasted in the entire review, and we have inserted our responses to the suggestions point-by-point (blue font). We hope that these changes lead to the acceptance of the manuscript, and look forward to hearing from you.*

*Sincerely,*

*Xiaotong Peng, on behalf of the co-authors.*

Editor: What does C+P stand for in the supplementary table 2 (which, by the way, lacks a caption)

*C+P represents the treatment inoculated with virus. The captions are added in revised version of supplementary table.*

RC1: I am satisfied with the author corrections. There are still few minor issues:

 "The ocean is recognized as a large carbon reservoir that contains approximately sixty times more carbon in the form of dissolved inorganic carbon than that in the pre-anthropogenic atmosphere": very awkwardly written: you mean that there is 60 times more inorganic C in the ocean than in the atmosphere. This is true for the modern of the pre anthropogenic Earth. But here one has the feeling that this number changed dramatically with anthropic effects which is not the case. Moreover, it is grammatically not correct. Please have it read by a native speaker

*Thanks for the suggestion. The sentence has been deleted.*

RC1: 6000 lux: waoh, this is huge. Is that conventional? Can you cite a paper using these conditions?

*The light condition is adopt from experimental cyanobacteria calcification conducted by Yang et al., (2016).*

RC1: L22: electron microscopy instead of electronic microscopy

*Thanks for the correction.*

RC1: "As DIC transportation by the growth of cyanobacteria, there was…": please rephrase since I do not understand what do you mean

*It has been rephrased.*
*"In Group B, there was a negative correlation between DIC and cell growth which was a reflection of photosynthetic carbon uptake (Fig. 2a, c)."*

RC2: After reviewing the authors' responses to the reviewers' comments on the original submission and the revised manuscript, the authors addressed some of the reviewers' comments and suggestions adequately but there are several remaining issues which must be addressed. Most of these issues will not be listed in this review because there are a few items on which the manuscript's hypotheses and data interpretations are based that cannot be supported. These major or core issues are the input data and constraints for the geochemical modeling using PHREEQC and the lack of data showing definitive lysis of the cyanobacterial cells by cyanophage. Based upon the statements in the Abstract and Discussion the reliability of geochemical data and active lysing of cyanobacterial cells by cyanophage are the two most critical aspects of this study and on which the authors base their conclusions. Comments on the issues associated with these concepts are listed below:

*We understand the concern pointed out by the reviewer. We agree with the reviewer that geochemical modeling and viral data are important for our conclusion. We are sorry that we don't interpret them well in the early version of manuscript. In the 2nd revised manuscript, input data for*

*the modeling are pasted and life cycles of viruses in our system are discussed based on our data.*

1. All experiments were conducted in a complex medium (i.e., f/2 medium). Therefore, all data from the geochemical modeling using PHREEQC and seawater carbonate chemistry using CO2calc are suspect when extending the data to natural seawater or freshwater. A reviewer's comment on the original submission requested the actual PHREEQC code and variable entries and respective concentrations (or activities) (i.e., the basis) be included in the revised manuscript. The code and basis entries were not included in the revised manuscript. This is critically important because the resulting data and interpretations are dependent upon what was entered into PHREEQC and the options the authors used to constrain the program's outputs. For example, the relevance and reliability of PHREEQC outputs are totally dependent upon how the components of f/2, as listed in Table 1, were entered into PHREEQC. The saturation indices (SI) for the calcium carbonate polymorphs listed in Table 2 and Table S1 are not relevant to natural freshwater or seawater, only to f/2 medium. With regard to using the CO2calc, the use of this program is not appropriate for this study because it is optimized for calculations of equilibrium carbonate chemistry variables in seawater and, to a lesser degree, freshwater.

*Thanks for the valuable comments from the reviewer. We agree with the reviewer that geochemical modelling with PHREEQC is very important for interpretations. In the response to the reviewer and revised version of manuscript, we provide the used database (Wateq4f) and all chemical data. We are sorry to get the basis entries in the software immediately visible. In the 2nd revised version of response, we prepare a detail appendix including initial data, database used, basis entries, results output, and data used in the manuscript.*

*Since the CO2Calc is not applicable to present complex medium. The use of CO2calc will be removed in the 2nd revised version of manuscript.*

*With regard to PHREEQC, it is designed by USGS to perform a wide variety of aqueous geochemical calculations, which has been used in previous carbonate modelling calculation, like experimental cyanobacteria calcification (e.g. Obst et al., 2009), biofilms in a $CO_2$-degassing karst-water creek (Shiraishi et al., 2008), and alkaline wetland (Power et al., 2007).*

*In the early version of manuscript, speciation of the carbonate system and saturation state of medium relative to a set of minerals were modelled for each of the subsample solution. The essential data needed for speciation calculation are the temperature, pH, and concentrations of elements and (or) element valence states. These data for subsamples were given in composition of F/2 medium, except for alkalinity, Ca, Mg, and DIC, which were determined of each subsample. The input files for calculation were shown in the appendix of this response. Because alkalinity was given, pH wouldl be adjusted to obtain desired alkalinity. The carbonate thermodynamic database in the model is based on the results of Plummer & Busenberg (1982).*

2. Cyanobacteria vs cyanophage data (Figure 2A): One of the recurring concepts in the original and revised manuscript is the role cyanophage play in changing the carbonate chemistry in a culture of cyanobacteria. By lysing the host cyanobacterial host cell, the infecting cyanophage not only release the intracellular contents of the host cell (e.g., HCO3-) but also relatively high numbers of progeny cyanophage. As this cycle of (1)

host cell infection, (2) cyanophage controlling host cell synthesis processes, (3) synthesis of infectious progeny cyanophage and (4) host cell lysis and release of more cyanophage progresses through a contained culture as described in this study, there should be a point in the cyanobacterial growth curve where the cell numbers significantly decrease with a concomitant increase in the number of cyanophage. The graphed data in Figure 2A does not show these trends in the respective counts. Therefore, the contribution of cyanophage lysis of host cyanobacterial cells to the changes in carbonate chemistry variables cannot be supported as Figure 2A shows no significant reduction in cyanobacterial counts (i.e., no significant viral lysis) over a 14 day period of incubation.

*The authors agree with reviewer that evidences support viral lysis of cyanobacteria are important. Traditionally, viruses have lytic cycle will break open host cell after immediate replication of the virion. As soon as the virus progeny released, they will find new hosts to infect. However, it has been reported that the cyanobacteria appeared to be less affected by viral infection in contrast to the results for heterotrophic bacteria (Ortmann et al., 2002). For example, the effect of viral infection on photosynthesis has been studied in the case of Synechococus sp. BBC1, where photosynthesis was not affected until near the onset of lysis (Suttle and Chen 1993). Moreover, the production rates of viruses infecting Synechococcus spp. was much low and it is discussed that most cyanobacteria were resistant to co-occurring cyanophage (Waterbury and Valois 1993). The research in Woods Hole Harbor conducted by Waterbury and Valois (1993) indicate that only 0.005% - 3.2% of the Synechococcus population was contacted and assumed to be infected. Thus, it was concluded that phages have a negligible effect in regulating in the densities of marine Synechoccus populations. Compared to the heterotrophic bacteria, there was less evidence that cyanophage had a large impact on gross cyanobacterial production (Ortmann et al., 2002). Thus, it is reasonable that the cell numbers are not decreased sharply in present study.*

*In the Group B, growth of cyanobacteria continued over the first 12 days of the incubations, suggesting that growth was not nutrient limited. After the inoculation of viruses, growth rate of the cyanobacteria was reduced. Decreased growth in the viral treatment was recognized as the result of induction of lysogens, resulting in cell lysis.*

*Determination of burst size is one of the important way to verify whether cyanobacteria are lysed by cyanobacteria. The burst size of virus can be estimated by calculating the ratio of viral particles to the number of killed host prokaryotes over the short-time intervals. The difference between observed and expected cell number is assumed to represent cells killed by viral lysis (Danovaro et al. 2008). Thus, the lysed cells were estimated either by decrease of bacteria number in group A or increase of bacteria number in group B. Two estimation came up with similar results. So, we can confirm that virus are responsible for the cyanobacterial mortality and the burst size is 3.01-3.29 (Table r1)*

*Table r1: Estimation of Burst size*

| Days | Virus ($10^7$/ml) | Bacteria (Group A, $10^7$/ml) | Bacteria (Group B, $10^7$/ml) | decrease of bacteria number in group A ($10^7$/ml) | increased of bacteria number in group B. ($10^7$/ml) |
|---|---|---|---|---|---|
| 8 | 1.45 | 3.34 | 3.55 | 0.82 | 0.75 |

| 9 | 3.92 | 2.52 | 4.30 | | |
|---|------|------|------|---|---|
| Burst size | | | | 3.01 | 3.29 |

*Burst size is the average number of viruses released when a single cell lyses. The results from Brigden et al., (2003) inferred that there were profound differences in cyanophage replication characteristics that were affected by host strain type and physiological status. Consequently, inferences on the effect of viruses on host dynamics in nature are fraught with uncertainty. Thus, it is reasonable with burst size of 3.01~3.29.*

*What is more, we propose that coprecipitation of viruses and calcium carbonate would result in underestimating the viral abundance in present study. The encrusted structure indicated by SEM images may support the hypothesis of carbonate formation on and near the virus particles (Fig. 4d). Subsamples for enumeration of virus particles were filtered to remove precipitated minerals. The yield of the filtrate may preclude viruses incorporated in minerals. This is a reasonable explanation as to why burst size is much low. Consequently, any loss of viruses due to the calcification would result in titer being underestimated.*

3. Table S1: For natural freshwater and seawater systems the values listed for pH, pCO2, CO3 and HCO3 are unrealistic when compared to data from the same variables from natural seawater. For example, the typical range for pCO2 should be between 140-770 micro-atm. The large deviations of these data sets from what is typically measured may be the result of improper geochemical modelling with PHREEQC. (See Comment #1). The authors have not described a laboratory system from which the data can be reliably extended to natural seawater or freshwater systems as they propose. The experimental design is more applicable to an industrial setting where cyanobacteria are used to scrub CO2 from product streams and/or precipitate carbonate-based minerals on the surfaces of the cells and/or viruses in those product streams or perhaps a solid phase reactor.

*First, sorry for the misuse of CO2Calc. In second revised version, we will delete the data obtained from the CO2Calc.*

*Carbonate parameters (pH, $pCO_2$, $CO_3^{2-}$, and $HCO_3^-$) are largely influenced by the growth of cyanobacteria. In restricted biotic experiments, these parameters are changeable due to the photosynthetic uptake of $HCO_3^-$ ions, $OH^-$ release and calcification. These changeable carbonate parameters were also reported in previous experimental cyanobacteria calcification (e.g. Brady et al., 2004; Mavromatis et al., 2012; Shirokova et al., 2013). There was a negative correlation between DIC and cell growth which was a reflection of photosynthetic carbon uptake. Photosynthetic carbon uptake raised the pH values of the medium and constructed a favorable calcification environment where carbonate was the dominant inorganic carbon species. So, it is reasonable that subsamples is distinguished from natural seawater.*

*However, it is important to make sure that the process established in the laboratory are comparable to the natural environment. The host cyanobacteria and viral isolated used in this study were isolated from the marine environment. Synechococcus is one of the most abundant photosynthetic cell in oligotrophic oceanic environments where it can be responsible for a significant amount of the total primary productivity. Cyanophage is the most frequently isolated*

*virus in the marine environment and Synechococcus spp. In laboratory studies on phage-host systems infection is commonly carried out under optimal conditions for host growth (modified F/2 medium in present study), but in natural environments bacteria are subject to an alternating feast and famine existence. It is important to note that the values for host density used for modelling studies tend to be much higher than those found for natural populations of marine environment. Such modeling experiments do not intend to mimic the processes occurring within the cells, which remain unknown, and generally do not provide an ultimate and direct answer as to which geobiological processes are involved in biomineralization. However, these experiments constrain, to some extent, the chemical conditions necessary to predict the geochemical processes similar to those in the aquatic environment. In any case, our experimental study give a detailed description of the virus-induced calcification.*

*For this reason, we have changed our title of manuscript to "Precipitation of Calcium Carbonate Mineral Induced by Viral Lysis of Cyanobacteria: Evidence from Laboratory Experiments", as well as introduction and discussion in the manuscript (like Fig. 7), to agree with the reviewer. In 2nd revision version of manuscript, we focus on the interpretation of new pathway of CaCO3 biomineralization that virus involved.*

**Reference:**

Brigden, S. M.: Dynamics of cyanophage replication, Master, University of British Columbia 2003.

Danovaro, R., Dell'Anno, A., Corinaldesi, C., Magagnini, M., Noble, R., Tamburini, C., and Weinbauer, M.: Major viral impact on the functioning of benthic deep-sea ecosystems, Nature, 454, 1084-1087, 2008.

Lee, B. D., Apel, W. A., and Walton, M. R.: Screening of Cyanobacterial Species for Calcification, Biotechnology Progress, 20, 1345-1351, http://dx.doi.org/10.1021/bp0343561, 2004.

Mavromatis, V., Pearce, C. R., Shirokova, L. S., Bundeleva, I. A., Pokrovsky, O. S., Benezeth, P., and Oelkers, E. H.: Magnesium isotope fractionation during hydrous magnesium carbonate precipitation with and without cyanobacteria, Geochimica et Cosmochimica Acta, 76, 161-174, http://dx.doi.org/10.1016/j.gca.2011.10.019, 2012.

Obst, M., Wehrli, B., and Dittrich, M.: CaCO3 nucleation by cyanobacteria: laboratory evidence for a passive, surface-induced mechanism, Geobiology, 7, 324-347, http://dx.doi.org/10.1111/j.1472-4669.2009.00200.x, 2009.

Ortmann, A. C., Lawrence, J. E., and Suttle, C. A.: Lysogeny and Lytic Viral Production during a Bloom of the Cyanobacterium Synechococcus spp, Microbial Ecology, 43, 225-231, http://dx.doi.org/10.1007/s00248-001-1058-9, 2002.

Plummer, L. N., and Busenberg, E.: The solubilities of calcite, aragonite and vaterite in CO2-H2O solutions between 0 and 90°C, and an evaluation of the aqueous model for the system CaCO3-CO2-H2O, Geochimica et Cosmochimica Acta, 46, 1011-1040, https://doi.org/10.1016/0016-7037(82)90056-4, 1982.

Power, I. M., Wilson, S. A., Thom, J. M., Dipple, G. M., and Southam, G.: Biologically induced mineralization of dypingite by cyanobacteria from an alkaline wetland near Atlin, British Columbia, Canada, Geochemical Transactions, 8, 13, http://dx.doi.org/10.1186/1467-4866-8-13, 2007.

Shiraishi, F., Reimer, A., Bissett, A., de Beer, D., and Arp, G.: Microbial effects on biofilm calcification, ambient water chemistry and stable isotope records in a highly supersaturated setting (Westerhöfer Bach, Germany), Palaeogeography, Palaeoclimatology, Palaeoecology, 262, 91-106, https://doi.org/10.1016/j.palaeo.2008.02.011, 2008.

Shirokova, L. S., Mavromatis, V., Bundeleva, I. A., Pokrovsky, O. S., Bénézeth, P., Gérard, E., Pearce, C. R., and Oelkers, E. H.: Using Mg Isotopes to Trace Cyanobacterially Mediated Magnesium Carbonate Precipitation in Alkaline Lakes, Aquatic Geochemistry, 19,

1-24, http://dx.doi.org/10.1007/s10498-012-9174-3, 2013.

Suttle, C. A., and Chan, A. M.: Marine cyanophages infecting oceanic and coastal strains of Synechococcus: abundance, morphology, cross-infectivity and growth characteristics, Marine Ecology Progress Series, 92, 99-109, 1993.

Waterbury, J. B., and Valois, F. W.: Resistance to co-occurring phages enables marine synechococcus communities to coexist with cyanophages abundant in seawater, Applied and environmental microbiology, 59, 3393-3399, 1993.

Yang, Z.-N., Li, X.-M., Umar, A., Fan, W.-H., and Wang, Y.: Insight into calcification of Synechocystis sp. enhanced by extracellular carbonic anhydrase, RSC Advances, 6, 29811-29817, http://dx.doi.org/10.1039/C5RA26159G, 2016.

**Appendix:**  Input file for speciation calculation in PHREEQC

| | Input file |
|---|---|
| Cyanobacteria 6 Day | Input file: C:\Users\XUHENGCHAO\Desktop\PHreeQC\culture\C1-6.pqi
Output file: C:\Users\XUHENGCHAO\Desktop\PHreeQC\culture\C1-6.pqo
Database file: C:\Program Files\USGS\Phreeqc Interactive 3.3.12-12704\database\wateq4f.dat
* * *
Reading data base.
* * *
    SOLUTION_MASTER_SPECIES
    SOLUTION_SPECIES
    PHASES
    EXCHANGE_MASTER_SPECIES
    EXCHANGE_SPECIES
    SURFACE_MASTER_SPECIES
    SURFACE_SPECIES
    RATES
    END
* * *
Reading input data for simulation 1.
* * *
    DATABASE C:\Program Files\USGS\Phreeqc Interactive 3.3.12-12704\database\wateq4f.dat
    SOLUTION 1
        temp      25
        pH        8
        pe        4
        redox     pe
        units     mmol/l
        density   1.02 |

Alkalinity 4.03

| | |
|---|---|
| B | 0.37 |
| Br | 0.72 |
| C | 1.96 |
| Ca | 8.45 |
| Cl | 483.51 |
| F | 0.07 |
| K | 8.04 |
| Mg | 49.79 |
| N(5) | 0.88 |
| Na | 417.34 |
| P | 0.04 |
| S(6) | 24.99 |
| Si | 0.11 |
| Sr | 0.08 |
| water | 1 # kg |
* * *
Beginning of initial solution calculations.
* * *
Initial solution 1.

pH will be adjusted to obtain desired alkalinity.

----------------------------Solution composition----------------------------

| Elements | Molality | Moles |
|---|---|---|
| Alkalinity | 4.077e-03 | 4.077e-03 |
| B | 3.743e-04 | 3.743e-04 |
| Br | 7.283e-04 | 7.283e-04 |
| C | 1.983e-03 | 1.983e-03 |
| Ca | 8.548e-03 | 8.548e-03 |
| Cl | 4.891e-01 | 4.891e-01 |
| F | 7.081e-05 | 7.081e-05 |
| K | 8.133e-03 | 8.133e-03 |
| Mg | 5.036e-02 | 5.036e-02 |
| N(5) | 8.902e-04 | 8.902e-04 |
| Na | 4.222e-01 | 4.222e-01 |
| P | 4.046e-05 | 4.046e-05 |

```
        S(6)              2.528e-02    2.528e-02
        Si               1.113e-04    1.113e-04
        Sr               8.092e-05    8.092e-05

--------------------------Description of solution--------------------------

                                    pH   =    9.551            Adjust
alkalinity
                                    pe   =    4.000
                        Activity of water   =    0.983
                Ionic strength (mol/kgw)   =    5.887e-01
                       Mass of water (kg)   =    1.000e+00
                     Total CO2 (mol/kg)   =    1.983e-03
                       Temperature (癈)   =    25.00
                  Electrical balance (eq)   =    2.822e-03
  Percent error, 100*(Cat-|An|)/(Cat+|An|)   =    0.27
                            Iterations   =    8
                            Total H   = 1.110145e+02
                            Total O   = 5.561796e+01

--------------------------Distribution of species--------------------------

                                                   Log              Log
Log        mole V
      Species              Molality    Activity   Molality  Activity      Gamma
cm?mol

      OH-                  5.645e-05   3.497e-05    -4.248    -4.456       -
0.208        (0)
      H+                   3.707e-10   2.814e-10    -9.431    -9.551       -
0.120         0.00
      H2O                  5.551e+01   9.831e-01     1.744    -0.007
0.000      18.07
B                3.743e-04
      H2BO3-               3.005e-04   1.729e-04    -3.522    -3.762       -
0.240        (0)
      H3BO3                7.382e-05   8.454e-05    -4.132    -4.073
0.059        (0)
      BF(OH)3-             1.360e-09   7.823e-10    -8.867    -9.107       -
0.240        (0)
      BF2(OH)2-            9.693e-16   5.577e-16   -15.014   -15.254       -
```

```
0.240        (0)
   BF3OH-                 7.072e-24    4.069e-24    -23.150    -23.391     -
0.240        (0)
   BF4-                   1.917e-31    1.103e-31    -30.717    -30.957     -
0.240        (0)
Br              7.283e-04
   Br-                    7.283e-04    5.382e-04     -3.138     -3.269     -
0.131        (0)
C(-4)           0.000e+00
   CH4                    0.000e+00    0.000e+00    -90.745    -90.686
0.059        (0)
C(4)            1.983e-03
   MgCO3                  5.565e-04    6.373e-04     -3.255     -3.196
0.059        (0)
   HCO3-                  4.695e-04    3.206e-04     -3.328     -3.494     -
0.166        (0)
   NaCO3-                 4.288e-04    2.927e-04     -3.368     -3.534     -
0.166        (0)
   CO3-2                  2.459e-04    5.342e-05     -3.609     -4.272     -
0.663        (0)
   CaCO3                  1.500e-04    1.718e-04     -3.824     -3.765
0.059        (0)
   MgHCO3+                7.345e-05    4.689e-05     -4.134     -4.329     -
0.195        (0)
   NaHCO3                 4.632e-05    5.305e-05     -4.334     -4.275
0.059        (0)
   CaHCO3+                1.123e-05    7.847e-06     -4.950     -5.105     -
0.156        (0)
   SrCO3                  6.192e-07    6.192e-07     -6.208     -6.208
0.000        (0)
   CO2                    1.802e-07    2.063e-07     -6.744     -6.685
0.059        (0)
   SrHCO3+                1.304e-07    8.904e-08     -6.885     -7.050     -
0.166        (0)
Ca              8.548e-03
   Ca+2                   7.560e-03    1.919e-03     -2.121     -2.717     -
0.595        (0)
   CaSO4                  8.198e-04    9.389e-04     -3.086     -3.027
0.059        (0)
   CaCO3                  1.500e-04    1.718e-04     -3.824     -3.765
0.059        (0)
```

| Species | Molality | Activity | Log Molality | Log Activity | Log Gamma | |
|---|---|---|---|---|---|---|
| CaHCO3+ | 1.123e-05 | 7.847e-06 | -4.950 | -5.105 | -0.156 | (0) |
| CaPO4- | 3.701e-06 | 2.526e-06 | -5.432 | -5.597 | -0.166 | (0) |
| CaOH+ | 1.593e-06 | 1.113e-06 | -5.798 | -5.954 | -0.156 | (0) |
| CaF+ | 5.668e-07 | 3.885e-07 | -6.247 | -6.411 | -0.164 | (0) |
| CaHPO4 | 2.624e-07 | 3.005e-07 | -6.581 | -6.522 | 0.059 | (0) |
| CaH2PO4+ | 9.310e-11 | 6.356e-11 | -10.031 | -10.197 | -0.166 | (0) |
| CaHSO4+ | 2.094e-12 | 1.548e-12 | -11.679 | -11.810 | -0.131 | (0) |
| Cl | 4.891e-01 | | | | | |
| Cl- | 4.891e-01 | 3.092e-01 | -0.311 | -0.510 | -0.199 | (0) |
| F | 7.081e-05 | | | | | |
| F- | 3.752e-05 | 2.324e-05 | -4.426 | -4.634 | -0.208 | (0) |
| MgF+ | 2.929e-05 | 1.920e-05 | -4.533 | -4.717 | -0.183 | (0) |
| NaF | 3.437e-06 | 3.936e-06 | -5.464 | -5.405 | 0.059 | (0) |
| CaF+ | 5.668e-07 | 3.885e-07 | -6.247 | -6.411 | -0.164 | (0) |
| BF(OH)3- | 1.360e-09 | 7.823e-10 | -8.867 | -9.107 | -0.240 | (0) |
| HF | 8.566e-12 | 9.810e-12 | -11.067 | -11.008 | 0.059 | (0) |
| HF2- | 1.412e-15 | 8.749e-16 | -14.850 | -15.058 | -0.208 | (0) |
| BF2(OH)2- | 9.693e-16 | 5.577e-16 | -15.014 | -15.254 | -0.240 | (0) |
| H2F2 | 2.190e-22 | 2.508e-22 | -21.660 | -21.601 | 0.059 | (0) |
| BF3OH- | 7.072e-24 | 4.069e-24 | -23.150 | -23.391 | -0.240 | (0) |
| BF4- | 1.917e-31 | 1.103e-31 | -30.717 | -30.957 | -0.240 | (0) |
| SiF6-2 | 5.185e-40 | 1.052e-40 | -39.285 | -39.978 | -0.693 | |

(0)
H(0)          9.791e-31
    H2                      4.896e-31    5.606e-31    -30.310    -30.251
0.059      (0)
K          8.133e-03
    K+                      8.005e-03    5.061e-03    -2.097    -2.296    -
0.199      (0)
    KSO4-                   1.277e-04    8.719e-05    -3.894    -4.060    -
0.166      (0)
    KHPO4-                  4.128e-09    2.818e-09    -8.384    -8.550    -
0.166      (0)
Mg          5.036e-02
    Mg+2                    4.317e-02    1.250e-02    -1.365    -1.903    -
0.538      (0)
    MgSO4                   6.275e-03    7.185e-03    -2.202    -2.144
0.059      (0)
    MgCO3                   5.565e-04    6.373e-04    -3.255    -3.196
0.059      (0)
    MgOH+                   2.231e-04    1.586e-04    -3.651    -3.800    -
0.148      (0)
    MgHCO3+                 7.345e-05    4.689e-05    -4.134    -4.329    -
0.195      (0)
    MgPO4-                  3.252e-05    2.220e-05    -4.488    -4.654    -
0.166      (0)
    MgF+                    2.929e-05    1.920e-05    -4.533    -4.717    -
0.183      (0)
    MgHPO4                  2.311e-06    2.647e-06    -5.636    -5.577
0.059      (0)
    MgH2PO4+                7.723e-10    5.273e-10    -9.112    -9.278    -
0.166      (0)
N(5)          8.902e-04
    NO3-                    8.902e-04    5.329e-04    -3.051    -3.273    -
0.223      (0)
Na          4.222e-01
    Na+                     4.164e-01    2.943e-01    -0.381    -0.531    -
0.151      (0)
    NaSO4-                  5.297e-03    3.616e-03    -2.276    -2.442    -
0.166      (0)
    NaCO3-                  4.288e-04    2.927e-04    -3.368    -3.534    -
0.166      (0)
    NaHCO3                  4.632e-05    5.305e-05    -4.334    -4.275

0.059      (0)
  NaF                    3.437e-06    3.936e-06    -5.464    -5.405
0.059      (0)
  NaHPO4-                2.400e-07    1.639e-07    -6.620    -6.785    -
0.166      (0)
O(0)          2.239e-32
  O2                     1.119e-32    1.282e-32    -31.951   -31.892
0.059      (0)
P          4.046e-05
  MgPO4-                 3.252e-05    2.220e-05    -4.488    -4.654    -
0.166      (0)
  CaPO4-                 3.701e-06    2.526e-06    -5.432    -5.597    -
0.166      (0)
  MgHPO4                 2.311e-06    2.647e-06    -5.636    -5.577
0.059      (0)
  HPO4-2                 1.407e-06    2.856e-07    -5.852    -6.544    -
0.693      (0)
  CaHPO4                 2.624e-07    3.005e-07    -6.581    -6.522
0.059      (0)
  NaHPO4-                2.400e-07    1.639e-07    -6.620    -6.785    -
0.166      (0)
  PO4-3                  1.654e-08    4.575e-10    -7.781    -9.340    -
1.558      (0)
  KHPO4-                 4.128e-09    2.818e-09    -8.384    -8.550    -
0.166      (0)
  H2PO4-                 1.896e-09    1.294e-09    -8.722    -8.888    -
0.166      (0)
  MgH2PO4+               7.723e-10    5.273e-10    -9.112    -9.278    -
0.166      (0)
  CaH2PO4+               9.310e-11    6.356e-11    -10.031   -10.197   -
0.166      (0)
S(6)          2.528e-02
  SO4-2                  1.275e-02    2.452e-03    -1.894    -2.611    -
0.716      (0)
  MgSO4                  6.275e-03    7.185e-03    -2.202    -2.144
0.059      (0)
  NaSO4-                 5.297e-03    3.616e-03    -2.276    -2.442    -
0.166      (0)
  CaSO4                  8.198e-04    9.389e-04    -3.086    -3.027
0.059      (0)
  KSO4-                  1.277e-04    8.719e-05    -3.894    -4.060    -

0.166      (0)
     SrSO4                  7.580e-06      8.680e-06        -5.120       -5.061
0.059      (0)
     HSO4-                 1.023e-10     6.708e-11       -9.990      -10.173        -
0.183      (0)
     CaHSO4+               2.094e-12     1.548e-12      -11.679     -11.810        -
0.131      (0)
Si            1.113e-04
     H4SiO4                5.734e-05      6.567e-05        -4.242       -4.183
0.059      (0)
     H3SiO4-               5.389e-05     3.441e-05       -4.268      -4.463        -
0.195      (0)
     H2SiO4-2              3.828e-08     8.317e-09       -7.417      -8.080        -
0.663      (0)
     SiF6-2               5.185e-40    1.052e-40     -39.285    -39.978     -0.693
(0)
Sr            8.092e-05
     Sr+2                 7.259e-05     1.816e-05       -4.139      -4.741        -
0.602      (0)
     SrSO4                  7.580e-06      8.680e-06        -5.120       -5.061
0.059      (0)
     SrCO3                  6.192e-07      6.192e-07        -6.208       -6.208
0.000      (0)
     SrHCO3+               1.304e-07     8.904e-08       -6.885      -7.050        -
0.166      (0)
     SrOH+                 4.847e-09     3.253e-09       -8.315      -8.488        -
0.173      (0)

----------------------------Saturation indices----------------------------

   Phase                    SI** log IAP      log K(298 K,     1 atm)

   Anhydrite          -0.97       -5.33      -4.36   CaSO4
   Aragonite           1.35       -6.99      -8.34   CaCO3
   Artinite            1.39       10.99       9.60    MgCO3:Mg(OH)2:3H2O
   Brucite             0.34       17.18      16.84    Mg(OH)2
   Calcite             1.49       -6.99      -8.48   CaCO3
   Celestite          -0.72       -7.35      -6.63   SrSO4
   CH4(g)            -87.83      -90.69      -2.86   CH4
   Chalcedony         -0.62       -4.17      -3.55   SiO2
   Chrysotile         11.02       43.22      32.20   Mg3Si2O5(OH)4

| Clinoenstatite | 1.68 | 13.02 | 11.34 | MgSiO3 |
| CO2(g) | -5.22 | -6.69 | -1.47 | CO2 |
| Cristobalite | -0.58 | -4.17 | -3.59 | SiO2 |
| Diopside | 5.34 | 25.23 | 19.89 | CaMgSi2O6 |
| Dolomite | 3.93 | -13.16 | -17.09 | CaMg(CO3)2 |
| Dolomite(d) | 3.38 | -13.16 | -16.54 | CaMg(CO3)2 |
| Epsomite | -2.43 | -4.57 | -2.14 | MgSO4:7H2O |
| FCO3Apatite | | 27.18 | -87.22 | -114.40 |
| Ca9.316Na0.36Mg0.144(PO4)4.8(CO3)1.2F2.48 | | | | |
| Fluorapatite | 8.40 | -9.20 | -17.60 | Ca5(PO4)3F |
| Fluorite | -1.38 | -11.98 | -10.60 | CaF2 |
| Forsterite | 1.91 | 30.21 | 28.31 | Mg2SiO4 |
| Gypsum | -0.76 | -5.34 | -4.58 | CaSO4:2H2O |
| H2(g) | -27.10 | -30.25 | -3.15 | H2 |
| H2O(g) | -1.52 | -0.01 | 1.51 | H2O |
| Halite | -2.62 | -1.04 | 1.58 | NaCl |
| Huntite | 4.45 | -25.52 | -29.97 | CaMg3(CO3)4 |
| Hydromagnesite | 1.21 | -7.55 | -8.76 | Mg5(CO3)4(OH)2:4H2O |
| Hydroxyapatite | 8.40 | 4.98 | -3.42 | Ca5(PO4)3OH |
| Magadiite | -5.89 | -20.19 | -14.30 | NaSi7O13(OH)3:3H2O |
| Magnesite | 1.85 | -6.18 | -8.03 | MgCO3 |
| Mirabilite | -2.63 | -3.75 | -1.11 | Na2SO4:10H2O |
| Nahcolite | -3.48 | -4.03 | -0.55 | NaHCO3 |
| Natron | -4.10 | -5.41 | -1.31 | Na2CO3:10H2O |
| Nesquehonite | -0.58 | -6.20 | -5.62 | MgCO3:3H2O |
| O2(g) | -29.00 | -31.89 | -2.89 | O2 |
| Portlandite | -6.43 | 16.37 | 22.80 | Ca(OH)2 |
| Quartz | -0.19 | -4.17 | -3.98 | SiO2 |
| Sepiolite | 6.09 | 21.85 | 15.76 | Mg2Si3O7.5OH:3H2O |
| Sepiolite(d) | 3.19 | 21.85 | 18.66 | Mg2Si3O7.5OH:3H2O |
| Silicagel | -1.15 | -4.17 | -3.02 | SiO2 |
| SiO2(a) | -1.46 | -4.17 | -2.71 | SiO2 |
| SrF2 | -5.47 | -14.01 | -8.54 | SrF2 |
| Strontianite | 0.26 | -9.01 | -9.27 | SrCO3 |
| Talc | 13.49 | 34.89 | 21.40 | Mg3Si4O10(OH)2 |
| Thenardite | -3.49 | -3.67 | -0.18 | Na2SO4 |
| Thermonatrite | -5.47 | -5.34 | 0.13 | Na2CO3:H2O |
| Tremolite | 28.78 | 85.36 | 56.57 | Ca2Mg5Si8O22(OH)2 |
| Trona | -8.58 | -9.37 | -0.80 | NaHCO3:Na2CO3:2H2O |

**For a gas, SI = log10(fugacity). Fugacity = pressure * phi / 1 atm.

| | |
|---|---|
| | For ideal gases, phi = 1.
* * *
End of simulation.
* * ** * *
Reading input data for simulation 2.
* * ** * *
End of Run after 0.156 Seconds.
----------------------------- |
| Cyanobacteria
8 Day | Input file: C:\Users\XUHENGCHAO\Desktop\PHreeQC\culture\C1-8.pqi
Output file: C:\Users\XUHENGCHAO\Desktop\PHreeQC\culture\C1-8.pqo
Database file: C:\Program Files\USGS\Phreeqc Interactive 3.3.12-12704\database\wateq4f.dat
* * *
Reading data base.
* * *
    SOLUTION_MASTER_SPECIES
    SOLUTION_SPECIES
    PHASES
    EXCHANGE_MASTER_SPECIES
    EXCHANGE_SPECIES
    SURFACE_MASTER_SPECIES
    SURFACE_SPECIES
    RATES
    END
* * *
Reading input data for simulation 1.
* * *
    DATABASE C:\Program Files\USGS\Phreeqc Interactive 3.3.12-12704\database\wateq4f.dat
    SOLUTION 1
        temp      25
        pH       8
        pe       4 |

```
        redox          pe
        units          mmol/l
        density        1.02
        Alkalinity 4.21
        B              0.37
        Br             0.72
        C              1.63
        Ca             8.58
        Cl             483.51
        F              0.07
        K              8.04
        Mg             50.56
        N(5)           0.88
        Na             417.34
        P              0.04
        S(6)           24.99
        Si             0.11
        Sr             0.08
        water          1 # kg
* * *
Beginning of initial solution calculations.
* * *
Initial solution 1.

pH will be adjusted to obtain desired alkalinity.

----------------------------Solution composition----------------------------

    Elements              Molality           Moles

    Alkalinity          4.259e-03    4.259e-03
    B                   3.743e-04    3.743e-04
    Br                  7.283e-04    7.283e-04
    C                   1.649e-03    1.649e-03
    Ca                  8.679e-03    8.679e-03
    Cl                  4.891e-01    4.891e-01
    F                   7.081e-05    7.081e-05
    K                   8.133e-03    8.133e-03
    Mg                  5.114e-02    5.114e-02
```

```
     N(5)                 8.902e-04    8.902e-04
     Na                   4.222e-01    4.222e-01
     P                    4.046e-05    4.046e-05
     S(6)                 2.528e-02    2.528e-02
     Si                   1.113e-04    1.113e-04
     Sr                   8.092e-05    8.092e-05
```

---------------------------Description of solution---------------------------

$$pH = 9.948 \quad Adjust$$

alkalinity

$$pe = 4.000$$

$$Activity\ of\ water = 0.983$$

$$Ionic\ strength\ (mol/kgw) = 5.899e-01$$

$$Mass\ of\ water\ (kg) = 1.000e+00$$

$$Total\ CO_2\ (mol/kg) = 1.649e-03$$

$$Temperature\ (癈) = 25.00$$

$$Electrical\ balance\ (eq) = 4.461e-03$$

$$Percent\ error,\ 100*(Cat-|An|)/(Cat+|An|) = 0.42$$

$$Iterations = 9$$

$$Total\ H = 1.110145e+02$$

$$Total\ O = 5.561739e+01$$

---------------------------Distribution of species---------------------------

```
                                            Log          Log
Log       mole V
   Species            Molality    Activity  Molality  Activity    Gamma
cm?mol

   OH-                1.408e-04   8.721e-05    -3.851    -4.059      -
0.208       (0)
   H+                 1.487e-10   1.128e-10    -9.828    -9.948      -
0.120        0.00
   H2O                5.551e+01   9.831e-01     1.744    -0.007
0.000       18.07
B              3.743e-04
   H2BO3-             3.407e-04   1.960e-04    -3.468    -3.708      -
0.240       (0)
   H3BO3              3.355e-05   3.843e-05    -4.474    -4.415
0.059       (0)
```

| | | | | | |
|---|---|---|---|---|---|
| BF(OH)3- | 6.156e-10 | 3.541e-10 | -9.211 | -9.451 | -0.240 (0) |
| BF2(OH)2- | 1.752e-16 | 1.008e-16 | -15.756 | -15.997 | -0.240 (0) |
| BF3OH- | 5.103e-25 | 2.935e-25 | -24.292 | -24.532 | -0.240 (0) |
| BF4- | 5.522e-33 | 3.176e-33 | -32.258 | -32.498 | -0.240 (0) |
| Br | 7.283e-04 | | | | |
| Br- | 7.283e-04 | 5.383e-04 | -3.138 | -3.269 | -0.131 (0) |
| C(-4) | 0.000e+00 | | | | |
| CH4 | 0.000e+00 | 0.000e+00 | -94.709 | -94.650 | 0.059 (0) |
| C(4) | 1.649e-03 | | | | |
| MgCO3 | 5.680e-04 | 6.507e-04 | -3.246 | -3.187 | 0.059 (0) |
| NaCO3- | 4.336e-04 | 2.960e-04 | -3.363 | -3.529 | -0.166 (0) |
| CO3-2 | 2.487e-04 | 5.400e-05 | -3.604 | -4.268 | -0.663 (0) |
| HCO3- | 1.904e-04 | 1.299e-04 | -3.720 | -3.886 | -0.166 (0) |
| CaCO3 | 1.540e-04 | 1.764e-04 | -3.813 | -3.754 | 0.059 (0) |
| MgHCO3+ | 3.007e-05 | 1.920e-05 | -4.522 | -4.717 | -0.195 (0) |
| NaHCO3 | 1.878e-05 | 2.151e-05 | -4.726 | -4.667 | 0.059 (0) |
| CaHCO3+ | 4.626e-06 | 3.231e-06 | -5.335 | -5.491 | -0.156 (0) |
| SrCO3 | 6.265e-07 | 6.265e-07 | -6.203 | -6.203 | 0.000 (0) |
| SrHCO3+ | 5.292e-08 | 3.612e-08 | -7.276 | -7.442 | -0.166 (0) |
| CO2 | 2.928e-08 | 3.354e-08 | -7.533 | -7.474 | 0.059 (0) |
| Ca | 8.679e-03 | | | | |
| Ca+2 | 7.682e-03 | 1.950e-03 | -2.115 | -2.710 | -0.596 (0) |
| CaSO4 | 8.298e-04 | 9.506e-04 | -3.081 | -3.022 | |

```
0.059      (0)
   CaCO3                 1.540e-04    1.764e-04      -3.813      -3.754
0.059      (0)
   CaHCO3+           4.626e-06    3.231e-06      -5.335      -5.491      -
0.156      (0)
   CaOH+              4.036e-06    2.819e-06      -5.394      -5.550      -
0.156      (0)
   CaPO4-             3.969e-06    2.709e-06      -5.401      -5.567      -
0.166      (0)
   CaF+               5.733e-07    3.930e-07      -6.242      -6.406      -
0.164      (0)
   CaHPO4                1.128e-07    1.292e-07      -6.948      -6.889
0.059      (0)
   CaH2PO4+          1.606e-11    1.096e-11    -10.794     -10.960      -
0.166      (0)
   CaHSO4+           8.502e-13    6.284e-13    -12.070     -12.202      -
0.131      (0)
Cl             4.891e-01
   Cl-              4.891e-01    3.092e-01      -0.311      -0.510      -0.199
(0)
F              7.081e-05
   F-               3.736e-05    2.314e-05      -4.428      -4.636      -
0.208      (0)
   MgF+              2.945e-05    1.930e-05      -4.531      -4.714      -
0.183      (0)
   NaF                 3.422e-06    3.919e-06      -5.466      -5.407
0.059      (0)
   CaF+               5.733e-07    3.930e-07      -6.242      -6.406      -
0.164      (0)
   BF(OH)3-          6.156e-10    3.541e-10      -9.211      -9.451      -
0.240      (0)
   HF                  3.419e-12    3.916e-12    -11.466     -11.407
0.059      (0)
   HF2-              5.614e-16    3.478e-16    -15.251     -15.459      -
0.208      (0)
   BF2(OH)2-         1.752e-16    1.008e-16    -15.756     -15.997      -
0.240      (0)
   H2F2                 3.489e-23    3.997e-23    -22.457     -22.398
0.059      (0)
   BF3OH-            5.103e-25    2.935e-25    -24.292     -24.532      -
0.240      (0)
```

|  | BF4- | 5.522e-33 | 3.176e-33 | -32.258 | -32.498 | -0.240 | (0) |
|  | SiF6-2 | 0.000e+00 | 0.000e+00 | -41.121 | -41.814 | -0.693 | (0) |
| H(0) | 1.574e-31 | | | | | | |
|  | H2 | 7.870e-32 | 9.015e-32 | -31.104 | -31.045 | 0.059 | (0) |
| K | 8.133e-03 | | | | | | |
|  | K+ | 8.006e-03 | 5.061e-03 | -2.097 | -2.296 | -0.199 | (0) |
|  | KSO4- | 1.273e-04 | 8.689e-05 | -3.895 | -4.061 | -0.166 | (0) |
|  | KHPO4- | 1.748e-09 | 1.193e-09 | -8.758 | -8.923 | -0.166 | (0) |
| Mg | 5.114e-02 | | | | | | |
|  | Mg+2 | 4.361e-02 | 1.263e-02 | -1.360 | -1.899 | -0.538 | (0) |
|  | MgSO4 | 6.314e-03 | 7.233e-03 | -2.200 | -2.141 | 0.059 | (0) |
|  | MgCO3 | 5.680e-04 | 6.507e-04 | -3.246 | -3.187 | 0.059 | (0) |
|  | MgOH+ | 5.620e-04 | 3.994e-04 | -3.250 | -3.399 | -0.148 | (0) |
|  | MgPO4- | 3.468e-05 | 2.367e-05 | -4.460 | -4.626 | -0.166 | (0) |
|  | MgHCO3+ | 3.007e-05 | 1.920e-05 | -4.522 | -4.717 | -0.195 | (0) |
|  | MgF+ | 2.945e-05 | 1.930e-05 | -4.531 | -4.714 | -0.183 | (0) |
|  | MgHPO4 | 9.878e-07 | 1.132e-06 | -6.005 | -5.946 | 0.059 | (0) |
|  | MgH2PO4+ | 1.324e-10 | 9.040e-11 | -9.878 | -10.044 | -0.166 | (0) |
| N(5) | 8.902e-04 | | | | | | |
|  | NO3- | 8.902e-04 | 5.327e-04 | -3.051 | -3.273 | -0.223 | (0) |
| Na | 4.222e-01 | | | | | | |
|  | Na+ | 4.164e-01 | 2.943e-01 | -0.380 | -0.531 | -0.151 | (0) |
|  | NaSO4- | 5.280e-03 | 3.604e-03 | -2.277 | -2.443 | -0.166 | (0) |

| Species | Molality | Activity | Log Molality | Log Activity | Log Gamma | |
|---|---|---|---|---|---|---|
| NaCO3- | 4.336e-04 | 2.960e-04 | -3.363 | -3.529 | -0.166 | (0) |
| NaHCO3 | 1.878e-05 | 2.151e-05 | -4.726 | -4.667 | 0.059 | (0) |
| NaF | 3.422e-06 | 3.919e-06 | -5.466 | -5.407 | 0.059 | (0) |
| NaHPO4- | 1.016e-07 | 6.938e-08 | -6.993 | -7.159 | -0.166 | (0) |
| O(0) | 8.656e-31 | | | | | |
| O2 | 4.328e-31 | 4.958e-31 | -30.364 | -30.305 | 0.059 | (0) |
| P | 4.046e-05 | | | | | |
| MgPO4- | 3.468e-05 | 2.367e-05 | -4.460 | -4.626 | -0.166 | (0) |
| CaPO4- | 3.969e-06 | 2.709e-06 | -5.401 | -5.567 | -0.166 | (0) |
| MgHPO4 | 9.878e-07 | 1.132e-06 | -6.005 | -5.946 | 0.059 | (0) |
| HPO4-2 | 5.960e-07 | 1.209e-07 | -6.225 | -6.918 | -0.693 | (0) |
| CaHPO4 | 1.128e-07 | 1.292e-07 | -6.948 | -6.889 | 0.059 | (0) |
| NaHPO4- | 1.016e-07 | 6.938e-08 | -6.993 | -7.159 | -0.166 | (0) |
| PO4-3 | 1.749e-08 | 4.830e-10 | -7.757 | -9.316 | -1.559 | (0) |
| KHPO4- | 1.748e-09 | 1.193e-09 | -8.758 | -8.923 | -0.166 | (0) |
| H2PO4- | 3.219e-10 | 2.197e-10 | -9.492 | -9.658 | -0.166 | (0) |
| MgH2PO4+ | 1.324e-10 | 9.040e-11 | -9.878 | -10.044 | -0.166 | (0) |
| CaH2PO4+ | 1.606e-11 | 1.096e-11 | -10.794 | -10.960 | -0.166 | (0) |
| S(6) | 2.528e-02 | | | | | |
| SO4-2 | 1.272e-02 | 2.444e-03 | -1.896 | -2.612 | -0.716 | (0) |
| MgSO4 | 6.314e-03 | 7.233e-03 | -2.200 | -2.141 | 0.059 | (0) |
| NaSO4- | 5.280e-03 | 3.604e-03 | -2.277 | -2.443 | -0.166 | (0) |

| CaSO4 | 8.298e-04 | 9.506e-04 | -3.081 | -3.022 | 0.059 | (0) |
| KSO4- | 1.273e-04 | 8.689e-05 | -3.895 | -4.061 | -0.166 | (0) |
| SrSO4 | 7.559e-06 | 8.659e-06 | -5.122 | -5.063 | 0.059 | (0) |
| HSO4- | 4.090e-11 | 2.681e-11 | -10.388 | -10.572 | -0.183 | (0) |
| CaHSO4+ | 8.502e-13 | 6.284e-13 | -12.070 | -12.202 | -0.131 | (0) |

Si            1.113e-04

| H3SiO4- | 7.791e-05 | 4.973e-05 | -4.108 | -4.303 | -0.195 | (0) |
| H4SiO4 | 3.322e-05 | 3.806e-05 | -4.479 | -4.420 | 0.059 | (0) |
| H2SiO4-2 | 1.381e-07 | 2.998e-08 | -6.860 | -7.523 | -0.663 | (0) |
| SiF6-2 | 0.000e+00 | 0.000e+00 | -41.121 | -41.814 | -0.693 | (0) |

Sr            8.092e-05

| Sr+2 | 7.267e-05 | 1.817e-05 | -4.139 | -4.741 | -0.602 | (0) |
| SrSO4 | 7.559e-06 | 8.659e-06 | -5.122 | -5.063 | 0.059 | (0) |
| SrCO3 | 6.265e-07 | 6.265e-07 | -6.203 | -6.203 | 0.000 | (0) |
| SrHCO3+ | 5.292e-08 | 3.612e-08 | -7.276 | -7.442 | -0.166 | (0) |
| SrOH+ | 1.210e-08 | 8.120e-09 | -7.917 | -8.090 | -0.173 | (0) |

-----------------------------Saturation indices-----------------------------

| Phase | SI** | log IAP | log K(298 K, | 1 atm) |
|---|---|---|---|---|
| Anhydrite | -0.96 | -5.32 | -4.36 | CaSO4 |
| Aragonite | 1.36 | -6.98 | -8.34 | CaCO3 |
| Artinite | 2.19 | 11.79 | 9.60 | MgCO3:Mg(OH)2:3H2O |
| Brucite | 1.14 | 17.98 | 16.84 | Mg(OH)2 |
| Calcite | 1.50 | -6.98 | -8.48 | CaCO3 |
| Celestite | -0.72 | -7.35 | -6.63 | SrSO4 |

| | | | | |
|---|---|---|---|---|
| CH4(g) | -91.79 | -94.65 | -2.86 | CH4 |
| Chalcedony | -0.85 | -4.40 | -3.55 | SiO2 |
| Chrysotile | 12.94 | 45.14 | 32.20 | Mg3Si2O5(OH)4 |
| Clinoenstatite | 2.24 | 13.58 | 11.34 | MgSiO3 |
| CO2(g) | -6.01 | -7.47 | -1.47 | CO2 |
| Cristobalite | -0.82 | -4.40 | -3.59 | SiO2 |
| Diopside | 6.46 | 26.36 | 19.89 | CaMgSi2O6 |
| Dolomite | 3.95 | -13.14 | -17.09 | CaMg(CO3)2 |
| Dolomite(d) | 3.40 | -13.14 | -16.54 | CaMg(CO3)2 |
| Epsomite | -2.42 | -4.56 | -2.14 | MgSO4:7H2O |
| FCO3Apatite | | 27.35 | -87.05 | -114.40 |
| Ca9.316Na0.36Mg0.144(PO4)4.8(CO3)1.2F2.48 | | | | |
| Fluorapatite | 8.50 | -9.10 | -17.60 | Ca5(PO4)3F |
| Fluorite | -1.38 | -11.98 | -10.60 | CaF2 |
| Forsterite | 3.27 | 31.57 | 28.31 | Mg2SiO4 |
| Gypsum | -0.76 | -5.34 | -4.58 | CaSO4:2H2O |
| H2(g) | -27.90 | -31.05 | -3.15 | H2 |
| H2O(g) | -1.52 | -0.01 | 1.51 | H2O |
| Halite | -2.62 | -1.04 | 1.58 | NaCl |
| Huntite | 4.49 | -25.48 | -29.97 | CaMg3(CO3)4 |
| Hydromagnesite | 2.05 | -6.71 | -8.76 | Mg5(CO3)4(OH)2:4H2O |
| Hydroxyapatite | 8.90 | 5.48 | -3.42 | Ca5(PO4)3OH |
| Magadiite | -7.15 | -21.45 | -14.30 | NaSi7O13(OH)3:3H2O |
| Magnesite | 1.86 | -6.17 | -8.03 | MgCO3 |
| Mirabilite | -2.63 | -3.75 | -1.11 | Na2SO4:10H2O |
| Nahcolite | -3.87 | -4.42 | -0.55 | NaHCO3 |
| Natron | -4.09 | -5.40 | -1.31 | Na2CO3:10H2O |
| Nesquehonite | -0.57 | -6.19 | -5.62 | MgCO3:3H2O |
| O2(g) | -27.41 | -30.30 | -2.89 | O2 |
| Portlandite | -5.63 | 17.17 | 22.80 | Ca(OH)2 |
| Quartz | -0.42 | -4.40 | -3.98 | SiO2 |
| Sepiolite | 6.98 | 22.74 | 15.76 | Mg2Si3O7.5OH:3H2O |
| Sepiolite(d) | 4.08 | 22.74 | 18.66 | Mg2Si3O7.5OH:3H2O |
| Silicagel | -1.39 | -4.40 | -3.02 | SiO2 |
| SiO2(a) | -1.69 | -4.40 | -2.71 | SiO2 |
| SrF2 | -5.47 | -14.01 | -8.54 | SrF2 |
| Strontianite | 0.26 | -9.01 | -9.27 | SrCO3 |
| Talc | 14.94 | 36.34 | 21.40 | Mg3Si4O10(OH)2 |
| Thenardite | -3.50 | -3.67 | -0.18 | Na2SO4 |
| Thermonatrite | -5.46 | -5.34 | 0.13 | Na2CO3:H2O |
| Tremolite | 32.48 | 89.05 | 56.57 | Ca2Mg5Si8O22(OH)2 |

| | |
|---|---|
| | Trona                 -8.97       -9.76      -0.80    NaHCO3:Na2CO3:2H2O

**For a gas, SI = log10(fugacity). Fugacity = pressure * phi / 1 atm.
    For ideal gases, phi = 1.
* * *
End of simulation.
* * ** * *
Reading input data for simulation 2.
* * ** * *
End of Run after 0.194 Seconds.
------------------------------ |
| Cyanobacteria 10 Day |    Input file: C:\Users\XUHENGCHAO\Desktop\PHreeQC\culture\C1-10.pqi
    Output file: C:\Users\XUHENGCHAO\Desktop\PHreeQC\culture\C1-10.pqo
Database    file:    C:\Program    Files\USGS\Phreeqc    Interactive    3.3.12-12704\database\wateq4f.dat
* * *
Reading data base.
* * *
     SOLUTION_MASTER_SPECIES
     SOLUTION_SPECIES
     PHASES
     EXCHANGE_MASTER_SPECIES
     EXCHANGE_SPECIES
     SURFACE_MASTER_SPECIES
     SURFACE_SPECIES
     RATES
     END
* * *
Reading input data for simulation 1.
* * *
     DATABASE    C:\Program    Files\USGS\Phreeqc    Interactive    3.3.12-12704\database\wateq4f.dat |

```
SOLUTION 1
        temp        25
        pH          8
        pe          4
        redox       pe
        units       mmol/l
        density     1.02
        Alkalinity 1.48
        B           0.37
        Br          0.72
        C           1.02
        Ca          8.4
        Cl          483.51
        F           0.07
        K           8.04
        Mg          48.69
        N(5)        0.88
        Na          417.34
        P           0.04
        S(6)        24.99
        Si          0.11
        Sr          0.08
        water       1 # kg
* * *
Beginning of initial solution calculations.
* * *
Initial solution 1.

pH will be adjusted to obtain desired alkalinity.

----------------------------Solution composition----------------------------

    Elements              Molality           Moles

    Alkalinity          1.497e-03     1.497e-03
    B                   3.742e-04     3.742e-04
    Br                  7.282e-04     7.282e-04
    C                   1.032e-03     1.032e-03
    Ca                  8.495e-03     8.495e-03
```

|      |            |            |
|------|------------|------------|
| Cl   | 4.890e-01  | 4.890e-01  |
| F    | 7.079e-05  | 7.079e-05  |
| K    | 8.131e-03  | 8.131e-03  |
| Mg   | 4.924e-02  | 4.924e-02  |
| N(5) | 8.900e-04  | 8.900e-04  |
| Na   | 4.221e-01  | 4.221e-01  |
| P    | 4.045e-05  | 4.045e-05  |
| S(6) | 2.527e-02  | 2.527e-02  |
| Si   | 1.112e-04  | 1.112e-04  |
| Sr   | 8.091e-05  | 8.091e-05  |

--------------------------Description of solution--------------------------

pH    =    8.652          Adjust alkalinity

pe    =    4.000

Activity of water   =    0.983

Ionic strength (mol/kgw)   =    5.874e-01

Mass of water (kg)   =    1.000e+00

Total CO2 (mol/kg)   =    1.032e-03

Temperature (癈)   =   25.00

Electrical balance (eq)   =    3.074e-03

Percent error, 100*(Cat-|An|)/(Cat+|An|)   =    0.29

Iterations   =   10

Total H   = 1.110147e+02

Total O   = 5.561484e+01

--------------------------Distribution of species--------------------------

|         |          |          | Log      | Log      | Log       | mole V   |
|---------|----------|----------|----------|----------|-----------|----------|
| Species | Molality | Activity | Molality | Activity | Gamma     | cm?mol   |
| OH-     | 7.134e-06 | 4.421e-06 | -5.147  | -5.354   | -0.208    | (0)      |
| H+      | 2.932e-09 | 2.226e-09 | -8.533  | -8.652   | -0.120    | 0.00     |
| H2O     | 5.551e+01 | 9.831e-01 | 1.744   | -0.007   | 0.000     | 18.07    |
| B       | 3.742e-04 |          |          |          |           |          |

| Species | Molality | Activity | Log Molality | Log Activity | Log Gamma |
|---|---|---|---|---|---|
| H3BO3 | 2.471e-04 | 2.829e-04 | -3.607 | -3.548 | 0.059 (0) |
| H2BO3- | 1.271e-04 | 7.314e-05 | -3.896 | -4.136 | -0.240 (0) |
| BF(OH)3- | 4.567e-09 | 2.628e-09 | -8.340 | -8.580 | -0.240 (0) |
| BF2(OH)2- | 2.586e-14 | 1.488e-14 | -13.587 | -13.827 | -0.240 (0) |
| BF3OH- | 1.498e-21 | 8.623e-22 | -20.824 | -21.064 | -0.240 (0) |
| BF4- | 3.225e-28 | 1.856e-28 | -27.492 | -27.731 | -0.240 (0) |

Br              7.282e-04

| Species | Molality | Activity | Log Molality | Log Activity | Log Gamma |
|---|---|---|---|---|---|
| Br- | 7.282e-04 | 5.380e-04 | -3.138 | -3.269 | -0.131 (0) |

C(-4)           0.000e+00

| Species | Molality | Activity | Log Molality | Log Activity | Log Gamma |
|---|---|---|---|---|---|
| CH4 | 0.000e+00 | 0.000e+00 | -82.537 | -82.478 | 0.059 (0) |

C(4)            1.032e-03

| Species | Molality | Activity | Log Molality | Log Activity | Log Gamma |
|---|---|---|---|---|---|
| HCO3- | 6.244e-04 | 4.264e-04 | -3.205 | -3.370 | -0.166 (0) |
| MgHCO3+ | 9.669e-05 | 6.175e-05 | -4.015 | -4.209 | -0.195 (0) |
| MgCO3 | 9.267e-05 | 1.061e-04 | -4.033 | -3.974 | 0.059 (0) |
| NaCO3- | 7.213e-05 | 4.926e-05 | -4.142 | -4.308 | -0.166 (0) |
| NaHCO3 | 6.167e-05 | 7.060e-05 | -4.210 | -4.151 | 0.059 (0) |
| CO3-2 | 4.131e-05 | 8.983e-06 | -4.384 | -5.047 | -0.663 (0) |
| CaCO3 | 2.544e-05 | 2.913e-05 | -4.594 | -4.536 | 0.059 (0) |
| CaHCO3+ | 1.506e-05 | 1.052e-05 | -4.822 | -4.978 | -0.156 (0) |
| CO2 | 1.896e-06 | 2.171e-06 | -5.722 | -5.663 | 0.059 (0) |
| SrHCO3+ | 1.744e-07 | 1.191e-07 | -6.758 | -6.924 | -0.166 (0) |
| SrCO3 | 1.047e-07 | 1.047e-07 | -6.980 | -6.980 | 0.000 (0) |

Ca                8.495e-03
   Ca+2                    7.622e-03    1.935e-03    -2.118    -2.713    -0.595    (0)
   CaSO4                    8.289e-04    9.489e-04    -3.082    -3.023    0.059    (0)
   CaCO3                    2.544e-05    2.913e-05    -4.594    -4.536    0.059    (0)
   CaHCO3+                  1.506e-05    1.052e-05    -4.822    -4.978    -0.156    (0)
   CaPO4-                   2.176e-06    1.486e-06    -5.662    -5.828    -0.166    (0)
   CaHPO4                   1.221e-06    1.398e-06    -5.913    -5.854    0.059    (0)
   CaF+                     5.738e-07    3.934e-07    -6.241    -6.405    -0.164    (0)
   CaOH+                    2.030e-07    1.419e-07    -6.692    -6.848    -0.156    (0)
   CaH2PO4+                 3.425e-09    2.339e-09    -8.465    -8.631    -0.166    (0)
   CaHSO4+                  1.675e-11    1.237e-11    -10.776   -10.907   -0.131    (0)
Cl                4.890e-01
   Cl-                      4.890e-01    3.093e-01    -0.311    -0.510    -0.199    (0)
F                 7.079e-05
   F-                       3.766e-05    2.334e-05    -4.424    -4.632    -0.208    (0)
   MgF+                     2.910e-05    1.908e-05    -4.536    -4.719    -0.183    (0)
   NaF                      3.454e-06    3.954e-06    -5.462    -5.403    0.059    (0)
   CaF+                     5.738e-07    3.934e-07    -6.241    -6.405    -0.164    (0)
   BF(OH)3-                 4.567e-09    2.628e-09    -8.340    -8.580    -0.240    (0)
   HF                       6.805e-11    7.791e-11    -10.167   -10.108   0.059    (0)
   BF2(OH)2-                2.586e-14    1.488e-14    -13.587   -13.827   -0.240    (0)
   HF2-                     1.126e-14    6.976e-15    -13.949   -14.156   -0.208    (0)

|  | H2F2 | 1.382e-20 | 1.582e-20 | -19.860 | -19.801 | 0.059 | (0) |
|  | BF3OH- | 1.498e-21 | 8.623e-22 | -20.824 | -21.064 | -0.240 | (0) |
|  | BF4- | 3.225e-28 | 1.856e-28 | -27.492 | -27.731 | -0.240 | (0) |
|  | SiF6-2 | 3.601e-36 | 7.316e-37 | -35.444 | -36.136 | -0.692 | (0) |
| H(0) | 6.128e-29 |  |  |  |  |  |  |
|  | H2 | 3.064e-29 | 3.508e-29 | -28.514 | -28.455 | 0.059 | (0) |
| K | 8.131e-03 |  |  |  |  |  |  |
|  | K+ | 8.003e-03 | 5.061e-03 | -2.097 | -2.296 | -0.199 | (0) |
|  | KSO4- | 1.280e-04 | 8.739e-05 | -3.893 | -4.059 | -0.166 | (0) |
|  | KHPO4- | 1.904e-08 | 1.300e-08 | -7.720 | -7.886 | -0.166 | (0) |
| Mg | 4.924e-02 |  |  |  |  |  |  |
|  | Mg+2 | 4.274e-02 | 1.238e-02 | -1.369 | -1.907 | -0.538 | (0) |
|  | MgSO4 | 6.228e-03 | 7.130e-03 | -2.206 | -2.147 | 0.059 | (0) |
|  | MgHCO3+ | 9.669e-05 | 6.175e-05 | -4.015 | -4.209 | -0.195 | (0) |
|  | MgCO3 | 9.267e-05 | 1.061e-04 | -4.033 | -3.974 | 0.059 | (0) |
|  | MgF+ | 2.910e-05 | 1.908e-05 | -4.536 | -4.719 | -0.183 | (0) |
|  | MgOH+ | 2.792e-05 | 1.985e-05 | -4.554 | -4.702 | -0.148 | (0) |
|  | MgPO4- | 1.877e-05 | 1.282e-05 | -4.726 | -4.892 | -0.166 | (0) |
|  | MgHPO4 | 1.056e-05 | 1.209e-05 | -4.976 | -4.918 | 0.059 | (0) |
|  | MgH2PO4+ | 2.790e-08 | 1.905e-08 | -7.554 | -7.720 | -0.166 | (0) |
| N(5) | 8.900e-04 |  |  |  |  |  |  |
|  | NO3- | 8.900e-04 | 5.329e-04 | -3.051 | -3.273 | -0.223 | (0) |
| Na | 4.221e-01 |  |  |  |  |  |  |

| Species | Molality | Activity | Log Molality | Log Activity | Log Gamma | |
|---|---|---|---|---|---|---|
| Na+ | 4.166e-01 | 2.945e-01 | -0.380 | -0.531 | -0.151 | (0) |
| NaSO4- | 5.311e-03 | 3.627e-03 | -2.275 | -2.440 | -0.166 | (0) |
| NaCO3- | 7.213e-05 | 4.926e-05 | -4.142 | -4.308 | -0.166 | (0) |
| NaHCO3 | 6.167e-05 | 7.060e-05 | -4.210 | -4.151 | 0.059 | (0) |
| NaF | 3.454e-06 | 3.954e-06 | -5.462 | -5.403 | 0.059 | (0) |
| NaHPO4- | 1.108e-06 | 7.565e-07 | -5.956 | -6.121 | -0.166 | (0) |
| O(0) | 5.720e-36 | | | | | |
| O2 | 2.860e-36 | 3.274e-36 | -35.544 | -35.485 | 0.059 | (0) |
| P | 4.045e-05 | | | | | |
| MgPO4- | 1.877e-05 | 1.282e-05 | -4.726 | -4.892 | -0.166 | (0) |
| MgHPO4 | 1.056e-05 | 1.209e-05 | -4.976 | -4.918 | 0.059 | (0) |
| HPO4-2 | 6.486e-06 | 1.318e-06 | -5.188 | -5.880 | -0.692 | (0) |
| CaPO4- | 2.176e-06 | 1.486e-06 | -5.662 | -5.828 | -0.166 | (0) |
| CaHPO4 | 1.221e-06 | 1.398e-06 | -5.913 | -5.854 | 0.059 | (0) |
| NaHPO4- | 1.108e-06 | 7.565e-07 | -5.956 | -6.121 | -0.166 | (0) |
| H2PO4- | 6.918e-08 | 4.724e-08 | -7.160 | -7.326 | -0.166 | (0) |
| MgH2PO4+ | 2.790e-08 | 1.905e-08 | -7.554 | -7.720 | -0.166 | (0) |
| KHPO4- | 1.904e-08 | 1.300e-08 | -7.720 | -7.886 | -0.166 | (0) |
| PO4-3 | 9.632e-09 | 2.668e-10 | -8.016 | -9.574 | -1.557 | (0) |
| CaH2PO4+ | 3.425e-09 | 2.339e-09 | -8.465 | -8.631 | -0.166 | (0) |
| S(6) | 2.527e-02 | | | | | |
| SO4-2 | 1.277e-02 | 2.457e-03 | -1.894 | -2.610 | -0.716 | (0) |

| Species | Molality | Activity | Log Molality | Log Activity | Log Gamma | |
|---|---|---|---|---|---|---|
| MgSO4 | 6.228e-03 | 7.130e-03 | -2.206 | -2.147 | 0.059 | (0) |
| NaSO4- | 5.311e-03 | 3.627e-03 | -2.275 | -2.440 | -0.166 | (0) |
| CaSO4 | 8.289e-04 | 9.489e-04 | -3.082 | -3.023 | 0.059 | (0) |
| KSO4- | 1.280e-04 | 8.739e-05 | -3.893 | -4.059 | -0.166 | (0) |
| SrSO4 | 7.644e-06 | 8.751e-06 | -5.117 | -5.058 | 0.059 | (0) |
| HSO4- | 8.110e-10 | 5.318e-10 | -9.091 | -9.274 | -0.183 | (0) |
| CaHSO4+ | 1.675e-11 | 1.237e-11 | -10.776 | -10.907 | -0.131 | (0) |

Si             1.112e-04

| Species | Molality | Activity | Log Molality | Log Activity | Log Gamma | |
|---|---|---|---|---|---|---|
| H4SiO4 | 9.944e-05 | 1.138e-04 | -4.002 | -3.944 | 0.059 | (0) |
| H3SiO4- | 1.181e-05 | 7.541e-06 | -4.928 | -5.123 | -0.195 | (0) |
| H2SiO4-2 | 1.060e-09 | 2.304e-10 | -8.975 | -9.637 | -0.663 | (0) |
| SiF6-2 | 3.601e-36 | 7.316e-37 | -35.444 | -36.136 | -0.692 | (0) |

Sr             8.091e-05

| Species | Molality | Activity | Log Molality | Log Activity | Log Gamma | |
|---|---|---|---|---|---|---|
| Sr+2 | 7.298e-05 | 1.826e-05 | -4.137 | -4.738 | -0.602 | (0) |
| SrSO4 | 7.644e-06 | 8.751e-06 | -5.117 | -5.058 | 0.059 | (0) |
| SrHCO3+ | 1.744e-07 | 1.191e-07 | -6.758 | -6.924 | -0.166 | (0) |
| SrCO3 | 1.047e-07 | 1.047e-07 | -6.980 | -6.980 | 0.000 | (0) |
| SrOH+ | 6.162e-10 | 4.137e-10 | -9.210 | -9.383 | -0.173 | (0) |

----------------------------Saturation indices----------------------------

| Phase | SI** | log IAP | log K(298 K, | 1 atm) | |
|---|---|---|---|---|---|
| Anhydrite | -0.96 | -5.32 | -4.36 | CaSO4 | |
| Aragonite | 0.58 | -7.76 | -8.34 | CaCO3 | |

| | | | | |
|---|---|---|---|---|
| Artinite | -1.19 | 8.41 | 9.60 | MgCO3:Mg(OH)2:3H2O |
| Brucite | -1.46 | 15.38 | 16.84 | Mg(OH)2 |
| Calcite | 0.72 | -7.76 | -8.48 | CaCO3 |
| Celestite | -0.72 | -7.35 | -6.63 | SrSO4 |
| CH4(g) | -79.62 | -82.48 | -2.86 | CH4 |
| Chalcedony | -0.38 | -3.93 | -3.55 | SiO2 |
| Chrysotile | 6.10 | 38.30 | 32.20 | Mg3Si2O5(OH)4 |
| Clinoenstatite | 0.12 | 11.46 | 11.34 | MgSiO3 |
| CO2(g) | -4.20 | -5.66 | -1.47 | CO2 |
| Cristobalite | -0.34 | -3.93 | -3.59 | SiO2 |
| Diopside | 2.22 | 22.12 | 19.89 | CaMgSi2O6 |
| Dolomite | 2.38 | -14.71 | -17.09 | CaMg(CO3)2 |
| Dolomite(d) | 1.83 | -14.71 | -16.54 | CaMg(CO3)2 |
| Epsomite | -2.43 | -4.57 | -2.14 | MgSO4:7H2O |
| FCO3Apatite | | 25.16 | | -89.24 | -114.40 |
| Ca9.316Na0.36Mg0.144(PO4)4.8(CO3)1.2F2.48 | | | | |
| Fluorapatite | 7.72 | -9.88 | -17.60 | Ca5(PO4)3F |
| Fluorite | -1.38 | -11.98 | -10.60 | CaF2 |
| Forsterite | -1.45 | 26.85 | 28.31 | Mg2SiO4 |
| Gypsum | -0.76 | -5.34 | -4.58 | CaSO4:2H2O |
| H2(g) | -25.30 | -28.45 | -3.15 | H2 |
| H2O(g) | -1.52 | -0.01 | 1.51 | H2O |
| Halite | -2.62 | -1.04 | 1.58 | NaCl |
| Huntite | 1.35 | -28.62 | -29.97 | CaMg3(CO3)4 |
| Hydromagnesite | -3.70 | -12.46 | -8.76 | Mg5(CO3)4(OH)2:4H2O |
| Hydroxyapatite | 6.82 | 3.40 | -3.42 | Ca5(PO4)3OH |
| Magadiite | -5.12 | -19.42 | -14.30 | NaSi7O13(OH)3:3H2O |
| Magnesite | 1.08 | -6.95 | -8.03 | MgCO3 |
| Mirabilite | -2.63 | -3.75 | -1.11 | Na2SO4:10H2O |
| Nahcolite | -3.35 | -3.90 | -0.55 | NaHCO3 |
| Natron | -4.87 | -6.18 | -1.31 | Na2CO3:10H2O |
| Nesquehonite | -1.36 | -6.98 | -5.62 | MgCO3:3H2O |
| O2(g) | -32.59 | -35.48 | -2.89 | O2 |
| Portlandite | -8.22 | 14.58 | 22.80 | Ca(OH)2 |
| Quartz | 0.05 | -3.93 | -3.98 | SiO2 |
| Sepiolite | 3.21 | 18.97 | 15.76 | Mg2Si3O7.5OH:3H2O |
| Sepiolite(d) | 0.31 | 18.97 | 18.66 | Mg2Si3O7.5OH:3H2O |
| Silicagel | -0.91 | -3.93 | -3.02 | SiO2 |
| SiO2(a) | -1.22 | -3.93 | -2.71 | SiO2 |
| SrF2 | -5.46 | -14.00 | -8.54 | SrF2 |
| Strontianite | -0.51 | -9.79 | -9.27 | SrCO3 |

| | |
|---|---|
| | Talc                 9.05      30.45     21.40    Mg3Si4O10(OH)2 |
| | Thenardite        -3.49      -3.67    -0.18    Na2SO4 |
| | Thermonatrite    -6.24      -6.12     0.13    Na2CO3:H2O |
| | Tremolite        18.11      74.68    56.57    Ca2Mg5Si8O22(OH)2 |
| | Trona            -9.23     -10.02    -0.80    NaHCO3:Na2CO3:2H2O |
| | |
| | **For a gas, SI = log10(fugacity). Fugacity = pressure * phi / 1 atm. |
| |    For ideal gases, phi = 1. |
| | |
| | ------------------ |
| | End of simulation. |
| | ------------------ |
| | |
| | |
| | ---------------------------------- |
| | Reading input data for simulation 2. |
| | ---------------------------------- |
| | |
| | |
| | ------------------------------ |
| | End of Run after 0.156 Seconds. |
| | ------------------------------ |
| Cyanobacteria 12 Day |     Input file: C:\Users\XUHENGCHAO\Desktop\PHreeQC\culture\C1-13.pqi |
| |     Output file: C:\Users\XUHENGCHAO\Desktop\PHreeQC\culture\C1-13.pqo |
| | Database    file:    C:\Program     Files\USGS\Phreeqc     Interactive     3.3.12-12704\database\wateq4f.dat |
| | |
| | ------------------ |
| | Reading data base. |
| | ------------------ |
| | |
| |     SOLUTION_MASTER_SPECIES |
| |     SOLUTION_SPECIES |
| |     PHASES |
| |     EXCHANGE_MASTER_SPECIES |
| |     EXCHANGE_SPECIES |
| |     SURFACE_MASTER_SPECIES |
| |     SURFACE_SPECIES |
| |     RATES |
| |     END |
| | ---------------------------------- |
| | Reading input data for simulation 1. |

```
* * *
     DATABASE    C:\Program    Files\USGS\Phreeqc    Interactive    3.3.12-
12704\database\wateq4f.dat
     SOLUTION 1
          temp        25
          pH          8
          pe          4
          redox       pe
          units       mmol/l
          density     1.02
          Alkalinity 1.48
          B           0.37
          Br          0.72
          C           0.84
          Ca          8.32
          Cl          483.51
          F           0.07
          K           8.04
          Mg          48.51
          N(5)        0.88
          Na          417.34
          P           0.04
          S(6)        24.99
          Si          0.11
          Sr          0.08
          water       1 # kg
* * *
Beginning of initial solution calculations.
* * *
Initial solution 1.

pH will be adjusted to obtain desired alkalinity.

----------------------------Solution composition----------------------------

     Elements              Molality          Moles

     Alkalinity        1.497e-03      1.497e-03
```

|       |           |           |
|-------|-----------|-----------|
| B     | 3.742e-04 | 3.742e-04 |
| Br    | 7.281e-04 | 7.281e-04 |
| C     | 8.495e-04 | 8.495e-04 |
| Ca    | 8.414e-03 | 8.414e-03 |
| Cl    | 4.890e-01 | 4.890e-01 |
| F     | 7.079e-05 | 7.079e-05 |
| K     | 8.131e-03 | 8.131e-03 |
| Mg    | 4.906e-02 | 4.906e-02 |
| N(5)  | 8.899e-04 | 8.899e-04 |
| Na    | 4.221e-01 | 4.221e-01 |
| P     | 4.045e-05 | 4.045e-05 |
| S(6)  | 2.527e-02 | 2.527e-02 |
| Si    | 1.112e-04 | 1.112e-04 |
| Sr    | 8.090e-05 | 8.090e-05 |

---------------------------Description of solution---------------------------

$$pH = 8.939 \quad \text{Adjust alkalinity}$$

$$pe = 4.000$$

Activity of water   =   0.983

Ionic strength (mol/kgw)   =   5.868e-01

Mass of water (kg)   =   1.000e+00

Total CO2 (mol/kg)   =   8.495e-04

Temperature (癈)   =   25.00

Electrical balance (eq)   =   2.548e-03

Percent error, 100*(Cat-|An|)/(Cat+|An|)   =   0.24

Iterations   =   9

Total H   = 1.110144e+02

Total O   = 5.561432e+01

---------------------------Distribution of species---------------------------

|         |          |          | Log      | Log      | Log    | mole V |
|---------|----------|----------|----------|----------|--------|--------|
| Species | Molality | Activity | Molality | Activity | Gamma  | cm?mol |
| OH-     | 1.380e-05 | 8.553e-06 | -4.860 | -5.068 | -0.208 | (0) |
| H+      | 1.516e-09 | 1.151e-09 | -8.819 | -8.939 | -      |      |

```
          0.120            0.00
   H2O                          5.551e+01      9.831e-01        1.744       -0.007
0.000         18.07
B                3.742e-04
   H3BO3                         1.876e-04      2.148e-04       -3.727       -3.668
0.059       (0)
   H2BO3-                        1.866e-04      1.074e-04       -3.729       -3.969        -
0.240       (0)
   BF(OH)3-                      3.473e-09      1.999e-09       -8.459       -8.699        -
0.240       (0)
   BF2(OH)2-                     1.019e-14      5.863e-15      -13.992      -14.232        -
0.240       (0)
   BF3OH-                        3.056e-22      1.760e-22      -21.515      -21.755        -
0.240       (0)
   BF4-                          3.408e-29      1.962e-29      -28.468      -28.707        -
0.240       (0)
Br               7.281e-04
   Br-                           7.281e-04      5.380e-04       -3.138       -3.269        -
0.131       (0)
C(-4)            0.000e+00
   CH4                           0.000e+00      0.000e+00      -85.282      -85.223
0.059       (0)
C(4)             8.495e-04
   HCO3-                         4.259e-04      2.909e-04       -3.371       -3.536        -
0.166       (0)
   MgCO3                         1.218e-04      1.394e-04       -3.914       -3.856
0.059       (0)
   NaCO3-                        9.518e-05      6.500e-05       -4.021       -4.187        -
0.166       (0)
   MgHCO3+                       6.566e-05      4.194e-05       -4.183       -4.377        -
0.195       (0)
   CO3-2                         5.451e-05      1.186e-05       -4.264       -4.926        -
0.663       (0)
   NaHCO3                        4.208e-05      4.817e-05       -4.376       -4.317
0.059       (0)
   CaCO3                         3.324e-05      3.805e-05       -4.478       -4.420
0.059       (0)
   CaHCO3+                       1.017e-05      7.108e-06       -4.993       -5.148        -
0.156       (0)
   CO2                           6.687e-07      7.654e-07       -6.175       -6.116
0.059       (0)
```

```
	SrCO3			1.383e-07	1.383e-07	-6.859	-6.859	0.000	(0)
	SrHCO3+			1.190e-07	8.128e-08	-6.924	-7.090	-0.166	(0)
Ca		8.414e-03
	Ca+2			7.544e-03	1.916e-03	-2.122	-2.718	-0.595	(0)
	CaSO4			8.220e-04	9.409e-04	-3.085	-3.026	0.059	(0)
	CaCO3			3.324e-05	3.805e-05	-4.478	-4.420	0.059	(0)
	CaHCO3+			1.017e-05	7.108e-06	-4.993	-5.148	-0.156	(0)
	CaPO4-			2.822e-06	1.927e-06	-5.549	-5.715	-0.166	(0)
	CaHPO4			8.189e-07	9.374e-07	-6.087	-6.028	0.059	(0)
	CaF+			5.691e-07	3.902e-07	-6.245	-6.409	-0.164	(0)
	CaOH+			3.888e-07	2.717e-07	-6.410	-6.566	-0.156	(0)
	CaH2PO4+		1.187e-09	8.107e-10	-8.925	-9.091	-0.166	(0)
	CaHSO4+			8.584e-12	6.342e-12	-11.066	-11.198	-0.131	(0)
Cl		4.890e-01
	Cl-			4.890e-01	3.093e-01	-0.311	-0.510	-0.199	(0)
F		7.079e-05
	F-			3.773e-05	2.338e-05	-4.423	-4.631	-0.208	(0)
	MgF+			2.903e-05	1.904e-05	-4.537	-4.720	-0.183	(0)
	NaF			3.461e-06	3.962e-06	-5.461	-5.402	0.059	(0)
	CaF+			5.691e-07	3.902e-07	-6.245	-6.409	-0.164	(0)
	BF(OH)3-		3.473e-09	1.999e-09	-8.459	-8.699	-0.240	(0)
	HF			3.525e-11	4.035e-11	-10.453	-10.394	0.059	(0)
```

| Species | Molality | Activity | Log Molality | Log Activity | Log Gamma |
|---|---|---|---|---|---|
| BF2(OH)2- | 1.019e-14 | 5.863e-15 | -13.992 | -14.232 | -0.240 (0) |
| HF2- | 5.842e-15 | 3.621e-15 | -14.233 | -14.441 | -0.208 (0) |
| H2F2 | 3.707e-21 | 4.244e-21 | -20.431 | -20.372 | 0.059 (0) |
| BF3OH- | 3.056e-22 | 1.760e-22 | -21.515 | -21.755 | -0.240 (0) |
| BF4- | 3.408e-29 | 1.962e-29 | -28.468 | -28.707 | -0.240 (0) |
| SiF6-2 | 2.367e-37 | 4.809e-38 | -36.626 | -37.318 | -0.692 (0) |

H(0)          1.638e-29

| Species | Molality | Activity | Log Molality | Log Activity | Log Gamma |
|---|---|---|---|---|---|
| H2 | 8.189e-30 | 9.373e-30 | -29.087 | -29.028 | 0.059 (0) |

K             8.131e-03

| Species | Molality | Activity | Log Molality | Log Activity | Log Gamma |
|---|---|---|---|---|---|
| K+ | 8.003e-03 | 5.062e-03 | -2.097 | -2.296 | -0.199 (0) |
| KSO4- | 1.282e-04 | 8.754e-05 | -3.892 | -4.058 | -0.166 (0) |
| KHPO4- | 1.290e-08 | 8.808e-09 | -7.890 | -8.055 | -0.166 (0) |

Mg            4.906e-02

| Species | Molality | Activity | Log Molality | Log Activity | Log Gamma |
|---|---|---|---|---|---|
| Mg+2 | 4.255e-02 | 1.232e-02 | -1.371 | -1.909 | -0.538 (0) |
| MgSO4 | 6.211e-03 | 7.110e-03 | -2.207 | -2.148 | 0.059 (0) |
| MgCO3 | 1.218e-04 | 1.394e-04 | -3.914 | -3.856 | 0.059 (0) |
| MgHCO3+ | 6.566e-05 | 4.194e-05 | -4.183 | -4.377 | -0.195 (0) |
| MgOH+ | 5.378e-05 | 3.823e-05 | -4.269 | -4.418 | -0.148 (0) |
| MgF+ | 2.903e-05 | 1.904e-05 | -4.537 | -4.720 | -0.183 (0) |
| MgPO4- | 2.449e-05 | 1.672e-05 | -4.611 | -4.777 | -0.166 (0) |
| MgHPO4 | 7.121e-06 | 8.152e-06 | -5.147 | -5.089 | 0.059 (0) |
| MgH2PO4+ | 9.724e-09 | 6.640e-09 | -8.012 | -8.178 | -0.166 (0) |

| | | | | | | |
|---|---|---|---|---|---|---|
| N(5) | 8.899e-04 | | | | | |
| NO3- | | 8.899e-04 | 5.330e-04 | -3.051 | -3.273 | -0.223 (0) |
| Na | 4.221e-01 | | | | | |
| Na+ | | 4.166e-01 | 2.945e-01 | -0.380 | -0.531 | -0.151 (0) |
| NaSO4- | | 5.319e-03 | 3.632e-03 | -2.274 | -2.440 | -0.166 (0) |
| NaCO3- | | 9.518e-05 | 6.500e-05 | -4.021 | -4.187 | -0.166 (0) |
| NaHCO3 | | 4.208e-05 | 4.817e-05 | -4.376 | -4.317 | 0.059 (0) |
| NaF | | 3.461e-06 | 3.962e-06 | -5.461 | -5.402 | 0.059 (0) |
| NaHPO4- | | 7.503e-07 | 5.124e-07 | -6.125 | -6.290 | -0.166 (0) |
| O(0) | 8.013e-35 | | | | | |
| O2 | | 4.006e-35 | 4.586e-35 | -34.397 | -34.339 | 0.059 (0) |
| P | 4.045e-05 | | | | | |
| MgPO4- | | 2.449e-05 | 1.672e-05 | -4.611 | -4.777 | -0.166 (0) |
| MgHPO4 | | 7.121e-06 | 8.152e-06 | -5.147 | -5.089 | 0.059 (0) |
| HPO4-2 | | 4.392e-06 | 8.924e-07 | -5.357 | -6.049 | -0.692 (0) |
| CaPO4- | | 2.822e-06 | 1.927e-06 | -5.549 | -5.715 | -0.166 (0) |
| CaHPO4 | | 8.189e-07 | 9.374e-07 | -6.087 | -6.028 | 0.059 (0) |
| NaHPO4- | | 7.503e-07 | 5.124e-07 | -6.125 | -6.290 | -0.166 (0) |
| H2PO4- | | 2.422e-08 | 1.654e-08 | -7.616 | -7.781 | -0.166 (0) |
| KHPO4- | | 1.290e-08 | 8.808e-09 | -7.890 | -8.055 | -0.166 (0) |
| PO4-3 | | 1.261e-08 | 3.496e-10 | -7.899 | -9.456 | -1.557 (0) |
| MgH2PO4+ | | 9.724e-09 | 6.640e-09 | -8.012 | -8.178 | -0.166 (0) |
| CaH2PO4+ | | 1.187e-09 | 8.107e-10 | -8.925 | -9.091 | - |

0.166      (0)

S(6)            2.527e-02
    SO4-2              1.278e-02    2.461e-03     -1.893     -2.609     -
0.716      (0)
    MgSO4                 6.211e-03    7.110e-03     -2.207     -2.148
0.059      (0)
    NaSO4-              5.319e-03    3.632e-03     -2.274     -2.440     -
0.166      (0)
    CaSO4                 8.220e-04    9.409e-04     -3.085     -3.026
0.059      (0)
    KSO4-               1.282e-04    8.754e-05     -3.892     -4.058     -
0.166      (0)
    SrSO4                 7.659e-06    8.767e-06     -5.116     -5.057
0.059      (0)
    HSO4-               4.199e-10    2.753e-10     -9.377     -9.560     -
0.183      (0)
    CaHSO4+              8.584e-12    6.342e-12    -11.066    -11.198     -
0.131      (0)
Si            1.112e-04
    H4SiO4                9.046e-05    1.036e-04     -4.044     -3.985
0.059      (0)
    H3SiO4-             2.078e-05    1.327e-05     -4.682     -4.877     -
0.195      (0)
    H2SiO4-2            3.607e-09    7.845e-10     -8.443     -9.105     -
0.663      (0)
    SiF6-2             2.367e-37    4.809e-38    -36.626    -37.318    -0.692
(0)
Sr            8.090e-05
    Sr+2                 7.299e-05    1.827e-05     -4.137     -4.738     -
0.602      (0)
    SrSO4                 7.659e-06    8.767e-06     -5.116     -5.057
0.059      (0)
    SrCO3                 1.383e-07    1.383e-07     -6.859     -6.859
0.000      (0)
    SrHCO3+             1.190e-07    8.128e-08     -6.924     -7.090     -
0.166      (0)
    SrOH+               1.192e-09    8.005e-10     -8.924     -9.097     -
0.173      (0)

-----------------------------Saturation indices-----------------------------

| Phase | SI** | log IAP | log K(298 K, | 1 atm) | |
|---|---|---|---|---|---|
| Anhydrite | -0.97 | -5.33 | -4.36 | CaSO4 | |
| Aragonite | 0.69 | -7.64 | -8.34 | CaCO3 | |
| Artinite | -0.50 | 9.10 | 9.60 | MgCO3:Mg(OH)2:3H2O | |
| Brucite | -0.89 | 15.95 | 16.84 | Mg(OH)2 | |
| Calcite | 0.84 | -7.64 | -8.48 | CaCO3 | |
| Celestite | -0.72 | -7.35 | -6.63 | SrSO4 | |
| CH4(g) | -82.36 | -85.22 | -2.86 | CH4 | |
| Chalcedony | -0.42 | -3.97 | -3.55 | SiO2 | |
| Chrysotile | 7.73 | 39.93 | 32.20 | Mg3Si2O5(OH)4 | |
| Clinoenstatite | 0.65 | 11.99 | 11.34 | MgSiO3 | |
| CO2(g) | -4.65 | -6.12 | -1.47 | CO2 | |
| Cristobalite | -0.38 | -3.97 | -3.59 | SiO2 | |
| Diopside | 3.28 | 23.17 | 19.89 | CaMgSi2O6 | |
| Dolomite | 2.61 | -14.48 | -17.09 | CaMg(CO3)2 | |
| Dolomite(d) | 2.06 | -14.48 | -16.54 | CaMg(CO3)2 | |
| Epsomite | -2.43 | -4.57 | -2.14 | MgSO4:7H2O | |
| FCO3Apatite | | 25.83 | | -88.57 | -114.40 |
| Ca9.316Na0.36Mg0.144(PO4)4.8(CO3)1.2F2.48 | | | | | |
| Fluorapatite | 8.05 | -9.55 | -17.60 | Ca5(PO4)3F | |
| Fluorite | -1.38 | -11.98 | -10.60 | CaF2 | |
| Forsterite | -0.35 | 27.95 | 28.31 | Mg2SiO4 | |
| Gypsum | -0.76 | -5.34 | -4.58 | CaSO4:2H2O | |
| H2(g) | -25.88 | -29.03 | -3.15 | H2 | |
| H2O(g) | -1.52 | -0.01 | 1.51 | H2O | |
| Halite | -2.62 | -1.04 | 1.58 | NaCl | |
| Huntite | 1.82 | -28.15 | -29.97 | CaMg3(CO3)4 | |
| Hydromagnesite | -2.66 | -11.42 | -8.76 | Mg5(CO3)4(OH)2:4H2O | |
| Hydroxyapatite | 7.43 | 4.01 | -3.42 | Ca5(PO4)3OH | |
| Magadiite | -5.12 | -19.42 | -14.30 | NaSi7O13(OH)3:3H2O | |
| Magnesite | 1.19 | -6.84 | -8.03 | MgCO3 | |
| Mirabilite | -2.63 | -3.74 | -1.11 | Na2SO4:10H2O | |
| Nahcolite | -3.52 | -4.07 | -0.55 | NaHCO3 | |
| Natron | -4.75 | -6.06 | -1.31 | Na2CO3:10H2O | |
| Nesquehonite | -1.24 | -6.86 | -5.62 | MgCO3:3H2O | |
| O2(g) | -31.45 | -34.34 | -2.89 | O2 | |
| Portlandite | -7.65 | 15.15 | 22.80 | Ca(OH)2 | |
| Quartz | 0.01 | -3.97 | -3.98 | SiO2 | |
| Sepiolite | 4.23 | 19.99 | 15.76 | Mg2Si3O7.5OH:3H2O | |
| Sepiolite(d) | 1.33 | 19.99 | 18.66 | Mg2Si3O7.5OH:3H2O | |

| | |
|---|---|
| | Silicagel       -0.95     -3.97   -3.02   SiO2
SiO2(a)       -1.26     -3.97   -2.71   SiO2
SrF2       -5.46   -14.00   -8.54   SrF2
Strontianite   -0.39     -9.66   -9.27   SrCO3
Talc       10.60    32.00   21.40   Mg3Si4O10(OH)2
Thenardite   -3.49    -3.67   -0.18   Na2SO4
Thermonatrite  -6.12    -6.00    0.13   Na2CO3:H2O
Tremolite    21.77    78.35   56.57   Ca2Mg5Si8O22(OH)2
Trona      -9.28   -10.07   -0.80   NaHCO3:Na2CO3:2H2O

**For a gas, SI = log10(fugacity). Fugacity = pressure * phi / 1 atm.
   For ideal gases, phi = 1.
* * *
End of simulation.
* * ** * *
Reading input data for simulation 2.
* * ** * *
End of Run after 0.172 Seconds.
------------------------------ |
| Cyanobacteria 14 Day |    Input file: C:\Users\XUHENGCHAO\Desktop\PHreeQC\culture\C1-14.pqi
   Output file: C:\Users\XUHENGCHAO\Desktop\PHreeQC\culture\C1-14.pqo
Database   file:   C:\Program   Files\USGS\Phreeqc   Interactive   3.3.12-
12704\database\wateq4f.dat
* * *
Reading data base.
* * *
    SOLUTION_MASTER_SPECIES
    SOLUTION_SPECIES
    PHASES
    EXCHANGE_MASTER_SPECIES
    EXCHANGE_SPECIES
    SURFACE_MASTER_SPECIES
    SURFACE_SPECIES |

```
        RATES
        END
* * *
Reading input data for simulation 1.
* * *
        DATABASE    C:\Program    Files\USGS\Phreeqc    Interactive    3.3.12-
12704\database\wateq4f.dat
        SOLUTION 1
                temp        25
                pH          8
                pe          4
                redox       pe
                units       mmol/l
                density     1.02
                Alkalinity 1.28
                B           0.37
                Br          0.72
                C           0.48
                Ca          8.35
                Cl          483.51
                F           0.07
                K           8.04
                Mg          48.77
                N(5)        0.88
                Na          417.34
                P           0.04
                S(6)        24.99
                Si          0.11
                Sr          0.08
                water       1 # kg
* * *
Beginning of initial solution calculations.
* * *
Initial solution 1.

pH will be adjusted to obtain desired alkalinity.

----------------------------Solution composition----------------------------
```

| Elements | Molality | Moles |
|---|---|---|
| Alkalinity | 1.294e-03 | 1.294e-03 |
| B | 3.742e-04 | 3.742e-04 |
| Br | 7.281e-04 | 7.281e-04 |
| C | 4.854e-04 | 4.854e-04 |
| Ca | 8.444e-03 | 8.444e-03 |
| Cl | 4.890e-01 | 4.890e-01 |
| F | 7.079e-05 | 7.079e-05 |
| K | 8.131e-03 | 8.131e-03 |
| Mg | 4.932e-02 | 4.932e-02 |
| N(5) | 8.899e-04 | 8.899e-04 |
| Na | 4.220e-01 | 4.220e-01 |
| P | 4.045e-05 | 4.045e-05 |
| S(6) | 2.527e-02 | 2.527e-02 |
| Si | 1.112e-04 | 1.112e-04 |
| Sr | 8.090e-05 | 8.090e-05 |

----------------------------Description of solution----------------------------

$$pH = 9.307 \quad \text{Adjust alkalinity}$$

$$pe = 4.000$$

Activity of water = 0.983

Ionic strength (mol/kgw) = 5.872e-01

Mass of water (kg) = 1.000e+00

Total CO2 (mol/kg) = 4.854e-04

Temperature (癈) = 25.00

Electrical balance (eq) = 3.337e-03

Percent error, 100*(Cat-|An|)/(Cat+|An|) = 0.32

Iterations = 9

Total H = 1.110141e+02

Total O = 5.561332e+01

----------------------------Distribution of species----------------------------

| Species | Log Molality | Log Activity | Log Molality | Log Activity | Gamma | mole V cm?mol |
|---|---|---|---|---|---|---|

| | | | | | |
|---|---|---|---|---|---|
| OH- | 3.223e-05 | 1.998e-05 | -4.492 | -4.700 | -0.208 (0) |
| H+ | 6.490e-10 | 4.927e-10 | -9.188 | -9.307 | -0.120 0.00 |
| H2O | 5.551e+01 | 9.831e-01 | 1.744 | -0.007 | 0.000 18.07 |
| B | 3.742e-04 | | | | |
| H2BO3- | 2.616e-04 | 1.506e-04 | -3.582 | -3.822 | -0.240 (0) |
| H3BO3 | 1.126e-04 | 1.289e-04 | -3.948 | -3.890 | 0.059 (0) |
| BF(OH)3- | 2.080e-09 | 1.197e-09 | -8.682 | -8.922 | -0.240 (0) |
| BF2(OH)2- | 2.607e-15 | 1.500e-15 | -14.584 | -14.824 | -0.240 (0) |
| BF3OH- | 3.342e-23 | 1.924e-23 | -22.476 | -22.716 | -0.240 (0) |
| BF4- | 1.592e-30 | 9.165e-31 | -29.798 | -30.038 | -0.240 (0) |
| Br | 7.281e-04 | | | | |
| Br- | 7.281e-04 | 5.380e-04 | -3.138 | -3.269 | -0.131 (0) |
| C(-4) | 0.000e+00 | | | | |
| CH4 | 0.000e+00 | 0.000e+00 | -89.011 | -88.952 | 0.059 (0) |
| C(4) | 4.854e-04 | | | | |
| HCO3- | 1.643e-04 | 1.122e-04 | -3.784 | -3.950 | -0.166 (0) |
| MgCO3 | 1.102e-04 | 1.262e-04 | -3.958 | -3.899 | 0.059 (0) |
| NaCO3- | 8.575e-05 | 5.856e-05 | -4.067 | -4.232 | -0.166 (0) |
| CO3-2 | 4.911e-05 | 1.068e-05 | -4.309 | -4.971 | -0.663 (0) |
| CaCO3 | 3.008e-05 | 3.444e-05 | -4.522 | -4.463 | 0.059 (0) |
| MgHCO3+ | 2.546e-05 | 1.626e-05 | -4.594 | -4.789 | -0.195 (0) |
| NaHCO3 | 1.623e-05 | 1.858e-05 | -4.790 | -4.731 | 0.059 (0) |

|       | CaHCO3+ | 3.942e-06 | 2.755e-06 | -5.404 | -5.560 | -0.156 | (0) |

CaHCO3+          3.942e-06   2.755e-06    -5.404    -5.560    -0.156    (0)
   SrCO3          1.247e-07   1.247e-07    -6.904    -6.904     0.000    (0)
   CO2            1.104e-07   1.264e-07    -6.957    -6.898     0.059    (0)
   SrHCO3+        4.596e-08   3.139e-08    -7.338    -7.503    -0.166    (0)
Ca            8.444e-03
   Ca+2           7.580e-03   1.925e-03    -2.120    -2.716    -0.595    (0)
   CaSO4          8.245e-04   9.438e-04    -3.084    -3.025     0.059    (0)
   CaCO3          3.008e-05   3.444e-05    -4.522    -4.463     0.059    (0)
   CaHCO3+        3.942e-06   2.755e-06    -5.404    -5.560    -0.156    (0)
   CaPO4-         3.467e-06   2.367e-06    -5.460    -5.626    -0.166    (0)
   CaOH+          9.123e-07   6.374e-07    -6.040    -6.196    -0.156    (0)
   CaF+           5.706e-07   3.912e-07    -6.244    -6.408    -0.164    (0)
   CaHPO4         4.306e-07   4.930e-07    -6.366    -6.307     0.059    (0)
   CaH2PO4+       2.673e-10   1.826e-10    -9.573    -9.739    -0.166    (0)
   CaHSO4+        3.687e-12   2.724e-12   -11.433   -11.565    -0.131    (0)
Cl            4.890e-01
   Cl-            4.890e-01   3.092e-01    -0.311    -0.510    -0.199    (0)
F             7.079e-05
   F-             3.765e-05   2.333e-05    -4.424    -4.632    -0.208    (0)
   MgF+           2.911e-05   1.909e-05    -4.536    -4.719    -0.183    (0)
   NaF            3.454e-06   3.954e-06    -5.462    -5.403     0.059    (0)
   CaF+           5.706e-07   3.912e-07    -6.244    -6.408    -0.164    (0)

| | | | | | |
|---|---|---|---|---|---|
| BF(OH)3- | 2.080e-09 | 1.197e-09 | -8.682 | -8.922 | -0.240 (0) |
| HF | 1.506e-11 | 1.724e-11 | -10.822 | -10.763 | 0.059 (0) |
| BF2(OH)2- | 2.607e-15 | 1.500e-15 | -14.584 | -14.824 | -0.240 (0) |
| HF2- | 2.491e-15 | 1.544e-15 | -14.604 | -14.811 | -0.208 (0) |
| H2F2 | 6.767e-22 | 7.747e-22 | -21.170 | -21.111 | 0.059 (0) |
| BF3OH- | 3.342e-23 | 1.924e-23 | -22.476 | -22.716 | -0.240 (0) |
| BF4- | 1.592e-30 | 9.165e-31 | -29.798 | -30.038 | -0.240 (0) |
| SiF6-2 | 6.285e-39 | 1.277e-39 | -38.202 | -38.894 | -0.692 (0) |
| H(0) | 3.003e-30 | | | | |
| H2 | 1.501e-30 | 1.719e-30 | -29.824 | -29.765 | 0.059 (0) |
| K | 8.131e-03 | | | | |
| K+ | 8.003e-03 | 5.061e-03 | -2.097 | -2.296 | -0.199 (0) |
| KSO4- | 1.280e-04 | 8.740e-05 | -3.893 | -4.059 | -0.166 (0) |
| KHPO4- | 6.751e-09 | 4.610e-09 | -8.171 | -8.336 | -0.166 (0) |
| Mg | 4.932e-02 | | | | |
| Mg+2 | 4.276e-02 | 1.238e-02 | -1.369 | -1.907 | -0.538 (0) |
| MgSO4 | 6.232e-03 | 7.135e-03 | -2.205 | -2.147 | 0.059 (0) |
| MgOH+ | 1.262e-04 | 8.972e-05 | -3.899 | -4.047 | -0.148 (0) |
| MgCO3 | 1.102e-04 | 1.262e-04 | -3.958 | -3.899 | 0.059 (0) |
| MgPO4- | 3.009e-05 | 2.055e-05 | -4.522 | -4.687 | -0.166 (0) |
| MgF+ | 2.911e-05 | 1.909e-05 | -4.536 | -4.719 | -0.183 (0) |
| MgHCO3+ | 2.546e-05 | 1.626e-05 | -4.594 | -4.789 | -0.195 (0) |

| | | Molality | Activity | Log Molality | Log Activity | Log Gamma | |
|---|---|---|---|---|---|---|---|
| | MgHPO4 | 3.746e-06 | 4.288e-06 | -5.426 | -5.368 | 0.059 | (0) |
| | MgH2PO4+ | 2.191e-09 | 1.496e-09 | -8.659 | -8.825 | -0.166 | (0) |
| N(5) | 8.899e-04 | | | | | | |
| | NO3- | 8.899e-04 | 5.329e-04 | -3.051 | -3.273 | -0.223 | (0) |
| Na | 4.220e-01 | | | | | | |
| | Na+ | 4.166e-01 | 2.945e-01 | -0.380 | -0.531 | -0.151 | (0) |
| | NaSO4- | 5.311e-03 | 3.627e-03 | -2.275 | -2.440 | -0.166 | (0) |
| | NaCO3- | 8.575e-05 | 5.856e-05 | -4.067 | -4.232 | -0.166 | (0) |
| | NaHCO3 | 1.623e-05 | 1.858e-05 | -4.790 | -4.731 | 0.059 | (0) |
| | NaF | 3.454e-06 | 3.954e-06 | -5.462 | -5.403 | 0.059 | (0) |
| | NaHPO4- | 3.928e-07 | 2.682e-07 | -6.406 | -6.572 | -0.166 | (0) |
| O(0) | 2.383e-33 | | | | | | |
| | O2 | 1.192e-33 | 1.364e-33 | -32.924 | -32.865 | 0.059 | (0) |
| P | 4.045e-05 | | | | | | |
| | MgPO4- | 3.009e-05 | 2.055e-05 | -4.522 | -4.687 | -0.166 | (0) |
| | MgHPO4 | 3.746e-06 | 4.288e-06 | -5.426 | -5.368 | 0.059 | (0) |
| | CaPO4- | 3.467e-06 | 2.367e-06 | -5.460 | -5.626 | -0.166 | (0) |
| | HPO4-2 | 2.299e-06 | 4.671e-07 | -5.638 | -6.331 | -0.692 | (0) |
| | CaHPO4 | 4.306e-07 | 4.930e-07 | -6.366 | -6.307 | 0.059 | (0) |
| | NaHPO4- | 3.928e-07 | 2.682e-07 | -6.406 | -6.572 | -0.166 | (0) |
| | PO4-3 | 1.542e-08 | 4.274e-10 | -7.812 | -9.369 | -1.557 | (0) |
| | KHPO4- | 6.751e-09 | 4.610e-09 | -8.171 | -8.336 | -0.166 | (0) |
| | H2PO4- | 5.428e-09 | 3.707e-09 | -8.265 | -8.431 | - | |

```
                                         0.166     (0)
       MgH2PO4+              2.191e-09    1.496e-09    -8.659    -8.825    -
0.166        (0)
       CaH2PO4+             2.673e-10    1.826e-10    -9.573    -9.739    -
0.166        (0)
S(6)          2.527e-02
       SO4-2                1.277e-02    2.457e-03    -1.894    -2.610    -
0.716        (0)
       MgSO4                6.232e-03    7.135e-03    -2.205    -2.147
0.059        (0)
       NaSO4-               5.311e-03    3.627e-03    -2.275    -2.440    -
0.166        (0)
       CaSO4                8.245e-04    9.438e-04    -3.084    -3.025
0.059        (0)
       KSO4-                1.280e-04    8.740e-05    -3.893    -4.059    -
0.166        (0)
       SrSO4                7.654e-06    8.763e-06    -5.116    -5.057
0.059        (0)
       HSO4-                1.795e-10    1.177e-10    -9.746    -9.929    -
0.183        (0)
       CaHSO4+              3.687e-12    2.724e-12    -11.433   -11.565   -
0.131        (0)
Si            1.112e-04
       H4SiO4               7.239e-05    8.287e-05    -4.140    -4.082
0.059        (0)
       H3SiO4-              3.883e-05    2.480e-05    -4.411    -4.606    -
0.195        (0)
       H2SiO4-2             1.575e-08    3.424e-09    -7.803    -8.465    -
0.663        (0)
       SiF6-2               6.285e-39    1.277e-39    -38.202   -38.894   -0.692
(0)
Sr            8.090e-05
       Sr+2                 7.307e-05    1.829e-05    -4.136    -4.738    -
0.602        (0)
       SrSO4                7.654e-06    8.763e-06    -5.116    -5.057
0.059        (0)
       SrCO3                1.247e-07    1.247e-07    -6.904    -6.904
0.000        (0)
       SrHCO3+              4.596e-08    3.139e-08    -7.338    -7.503    -
0.166        (0)
       SrOH+                2.787e-09    1.871e-09    -8.555    -8.728    -
```

0.173        (0)

----------------------------Saturation indices----------------------------

Phase                    SI** log IAP     log K(298 K,     1 atm)

Anhydrite          -0.96      -5.33    -4.36   CaSO4
Aragonite           0.65      -7.69    -8.34   CaCO3
Artinite            0.19       9.79     9.60   MgCO3:Mg(OH)2:3H2O
Brucite            -0.15      16.69    16.84   Mg(OH)2
Calcite             0.79      -7.69    -8.48   CaCO3
Celestite          -0.72      -7.35    -6.63   SrSO4
CH4(g)            -86.09     -88.95    -2.86   CH4
Chalcedony         -0.52      -4.07    -3.55   SiO2
Chrysotile          9.75      41.95    32.20   Mg3Si2O5(OH)4
Clinoenstatite      1.29      12.63    11.34   MgSiO3
CO2(g)             -5.43      -6.90    -1.47   CO2
Cristobalite       -0.48      -4.07    -3.59   SiO2
Diopside            4.56      24.46    19.89   CaMgSi2O6
Dolomite            2.52     -14.57   -17.09   CaMg(CO3)2
Dolomite(d)         1.97     -14.57   -16.54   CaMg(CO3)2
Epsomite           -2.43      -4.57    -2.14   MgSO4:7H2O
FCO3Apatite                   26.21             -88.19      -114.40
Ca9.316Na0.36Mg0.144(PO4)4.8(CO3)1.2F2.48
Fluorapatite        8.32      -9.28   -17.60   Ca5(PO4)3F
Fluorite           -1.38     -11.98   -10.60   CaF2
Forsterite          1.03      29.33    28.31   Mg2SiO4
Gypsum             -0.76      -5.34    -4.58   CaSO4:2H2O
H2(g)             -26.61     -29.76    -3.15   H2
H2O(g)             -1.52      -0.01     1.51   H2O
Halite             -2.62      -1.04     1.58   NaCl
Huntite             1.65     -28.32   -29.97   CaMg3(CO3)4
Hydromagnesite     -2.09     -10.85    -8.76   Mg5(CO3)4(OH)2:4H2O
Hydroxyapatite      8.07       4.65    -3.42   Ca5(PO4)3OH
Magadiite          -5.43     -19.73   -14.30   NaSi7O13(OH)3:3H2O
Magnesite           1.15      -6.88    -8.03   MgCO3
Mirabilite         -2.63      -3.75    -1.11   Na2SO4:10H2O
Nahcolite          -3.93      -4.48    -0.55   NaHCO3
Natron             -4.80      -6.11    -1.31   Na2CO3:10H2O
Nesquehonite       -1.28      -6.90    -5.62   MgCO3:3H2O
O2(g)             -29.97     -32.87    -2.89   O2

| | |
|---|---|
| | Portlandite     -6.92     15.88    22.80   Ca(OH)2 |
| | Quartz         -0.09     -4.07    -3.98   SiO2 |
| | Sepiolite      5.41     21.17    15.76   Mg2Si3O7.5OH:3H2O |
| | Sepiolite(d)    2.51     21.17    18.66   Mg2Si3O7.5OH:3H2O |
| | Silicagel     -1.05     -4.07    -3.02   SiO2 |
| | SiO2(a)      -1.36     -4.07    -2.71   SiO2 |
| | SrF2         -5.46    -14.00   -8.54   SrF2 |
| | Strontianite   -0.44     -9.71    -9.27   SrCO3 |
| | Talc         12.43     33.83    21.40   Mg3Si4O10(OH)2 |
| | Thenardite   -3.49     -3.67    -0.18   Na2SO4 |
| | Thermonatrite  -6.17     -6.04    0.13   Na2CO3:H2O |
| | Tremolite     26.17    82.74    56.57   Ca2Mg5Si8O22(OH)2 |
| | Trona        -9.73    -10.53    -0.80   NaHCO3:Na2CO3:2H2O |
| | **For a gas, SI = log10(fugacity). Fugacity = pressure * phi / 1 atm.
For ideal gases, phi = 1. |
| | ------------------
End of simulation.
------------------ |
| | ----------------------------------
Reading input data for simulation 2.
---------------------------------- |
| | ------------------------------
End of Run after 0.156 Seconds.
------------------------------ |
| Cyanobacteria 16 Day | Input file: C:\Users\XUHENGCHAO\Desktop\PHreeQC\culture\C1-15.pqi
Output file: C:\Users\XUHENGCHAO\Desktop\PHreeQC\culture\C1-15.pqo
Database   file:   C:\Program   Files\USGS\Phreeqc   Interactive   3.3.12-12704\database\wateq4f.dat
* * *
Reading data base.
* * *
    SOLUTION_MASTER_SPECIES
    SOLUTION_SPECIES
    PHASES |

```
        EXCHANGE_MASTER_SPECIES
        EXCHANGE_SPECIES
        SURFACE_MASTER_SPECIES
        SURFACE_SPECIES
        RATES
        END
* * *
Reading input data for simulation 1.
* * *
        DATABASE    C:\Program    Files\USGS\Phreeqc    Interactive    3.3.12-
12704\database\wateq4f.dat
        SOLUTION 1
                temp        25
                pH          8
                pe          4
                redox       pe
                units       mmol/l
                density     1.02
                Alkalinity 1.43
                B           0.37
                Br          0.72
                C           0.59
                Ca          8.37
                Cl          483.51
                F           0.07
                K           8.04
                Mg          48.84
                N(5)        0.88
                Na          417.34
                P           0.04
                S(6)        24.99
                Si          0.11
                Sr          0.08
                water       1 # kg
* * *
Beginning of initial solution calculations.
* * *
Initial solution 1.
```

pH will be adjusted to obtain desired alkalinity.

----------------------------Solution composition----------------------------

| Elements | Molality | Moles |
|----------|----------|-------|
| Alkalinity | 1.446e-03 | 1.446e-03 |
| B | 3.742e-04 | 3.742e-04 |
| Br | 7.281e-04 | 7.281e-04 |
| C | 5.967e-04 | 5.967e-04 |
| Ca | 8.465e-03 | 8.465e-03 |
| Cl | 4.890e-01 | 4.890e-01 |
| F | 7.079e-05 | 7.079e-05 |
| K | 8.131e-03 | 8.131e-03 |
| Mg | 4.939e-02 | 4.939e-02 |
| N(5) | 8.899e-04 | 8.899e-04 |
| Na | 4.221e-01 | 4.221e-01 |
| P | 4.045e-05 | 4.045e-05 |
| S(6) | 2.527e-02 | 2.527e-02 |
| Si | 1.112e-04 | 1.112e-04 |
| Sr | 8.090e-05 | 8.090e-05 |

--------------------------Description of solution--------------------------

                                    pH    =    9.283          Adjust
alkalinity

                                    pe    =    4.000
                        Activity of water    =    0.983
                Ionic strength (mol/kgw)    =    5.874e-01
                    Mass of water (kg)    =    1.000e+00
                    Total CO2 (mol/kg)    =    5.967e-04
                    Temperature (癈)    =    25.00
                Electrical balance (eq)    =    3.368e-03
    Percent error, 100*(Cat-|An|)/(Cat+|An|)    =    0.32
                            Iterations    =    9
                            Total H    = 1.110141e+02
                            Total O    = 5.561365e+01

--------------------------Distribution of species--------------------------

|  |  |  |  | Log | Log | Log | mole V |
| Species | Molality | Activity | Molality | Activity | Gamma | cm?mol |
| --- | --- | --- | --- | --- | --- | --- |
| OH- | 3.050e-05 | 1.890e-05 | -4.516 | -4.724 | -0.208 | (0) |
| H+ | 6.859e-10 | 5.207e-10 | -9.164 | -9.283 | -0.120 | 0.00 |
| H2O | 5.551e+01 | 9.831e-01 | 1.744 | -0.007 | 0.000 | 18.07 |
| B | 3.742e-04 |  |  |  |  |  |
| H2BO3- | 2.572e-04 | 1.480e-04 | -3.590 | -3.830 | -0.240 | (0) |
| H3BO3 | 1.170e-04 | 1.339e-04 | -3.932 | -3.873 | 0.059 | (0) |
| BF(OH)3- | 2.161e-09 | 1.244e-09 | -8.665 | -8.905 | -0.240 | (0) |
| BF2(OH)2- | 2.860e-15 | 1.646e-15 | -14.544 | -14.784 | -0.240 | (0) |
| BF3OH- | 3.874e-23 | 2.230e-23 | -22.412 | -22.652 | -0.240 | (0) |
| BF4- | 1.949e-30 | 1.122e-30 | -29.710 | -29.950 | -0.240 | (0) |
| Br | 7.281e-04 |  |  |  |  |  |
| Br- | 7.281e-04 | 5.380e-04 | -3.138 | -3.269 | -0.131 | (0) |
| C(-4) | 0.000e+00 |  |  |  |  |  |
| CH4 | 0.000e+00 | 0.000e+00 | -88.692 | -88.633 | 0.059 | (0) |
| C(4) | 5.967e-04 |  |  |  |  |  |
| HCO3- | 2.082e-04 | 1.422e-04 | -3.681 | -3.847 | -0.166 | (0) |
| MgCO3 | 1.323e-04 | 1.515e-04 | -3.878 | -3.820 | 0.059 | (0) |
| NaCO3- | 1.028e-04 | 7.022e-05 | -3.988 | -4.154 | -0.166 | (0) |
| CO3-2 | 5.890e-05 | 1.281e-05 | -4.230 | -4.893 | -0.663 | (0) |
| CaCO3 | 3.613e-05 | 4.136e-05 | -4.442 | -4.383 | 0.059 | (0) |

```
            MgHCO3+              3.230e-05    2.063e-05      -4.491      -4.686      -
0.195      (0)
            NaHCO3                2.057e-05    2.355e-05      -4.687      -4.628
0.059      (0)
            CaHCO3+              5.004e-06    3.497e-06      -5.301      -5.456      -
0.156      (0)
            SrCO3                 1.494e-07    1.494e-07      -6.826      -6.826
0.000      (0)
            CO2                   1.479e-07    1.693e-07      -6.830      -6.771
0.059      (0)
            SrHCO3+              5.822e-08    3.976e-08      -7.235      -7.401      -
0.166      (0)
Ca              8.465e-03
            Ca+2                  7.593e-03    1.928e-03      -2.120      -2.715      -
0.595      (0)
            CaSO4                 8.255e-04    9.450e-04      -3.083      -3.025
0.059      (0)
            CaCO3                 3.613e-05    4.136e-05      -4.442      -4.383
0.059      (0)
            CaHCO3+              5.004e-06    3.497e-06      -5.301      -5.456      -
0.156      (0)
            CaPO4-                3.435e-06    2.346e-06      -5.464      -5.630      -
0.166      (0)
            CaOH+                 8.646e-07    6.041e-07      -6.063      -6.219      -
0.156      (0)
            CaF+                  5.712e-07    3.916e-07      -6.243      -6.407      -
0.164      (0)
            CaHPO4                4.510e-07    5.163e-07      -6.346      -6.287
0.059      (0)
            CaH2PO4+            2.959e-10    2.021e-10      -9.529      -9.695      -
0.166      (0)
            CaHSO4+              3.901e-12    2.883e-12     -11.409     -11.540      -
0.131      (0)
Cl              4.890e-01
            Cl-                   4.890e-01    3.092e-01      -0.311      -0.510      -0.199
(0)
F               7.079e-05
            F-                    3.763e-05    2.332e-05      -4.424      -4.632      -
0.208      (0)
            MgF+                  2.913e-05    1.910e-05      -4.536      -4.719      -
0.183      (0)
```

| | | | | | |
|---|---|---|---|---|---|
| NaF | 3.452e-06 | 3.952e-06 | -5.462 | -5.403 | 0.059 (0) |
| CaF+ | 5.712e-07 | 3.916e-07 | -6.243 | -6.407 | -0.164 (0) |
| BF(OH)3- | 2.161e-09 | 1.244e-09 | -8.665 | -8.905 | -0.240 (0) |
| HF | 1.591e-11 | 1.821e-11 | -10.798 | -10.740 | 0.059 (0) |
| BF2(OH)2- | 2.860e-15 | 1.646e-15 | -14.544 | -14.784 | -0.240 (0) |
| HF2- | 2.630e-15 | 1.630e-15 | -14.580 | -14.788 | -0.208 (0) |
| H2F2 | 7.551e-22 | 8.645e-22 | -21.122 | -21.063 | 0.059 (0) |
| BF3OH- | 3.874e-23 | 2.230e-23 | -22.412 | -22.652 | -0.240 (0) |
| BF4- | 1.949e-30 | 1.122e-30 | -29.710 | -29.950 | -0.240 (0) |
| SiF6-2 | 7.970e-39 | 1.619e-39 | -38.099 | -38.791 | -0.692 (0) |
| H(0) | 3.353e-30 | | | | |
| H2 | 1.677e-30 | 1.919e-30 | -29.776 | -29.717 | 0.059 (0) |
| K | 8.131e-03 | | | | |
| K+ | 8.003e-03 | 5.061e-03 | -2.097 | -2.296 | -0.199 (0) |
| KSO4- | 1.279e-04 | 8.737e-05 | -3.893 | -4.059 | -0.166 (0) |
| KHPO4- | 7.059e-09 | 4.820e-09 | -8.151 | -8.317 | -0.166 (0) |
| Mg | 4.939e-02 | | | | |
| Mg+2 | 4.281e-02 | 1.240e-02 | -1.368 | -1.907 | -0.538 (0) |
| MgSO4 | 6.236e-03 | 7.140e-03 | -2.205 | -2.146 | 0.059 (0) |
| MgCO3 | 1.323e-04 | 1.515e-04 | -3.878 | -3.820 | 0.059 (0) |
| MgOH+ | 1.196e-04 | 8.499e-05 | -3.922 | -4.071 | -0.148 (0) |
| MgHCO3+ | 3.230e-05 | 2.063e-05 | -4.491 | -4.686 | -0.195 (0) |

| Species | Molality | Activity | Log Molality | Log Activity | Log Gamma | |
|---|---|---|---|---|---|---|
| MgPO4- | 2.980e-05 | 2.035e-05 | -4.526 | -4.691 | -0.166 | (0) |
| MgF+ | 2.913e-05 | 1.910e-05 | -4.536 | -4.719 | -0.183 | (0) |
| MgHPO4 | 3.921e-06 | 4.489e-06 | -5.407 | -5.348 | 0.059 | (0) |
| MgH2PO4+ | 2.423e-09 | 1.655e-09 | -8.616 | -8.781 | -0.166 | (0) |
| N(5) | 8.899e-04 | | | | | |
| NO3- | 8.899e-04 | 5.329e-04 | -3.051 | -3.273 | -0.223 | (0) |
| Na | 4.221e-01 | | | | | |
| Na+ | 4.166e-01 | 2.945e-01 | -0.380 | -0.531 | -0.151 | (0) |
| NaSO4- | 5.309e-03 | 3.626e-03 | -2.275 | -2.441 | -0.166 | (0) |
| NaCO3- | 1.028e-04 | 7.022e-05 | -3.988 | -4.154 | -0.166 | (0) |
| NaHCO3 | 2.057e-05 | 2.355e-05 | -4.687 | -4.628 | 0.059 | (0) |
| NaF | 3.452e-06 | 3.952e-06 | -5.462 | -5.403 | 0.059 | (0) |
| NaHPO4- | 4.107e-07 | 2.804e-07 | -6.387 | -6.552 | -0.166 | (0) |
| O(0) | 1.910e-33 | | | | | |
| O2 | 9.552e-34 | 1.094e-33 | -33.020 | -32.961 | 0.059 | (0) |
| P | 4.045e-05 | | | | | |
| MgPO4- | 2.980e-05 | 2.035e-05 | -4.526 | -4.691 | -0.166 | (0) |
| MgHPO4 | 3.921e-06 | 4.489e-06 | -5.407 | -5.348 | 0.059 | (0) |
| CaPO4- | 3.435e-06 | 2.346e-06 | -5.464 | -5.630 | -0.166 | (0) |
| HPO4-2 | 2.404e-06 | 4.884e-07 | -5.619 | -6.311 | -0.692 | (0) |
| CaHPO4 | 4.510e-07 | 5.163e-07 | -6.346 | -6.287 | 0.059 | (0) |
| NaHPO4- | 4.107e-07 | 2.804e-07 | -6.387 | -6.552 | -0.166 | (0) |
| PO4-3 | 1.526e-08 | 4.229e-10 | -7.816 | -9.374 | - | |

1.557        (0)
   KHPO4-              7.059e-09    4.820e-09    -8.151    -8.317    -
0.166        (0)
   H2PO4-              5.999e-09    4.096e-09    -8.222    -8.388    -
0.166        (0)
   MgH2PO4+            2.423e-09    1.655e-09    -8.616    -8.781    -
0.166        (0)
   CaH2PO4+            2.959e-10    2.021e-10    -9.529    -9.695    -
0.166        (0)
S(6)          2.527e-02
   SO4-2               1.277e-02    2.457e-03    -1.894    -2.610    -
0.716        (0)
   MgSO4               6.236e-03    7.140e-03    -2.205    -2.146
0.059        (0)
   NaSO4-              5.309e-03    3.626e-03    -2.275    -2.441    -
0.166        (0)
   CaSO4               8.255e-04    9.450e-04    -3.083    -3.025
0.059        (0)
   KSO4-               1.279e-04    8.737e-05    -3.893    -4.059    -
0.166        (0)
   SrSO4               7.648e-06    8.756e-06    -5.116    -5.058
0.059        (0)
   HSO4-               1.897e-10    1.244e-10    -9.722    -9.905    -
0.183        (0)
   CaHSO4+             3.901e-12    2.883e-12    -11.409   -11.540   -
0.131        (0)
Si            1.112e-04
   H4SiO4              7.378e-05    8.446e-05    -4.132    -4.073
0.059        (0)
   H3SiO4-             3.745e-05    2.392e-05    -4.427    -4.621    -
0.195        (0)
   H2SiO4-2            1.437e-08    3.124e-09    -7.843    -8.505    -
0.663        (0)
   SiF6-2              7.970e-39    1.619e-39    -38.099   -38.791   -0.692
(0)
Sr            8.090e-05
   Sr+2                7.305e-05    1.828e-05    -4.136    -4.738    -
0.602        (0)
   SrSO4               7.648e-06    8.756e-06    -5.116    -5.058
0.059        (0)
   SrCO3               1.494e-07    1.494e-07    -6.826    -6.826

```
   0.000         (0)
     SrHCO3+                5.822e-08    3.976e-08      -7.235      -7.401        -
0.166         (0)
     SrOH+                  2.636e-09    1.770e-09      -8.579      -8.752        -
0.173         (0)

----------------------------Saturation indices----------------------------

   Phase                    SI** log IAP      log K(298 K,     1 atm)

   Anhydrite              -0.96       -5.32      -4.36   CaSO4
   Aragonite               0.73       -7.61      -8.34   CaCO3
   Artinite                0.22        9.82       9.60   MgCO3:Mg(OH)2:3H2O
   Brucite                -0.19       16.65      16.84   Mg(OH)2
   Calcite                 0.87       -7.61      -8.48   CaCO3
   Celestite              -0.72       -7.35      -6.63   SrSO4
   CH4(g)                -85.77      -88.63      -2.86   CH4
   Chalcedony             -0.51       -4.06      -3.55   SiO2
   Chrysotile              9.63       41.83      32.20   Mg3Si2O5(OH)4
   Clinoenstatite          1.25       12.59      11.34   MgSiO3
   CO2(g)                 -5.30       -6.77      -1.47   CO2
   Cristobalite           -0.47       -4.06      -3.59   SiO2
   Diopside                4.49       24.38      19.89   CaMgSi2O6
   Dolomite                2.68      -14.41     -17.09   CaMg(CO3)2
   Dolomite(d)             2.13      -14.41     -16.54   CaMg(CO3)2
   Epsomite               -2.43       -4.57      -2.14   MgSO4:7H2O
   FCO3Apatite                        26.29              -88.11     -114.40
Ca9.316Na0.36Mg0.144(PO4)4.8(CO3)1.2F2.48
   Fluorapatite            8.31       -9.29     -17.60   Ca5(PO4)3F
   Fluorite               -1.38      -11.98     -10.60   CaF2
   Forsterite              0.94       29.25      28.31   Mg2SiO4
   Gypsum                 -0.76       -5.34      -4.58   CaSO4:2H2O
   H2(g)                 -26.57      -29.72      -3.15   H2
   H2O(g)                 -1.52       -0.01       1.51   H2O
   Halite                 -2.62       -1.04       1.58   NaCl
   Huntite                 1.96      -28.01     -29.97   CaMg3(CO3)4
   Hydromagnesite         -1.82      -10.58      -8.76   Mg5(CO3)4(OH)2:4H2O
   Hydroxyapatite          8.04        4.62      -3.42   Ca5(PO4)3OH
   Magadiite              -5.39      -19.69     -14.30   NaSi7O13(OH)3:3H2O
   Magnesite               1.23       -6.80      -8.03   MgCO3
   Mirabilite             -2.63       -3.75      -1.11   Na2SO4:10H2O
```

| | |
|---|---|
| | Nahcolite      -3.83      -4.38      -0.55     NaHCO3
 Natron      -4.72      -6.03      -1.31     Na2CO3:10H2O
 Nesquehonite     -1.20      -6.82      -5.62     MgCO3:3H2O
 O2(g)      -30.07     -32.96      -2.89     O2
 Portlandite     -6.96      15.84      22.80     Ca(OH)2
 Quartz      -0.08      -4.06      -3.98     SiO2
 Sepiolite      5.34      21.10      15.76     Mg2Si3O7.5OH:3H2O
 Sepiolite(d)     2.44      21.10      18.66     Mg2Si3O7.5OH:3H2O
 Silicagel      -1.04      -4.06      -3.02     SiO2
 SiO2(a)      -1.35      -4.06      -2.71     SiO2
 SrF2      -5.46      -14.00      -8.54     SrF2
 Strontianite    -0.36      -9.63      -9.27     SrCO3
 Talc      12.32      33.72      21.40     Mg3Si4O10(OH)2
 Thenardite     -3.49      -3.67      -0.18     Na2SO4
 Thermonatrite    -6.09      -5.96      0.13     Na2CO3:H2O
 Tremolite     25.90      82.48      56.57     Ca2Mg5Si8O22(OH)2
 Trona      -9.55      -10.35      -0.80     NaHCO3:Na2CO3:2H2O

 **For a gas, SI = log10(fugacity). Fugacity = pressure * phi / 1 atm.
    For ideal gases, phi = 1.
* * *
 End of simulation.
* * ** * *
 Reading input data for simulation 2.
* * ** * *
 End of Run after 0.156 Seconds.
 ------------------------------ |
| Cyanobacteria+
 virus
 6 Day |    Input file: C:\Users\XUHENGCHAO\Desktop\PHreeQC\culture\CP1-6.pqi
    Output file: C:\Users\XUHENGCHAO\Desktop\PHreeQC\culture\CP1-6.pqo
 Database    file:    C:\Program    Files\USGS\Phreeqc    Interactive    3.3.12-
 12704\database\wateq4f.dat
* * *
 Reading data base.
 ------------------ |

```
        SOLUTION_MASTER_SPECIES
        SOLUTION_SPECIES
        PHASES
        EXCHANGE_MASTER_SPECIES
        EXCHANGE_SPECIES
        SURFACE_MASTER_SPECIES
        SURFACE_SPECIES
        RATES
        END
* * *
Reading input data for simulation 1.
* * *
        DATABASE    C:\Program    Files\USGS\Phreeqc    Interactive    3.3.12-
12704\database\wateq4f.dat
        SOLUTION 1
                temp        25
                pH          8
                pe          4
                redox       pe
                units       mmol/l
                density     1.02
                Alkalinity 4.14
                B           0.37
                Br          0.72
                C           1.93
                Ca          8.51
                Cl          483.51
                F           0.07
                K           8.04
                Mg          50.64
                N(5)        0.88
                Na          417.34
                P           0.04
                S(6)        24.99
                Si          0.11
                Sr          0.08
                water       1 # kg
* * *
Beginning of initial solution calculations.
```
* * *
Initial solution 1.

pH will be adjusted to obtain desired alkalinity.

----------------------------Solution composition----------------------------

| Elements | Molality | Moles |
|---|---|---|
| Alkalinity | 4.188e-03 | 4.188e-03 |
| B | 3.743e-04 | 3.743e-04 |
| Br | 7.283e-04 | 7.283e-04 |
| C | 1.952e-03 | 1.952e-03 |
| Ca | 8.608e-03 | 8.608e-03 |
| Cl | 4.891e-01 | 4.891e-01 |
| F | 7.081e-05 | 7.081e-05 |
| K | 8.133e-03 | 8.133e-03 |
| Mg | 5.123e-02 | 5.123e-02 |
| N(5) | 8.902e-04 | 8.902e-04 |
| Na | 4.222e-01 | 4.222e-01 |
| P | 4.046e-05 | 4.046e-05 |
| S(6) | 2.528e-02 | 2.528e-02 |
| Si | 1.113e-04 | 1.113e-04 |
| Sr | 8.093e-05 | 8.093e-05 |

----------------------------Description of solution----------------------------

pH = 9.636 Adjust alkalinity

pe = 4.000

Activity of water = 0.983

Ionic strength (mol/kgw) = 5.902e-01

Mass of water (kg) = 1.000e+00

Total CO2 (mol/kg) = 1.952e-03

Temperature (癈) = 25.00

Electrical balance (eq) = 4.552e-03

Percent error, 100*(Cat-|An|)/(Cat+|An|) = 0.43

Iterations = 8

Total H = 1.110145e+02

Total O    = 5.561794e+01

--------------------------Distribution of species--------------------------

| Species | Molality | Activity | Log Molality | Log Activity | Log Gamma | mole V cm?mol |
|---|---|---|---|---|---|---|
| OH- | 6.874e-05 | 4.258e-05 | -4.163 | -4.371 | -0.208 | (0) |
| H+ | 3.046e-10 | 2.312e-10 | -9.516 | -9.636 | -0.120 | 0.00 |
| H2O | 5.551e+01 | 9.831e-01 | 1.744 | -0.007 | 0.000 | 18.07 |
| B | 3.743e-04 | | | | | |
| H2BO3- | 3.115e-04 | 1.791e-04 | -3.507 | -3.747 | -0.240 | (0) |
| H3BO3 | 6.281e-05 | 7.195e-05 | -4.202 | -4.143 | 0.059 | (0) |
| BF(OH)3- | 1.150e-09 | 6.611e-10 | -8.939 | -9.180 | -0.240 | (0) |
| BF2(OH)2- | 6.685e-16 | 3.844e-16 | -15.175 | -15.415 | -0.240 | (0) |
| BF3OH- | 3.978e-24 | 2.288e-24 | -23.400 | -23.641 | -0.240 | (0) |
| BF4- | 8.793e-32 | 5.057e-32 | -31.056 | -31.296 | -0.240 | (0) |
| Br | 7.283e-04 | | | | | |
| Br- | 7.283e-04 | 5.383e-04 | -3.138 | -3.269 | -0.131 | (0) |
| C(-4) | 0.000e+00 | | | | | |
| CH4 | 0.000e+00 | 0.000e+00 | -91.584 | -91.525 | 0.059 | (0) |
| C(4) | 1.952e-03 | | | | | |
| MgCO3 | 5.852e-04 | 6.704e-04 | -3.233 | -3.174 | 0.059 | (0) |
| NaCO3- | 4.438e-04 | 3.029e-04 | -3.353 | -3.519 | -0.166 | (0) |
| HCO3- | 3.992e-04 | 2.725e-04 | -3.399 | -3.565 | -0.166 | (0) |

| Species | Molality | Activity | Log Molality | Log Activity | Log Gamma | |
|---|---|---|---|---|---|---|
| CO3-2 | 2.546e-04 | 5.528e-05 | -3.594 | -4.257 | -0.663 | (0) |
| CaCO3 | 1.562e-04 | 1.790e-04 | -3.806 | -3.747 | 0.059 | (0) |
| MgHCO3+ | 6.348e-05 | 4.052e-05 | -4.197 | -4.392 | -0.195 | (0) |
| NaHCO3 | 3.936e-05 | 4.509e-05 | -4.405 | -4.346 | 0.059 | (0) |
| CaHCO3+ | 9.615e-06 | 6.716e-06 | -5.017 | -5.173 | -0.156 | (0) |
| SrCO3 | 6.408e-07 | 6.408e-07 | -6.193 | -6.193 | 0.000 | (0) |
| CO2 | 1.257e-07 | 1.440e-07 | -6.901 | -6.842 | 0.059 | (0) |
| SrHCO3+ | 1.109e-07 | 7.569e-08 | -6.955 | -7.121 | -0.166 | (0) |

Ca            8.608e-03

| Species | Molality | Activity | Log Molality | Log Activity | Log Gamma | |
|---|---|---|---|---|---|---|
| Ca+2 | 7.615e-03 | 1.932e-03 | -2.118 | -2.714 | -0.596 | (0) |
| CaSO4 | 8.213e-04 | 9.408e-04 | -3.086 | -3.026 | 0.059 | (0) |
| CaCO3 | 1.562e-04 | 1.790e-04 | -3.806 | -3.747 | 0.059 | (0) |
| CaHCO3+ | 9.615e-06 | 6.716e-06 | -5.017 | -5.173 | -0.156 | (0) |
| CaPO4- | 3.741e-06 | 2.554e-06 | -5.427 | -5.593 | -0.166 | (0) |
| CaOH+ | 1.953e-06 | 1.364e-06 | -5.709 | -5.865 | -0.156 | (0) |
| CaF+ | 5.668e-07 | 3.885e-07 | -6.247 | -6.411 | -0.164 | (0) |
| CaHPO4 | 2.178e-07 | 2.495e-07 | -6.662 | -6.603 | 0.059 | (0) |
| CaH2PO4+ | 6.350e-11 | 4.334e-11 | -10.197 | -10.363 | -0.166 | (0) |
| CaHSO4+ | 1.724e-12 | 1.274e-12 | -11.764 | -11.895 | -0.131 | (0) |

Cl            4.891e-01

| Species | Molality | Activity | Log Molality | Log Activity | Log Gamma | |
|---|---|---|---|---|---|---|
| Cl- | 4.891e-01 | 3.092e-01 | -0.311 | -0.510 | -0.199 | (0) |

F            7.081e-05

|  |  |  |  |  |  |
| --- | --- | --- | --- | --- | --- |
| F- | 3.726e-05 | 2.308e-05 | -4.429 | -4.637 | -0.208 (0) |
| MgF+ | 2.957e-05 | 1.938e-05 | -4.529 | -4.713 | -0.183 (0) |
| NaF | 3.412e-06 | 3.909e-06 | -5.467 | -5.408 | 0.059 (0) |
| CaF+ | 5.668e-07 | 3.885e-07 | -6.247 | -6.411 | -0.164 (0) |
| BF(OH)3- | 1.150e-09 | 6.611e-10 | -8.939 | -9.180 | -0.240 (0) |
| HF | 6.984e-12 | 8.001e-12 | -11.156 | -11.097 | 0.059 (0) |
| HF2- | 1.144e-15 | 7.085e-16 | -14.942 | -15.150 | -0.208 (0) |
| BF2(OH)2- | 6.685e-16 | 3.844e-16 | -15.175 | -15.415 | -0.240 (0) |
| H2F2 | 1.456e-22 | 1.668e-22 | -21.837 | -21.778 | 0.059 (0) |
| BF3OH- | 3.978e-24 | 2.288e-24 | -23.400 | -23.641 | -0.240 (0) |
| BF4- | 8.793e-32 | 5.057e-32 | -31.056 | -31.296 | -0.240 (0) |
| SiF6-2 | 2.048e-40 | 0.000e+00 | -39.689 | -40.382 | -0.693 (0) |
| H(0) | 6.604e-31 |  |  |  |  |
| H2 | 3.302e-31 | 3.783e-31 | -30.481 | -30.422 | 0.059 (0) |
| K | 8.133e-03 |  |  |  |  |
| K+ | 8.006e-03 | 5.061e-03 | -2.097 | -2.296 | -0.199 (0) |
| KSO4- | 1.271e-04 | 8.677e-05 | -3.896 | -4.062 | -0.166 (0) |
| KHPO4- | 3.404e-09 | 2.324e-09 | -8.468 | -8.634 | -0.166 (0) |
| Mg | 5.123e-02 |  |  |  |  |
| Mg+2 | 4.389e-02 | 1.271e-02 | -1.358 | -1.896 | -0.538 (0) |
| MgSO4 | 6.346e-03 | 7.269e-03 | -2.198 | -2.138 | 0.059 (0) |
| MgCO3 | 5.852e-04 | 6.704e-04 | -3.233 | -3.174 | 0.059 (0) |

| | | | | | |
|---|---|---|---|---|---|
| MgOH+ | 2.762e-04 | 1.962e-04 | -3.559 | -3.707 | -0.148 (0) |
| MgHCO3+ | 6.348e-05 | 4.052e-05 | -4.197 | -4.392 | -0.195 (0) |
| MgPO4- | 3.319e-05 | 2.265e-05 | -4.479 | -4.645 | -0.166 (0) |
| MgF+ | 2.957e-05 | 1.938e-05 | -4.529 | -4.713 | -0.183 (0) |
| MgHPO4 | 1.936e-06 | 2.218e-06 | -5.713 | -5.654 | 0.059 (0) |
| MgH2PO4+ | 5.318e-10 | 3.630e-10 | -9.274 | -9.440 | -0.166 (0) |
| N(5) | 8.902e-04 | | | | |
| NO3- | 8.902e-04 | 5.327e-04 | -3.051 | -3.274 | -0.223 (0) |
| Na | 4.222e-01 | | | | |
| Na+ | 4.164e-01 | 2.943e-01 | -0.380 | -0.531 | -0.151 (0) |
| NaSO4- | 5.273e-03 | 3.599e-03 | -2.278 | -2.444 | -0.166 (0) |
| NaCO3- | 4.438e-04 | 3.029e-04 | -3.353 | -3.519 | -0.166 (0) |
| NaHCO3 | 3.936e-05 | 4.509e-05 | -4.405 | -4.346 | 0.059 (0) |
| NaF | 3.412e-06 | 3.909e-06 | -5.467 | -5.408 | 0.059 (0) |
| NaHPO4- | 1.980e-07 | 1.351e-07 | -6.703 | -6.869 | -0.166 (0) |
| O(0) | 4.916e-32 | | | | |
| O2 | 2.458e-32 | 2.816e-32 | -31.609 | -31.550 | 0.059 (0) |
| P | 4.046e-05 | | | | |
| MgPO4- | 3.319e-05 | 2.265e-05 | -4.479 | -4.645 | -0.166 (0) |
| CaPO4- | 3.741e-06 | 2.554e-06 | -5.427 | -5.593 | -0.166 (0) |
| MgHPO4 | 1.936e-06 | 2.218e-06 | -5.713 | -5.654 | 0.059 (0) |
| HPO4-2 | 1.161e-06 | 2.355e-07 | -5.935 | -6.628 | -0.693 (0) |
| CaHPO4 | 2.178e-07 | 2.495e-07 | -6.662 | -6.603 | |

0.059      (0)

     NaHPO4-            1.980e-07    1.351e-07    -6.703     -6.869     -
0.166      (0)

     PO4-3             1.664e-08    4.592e-10    -7.779     -9.338     -
1.559      (0)

     KHPO4-            3.404e-09    2.324e-09    -8.468     -8.634     -
0.166      (0)

     H2PO4-            1.284e-09    8.767e-10    -8.891     -9.057     -
0.166      (0)

     MgH2PO4+           5.318e-10    3.630e-10    -9.274     -9.440     -
0.166      (0)

     CaH2PO4+           6.350e-11    4.334e-11    -10.197    -10.363    -
0.166      (0)

S(6)          2.528e-02

     SO4-2             1.270e-02    2.440e-03    -1.896     -2.613     -
0.717      (0)

     MgSO4               6.346e-03    7.269e-03    -2.198     -2.138
0.059      (0)

     NaSO4-            5.273e-03    3.599e-03    -2.278     -2.444     -
0.166      (0)

     CaSO4               8.213e-04    9.408e-04    -3.086     -3.026
0.059      (0)

     KSO4-             1.271e-04    8.677e-05    -3.896     -4.062     -
0.166      (0)

     SrSO4               7.542e-06    8.640e-06    -5.123     -5.063
0.059      (0)

     HSO4-             8.367e-11    5.484e-11    -10.077    -10.261    -
0.183      (0)

     CaHSO4+             1.724e-12    1.274e-12    -11.764    -11.895    -
0.131      (0)

Si          1.113e-04

     H3SiO4-            5.937e-05    3.789e-05    -4.226     -4.421     -
0.195      (0)

     H4SiO4              5.185e-05    5.940e-05    -4.285     -4.226
0.059      (0)

     H2SiO4-2            5.136e-08    1.115e-08    -7.289     -7.953     -
0.663      (0)

     SiF6-2             2.048e-40    0.000e+00    -39.689    -40.382    -0.693
(0)

Sr          8.093e-05

     Sr+2              7.263e-05    1.816e-05    -4.139     -4.741     -

0.602        (0)
    SrSO4                    7.542e-06      8.640e-06       -5.123        -5.063
0.059        (0)
    SrCO3                    6.408e-07      6.408e-07       -6.193        -6.193
0.000        (0)
    SrHCO3+               1.109e-07      7.569e-08       -6.955        -7.121       -
0.166        (0)
    SrOH+                    5.902e-09      3.961e-09       -8.229        -8.402       -
0.173        (0)

----------------------------Saturation indices----------------------------

    Phase                     SI** log IAP      log K(298 K,      1 atm)

    Anhydrite            -0.97        -5.33      -4.36    CaSO4
    Aragonite             1.36        -6.97      -8.34    CaCO3
    Artinite              1.59        11.19       9.60    MgCO3:Mg(OH)2:3H2O
    Brucite               0.52        17.36      16.84    Mg(OH)2
    Calcite               1.51        -6.97      -8.48    CaCO3
    Celestite            -0.72        -7.35      -6.63    SrSO4
    CH4(g)              -88.67       -91.53      -2.86    CH4
    Chalcedony           -0.66        -4.21      -3.55    SiO2
    Chrysotile           11.47        43.67      32.20    Mg3Si2O5(OH)4
    Clinoenstatite        1.82        13.16      11.34    MgSiO3
    CO2(g)               -5.37        -6.84      -1.47    CO2
    Cristobalite         -0.62        -4.21      -3.59    SiO2
    Diopside              5.60        25.50      19.89    CaMgSi2O6
    Dolomite              3.97       -13.12     -17.09    CaMg(CO3)2
    Dolomite(d)           3.42       -13.12     -16.54    CaMg(CO3)2
    Epsomite             -2.42        -4.56      -2.14    MgSO4:7H2O
    FCO3Apatite                       27.22                -87.18     -114.40
Ca9.316Na0.36Mg0.144(PO4)4.8(CO3)1.2F2.48
    Fluorapatite          8.42        -9.18     -17.60    Ca5(PO4)3F
    Fluorite             -1.39       -11.99     -10.60    CaF2
    Forsterite            2.22        30.53      28.31    Mg2SiO4
    Gypsum               -0.76        -5.34      -4.58    CaSO4:2H2O
    H2(g)               -27.27       -30.42      -3.15    H2
    H2O(g)               -1.52        -0.01       1.51    H2O
    Halite               -2.62        -1.04       1.58    NaCl
    Huntite               4.54       -25.43     -29.97    CaMg3(CO3)4
    Hydromagnesite        1.48        -7.28      -8.76    Mg5(CO3)4(OH)2:4H2O

| | | | | |
|---|---|---|---|---|
| Hydroxyapatite | 8.50 | 5.08 | -3.42 | Ca5(PO4)3OH |
| Magadiite | -6.11 | -20.41 | -14.30 | NaSi7O13(OH)3:3H2O |
| Magnesite | 1.88 | -6.15 | -8.03 | MgCO3 |
| Mirabilite | -2.64 | -3.75 | -1.11 | Na2SO4:10H2O |
| Nahcolite | -3.55 | -4.10 | -0.55 | NaHCO3 |
| Natron | -4.08 | -5.39 | -1.31 | Na2CO3:10H2O |
| Nesquehonite | -0.55 | -6.18 | -5.62 | MgCO3:3H2O |
| O2(g) | -28.66 | -31.55 | -2.89 | O2 |
| Portlandite | -6.26 | 16.54 | 22.80 | Ca(OH)2 |
| Quartz | -0.23 | -4.21 | -3.98 | SiO2 |
| Sepiolite | 6.32 | 22.08 | 15.76 | Mg2Si3O7.5OH:3H2O |
| Sepiolite(d) | 3.42 | 22.08 | 18.66 | Mg2Si3O7.5OH:3H2O |
| Silicagel | -1.19 | -4.21 | -3.02 | SiO2 |
| SiO2(a) | -1.50 | -4.21 | -2.71 | SiO2 |
| SrF2 | -5.47 | -14.01 | -8.54 | SrF2 |
| Strontianite | 0.27 | -9.00 | -9.27 | SrCO3 |
| Talc | 13.85 | 35.25 | 21.40 | Mg3Si4O10(OH)2 |
| Thenardite | -3.50 | -3.67 | -0.18 | Na2SO4 |
| Thermonatrite | -5.45 | -5.33 | 0.13 | Na2CO3:H2O |
| Tremolite | 29.67 | 86.25 | 56.57 | Ca2Mg5Si8O22(OH)2 |
| Trona | -8.64 | -9.43 | -0.80 | NaHCO3:Na2CO3:2H2O |

**For a gas, SI = log10(fugacity). Fugacity = pressure * phi / 1 atm.
   For ideal gases, phi = 1.
* * *
End of simulation.
* * ** * *
Reading input data for simulation 2.
* * ** * *
End of Run after 0.185 Seconds.
* * *
| | |
|---|---|
| Cyanobacteria+ virus 8 Day | Input file: C:\Users\XUHENGCHAO\Desktop\PHreeQC\culture\CP1-8.pqi
Output file: C:\Users\XUHENGCHAO\Desktop\PHreeQC\culture\CP1-8.pqo
Database file: C:\Program Files\USGS\Phreeqc Interactive 3.3.12-12704\database\wateq4f.dat |
* * *
Reading data base.
* * *
    SOLUTION_MASTER_SPECIES
    SOLUTION_SPECIES
    PHASES
    EXCHANGE_MASTER_SPECIES
    EXCHANGE_SPECIES
    SURFACE_MASTER_SPECIES
    SURFACE_SPECIES
    RATES
    END
* * *
Reading input data for simulation 1.
* * *
    DATABASE    C:\Program    Files\USGS\Phreeqc    Interactive    3.3.12-12704\database\wateq4f.dat
    SOLUTION 1
        temp    25
        pH    8
        pe    4
        redox    pe
        units    mmol/l
        density    1.02
        Alkalinity 3.74
        B    0.37
        Br    0.72
        C    1.5
        Ca    8.5
        Cl    483.51
        F    0.07
        K    8.04
        Mg    50.64
        N(5)    0.88
        Na    417.34
        P    0.04
        S(6)    24.99
        Si    0.11

```
              Sr              0.08
              water      1 # kg
* * *
Beginning of initial solution calculations.
* * *
Initial solution 1.

pH will be adjusted to obtain desired alkalinity.

---------------------------Solution composition---------------------------

    Elements              Molality          Moles

    Alkalinity        3.783e-03    3.783e-03
    B                    3.743e-04    3.743e-04
    Br                   7.283e-04    7.283e-04
    C                    1.517e-03    1.517e-03
    Ca                   8.598e-03    8.598e-03
    Cl                   4.891e-01    4.891e-01
    F                    7.081e-05    7.081e-05
    K                    8.133e-03    8.133e-03
    Mg                   5.122e-02    5.122e-02
    N(5)                 8.901e-04    8.901e-04
    Na                   4.221e-01    4.221e-01
    P                    4.046e-05    4.046e-05
    S(6)                 2.528e-02    2.528e-02
    Si                   1.113e-04    1.113e-04
    Sr                   8.092e-05    8.092e-05

---------------------------Description of solution---------------------------

                                    pH    =    9.833         Adjust
alkalinity
                                    pe    =    4.000
                      Activity of water   =    0.983
                Ionic strength (mol/kgw)  =    5.901e-01
                      Mass of water (kg)  =    1.000e+00
                     Total CO2 (mol/kg)   =    1.517e-03
                     Temperature (癈)     =    25.00
```

```
                    Electrical balance (eq)   =    4.936e-03
        Percent error, 100*(Cat-|An|)/(Cat+|An|)   =    0.47
                            Iterations   =    8
                            Total H   = 1.110144e+02
                            Total O   = 5.561683e+01

--------------------------Distribution of species--------------------------

                                                    Log              Log
Log        mole V
    Species              Molality    Activity   Molality   Activity      Gamma
cm?mol

    OH-                  1.083e-04    6.705e-05    -3.966      -4.174        -
0.208       (0)
    H+                   1.934e-10    1.468e-10    -9.714      -9.833        -
0.120        0.00
    H2O                  5.551e+01    9.831e-01     1.744      -0.007
0.000      18.07
B             3.743e-04
    H2BO3-               3.318e-04    1.908e-04    -3.479      -3.719        -
0.240       (0)
    H3BO3                4.249e-05    4.867e-05    -4.372      -4.313
0.059       (0)
    BF(OH)3-             7.779e-10    4.474e-10    -9.109      -9.349        -
0.240       (0)
    BF2(OH)2-            2.873e-16    1.653e-16   -15.542     -15.782        -
0.240       (0)
    BF3OH-               1.086e-24    6.247e-25   -23.964     -24.204        -
0.240       (0)
    BF4-                 1.525e-32    8.772e-33   -31.817     -32.057        -
0.240       (0)
Br            7.283e-04
    Br-                  7.283e-04    5.383e-04    -3.138      -3.269        -
0.131       (0)
C(-4)         0.000e+00
    CH4                  0.000e+00    0.000e+00   -93.622     -93.563
0.059       (0)
C(4)          1.517e-03
    MgCO3                5.025e-04    5.756e-04    -3.299      -3.240
0.059       (0)
```

| Species | Molality | Activity | Log Molality | Log Activity | Log Gamma | |
|---|---|---|---|---|---|---|
| NaCO3- | 3.815e-04 | 2.604e-04 | -3.418 | -3.584 | -0.166 | (0) |
| CO3-2 | 2.189e-04 | 4.752e-05 | -3.660 | -4.323 | -0.663 | (0) |
| HCO3- | 2.179e-04 | 1.487e-04 | -3.662 | -3.828 | -0.166 | (0) |
| CaCO3 | 1.345e-04 | 1.541e-04 | -3.871 | -3.812 | 0.059 | (0) |
| MgHCO3+ | 3.461e-05 | 2.209e-05 | -4.461 | -4.656 | -0.195 | (0) |
| NaHCO3 | 2.149e-05 | 2.462e-05 | -4.668 | -4.609 | 0.059 | (0) |
| CaHCO3+ | 5.257e-06 | 3.672e-06 | -5.279 | -5.435 | -0.156 | (0) |
| SrCO3 | 5.517e-07 | 5.517e-07 | -6.258 | -6.258 | 0.000 | (0) |
| SrHCO3+ | 6.062e-08 | 4.138e-08 | -7.217 | -7.383 | -0.166 | (0) |
| CO2 | 4.358e-08 | 4.992e-08 | -7.361 | -7.302 | 0.059 | (0) |
| Ca | 8.598e-03 | | | | | |
| Ca+2 | 7.628e-03 | 1.936e-03 | -2.118 | -2.713 | -0.596 | (0) |
| CaSO4 | 8.229e-04 | 9.426e-04 | -3.085 | -3.026 | 0.059 | (0) |
| CaCO3 | 1.345e-04 | 1.541e-04 | -3.871 | -3.812 | 0.059 | (0) |
| CaHCO3+ | 5.257e-06 | 3.672e-06 | -5.279 | -5.435 | -0.156 | (0) |
| CaPO4- | 3.873e-06 | 2.644e-06 | -5.412 | -5.578 | -0.166 | (0) |
| CaOH+ | 3.080e-06 | 2.152e-06 | -5.511 | -5.667 | -0.156 | (0) |
| CaF+ | 5.680e-07 | 3.893e-07 | -6.246 | -6.410 | -0.164 | (0) |
| CaHPO4 | 1.432e-07 | 1.640e-07 | -6.844 | -6.785 | 0.059 | (0) |
| CaH2PO4+ | 2.651e-11 | 1.809e-11 | -10.577 | -10.742 | -0.166 | (0) |
| CaHSO4+ | 1.097e-12 | 8.105e-13 | -11.960 | -12.091 | -0.131 | (0) |

Cl                4.891e-01

   Cl-              4.891e-01   3.092e-01    -0.311    -0.510    -0.199
(0)

F                7.081e-05

   F-               3.728e-05   2.309e-05    -4.429    -4.637    -
0.208        (0)

   MgF+             2.955e-05   1.937e-05    -4.529    -4.713    -
0.183        (0)

   NaF              3.414e-06   3.911e-06    -5.467    -5.408
0.059        (0)

   CaF+             5.680e-07   3.893e-07    -6.246    -6.410    -
0.164        (0)

   BF(OH)3-         7.779e-10   4.474e-10    -9.109    -9.349    -
0.240        (0)

   HF               4.437e-12   5.083e-12   -11.353   -11.294
0.059        (0)

   HF2-             7.270e-16   4.503e-16   -15.138   -15.347    -
0.208        (0)

   BF2(OH)2-        2.873e-16   1.653e-16   -15.542   -15.782    -
0.240        (0)

   H2F2             5.877e-23   6.732e-23   -22.231   -22.172
0.059        (0)

   BF3OH-           1.086e-24   6.247e-25   -23.964   -24.204    -
0.240        (0)

   BF4-             1.525e-32   8.772e-33   -31.817   -32.057    -
0.240        (0)

   SiF6-2           0.000e+00   0.000e+00   -40.593   -41.286    -
0.693        (0)

H(0)             2.663e-31

   H2               1.332e-31   1.525e-31   -30.876   -30.817
0.059        (0)

K                8.133e-03

   K+               8.006e-03   5.061e-03    -2.097    -2.296    -
0.199        (0)

   KSO4-            1.271e-04   8.678e-05    -3.896    -4.062    -
0.166        (0)

   KHPO4-           2.234e-09   1.525e-09    -8.651    -8.817    -
0.166        (0)

Mg               5.122e-02

   Mg+2             4.385e-02   1.270e-02    -1.358    -1.896    -
0.538        (0)

```
     MgSO4            6.341e-03   7.263e-03    -2.198    -2.139    0.059    (0)
     MgCO3            5.025e-04   5.756e-04    -3.299    -3.240    0.059    (0)
     MgOH+            4.345e-04   3.087e-04    -3.362    -3.510   -0.148    (0)
     MgHCO3+          3.461e-05   2.209e-05    -4.461    -4.656   -0.195    (0)
     MgPO4-           3.426e-05   2.339e-05    -4.465    -4.631   -0.166    (0)
     MgF+             2.955e-05   1.937e-05    -4.529    -4.713   -0.183    (0)
     MgHPO4           1.270e-06   1.454e-06    -5.896    -5.837    0.059    (0)
     MgH2PO4+         2.214e-10   1.511e-10    -9.655    -9.821   -0.166    (0)
N(5)           8.901e-04
     NO3-             8.901e-04   5.327e-04    -3.051    -3.274   -0.223    (0)
Na             4.221e-01
     Na+              4.165e-01   2.943e-01    -0.380    -0.531   -0.151    (0)
     NaSO4-           5.274e-03   3.600e-03    -2.278    -2.444   -0.166    (0)
     NaCO3-           3.815e-04   2.604e-04    -3.418    -3.584   -0.166    (0)
     NaHCO3           2.149e-05   2.462e-05    -4.668    -4.609    0.059    (0)
     NaF              3.414e-06   3.911e-06    -5.467    -5.408    0.059    (0)
     NaHPO4-          1.299e-07   8.869e-08    -6.886    -7.052   -0.166    (0)
O(0)           3.023e-31
     O2               1.512e-31   1.732e-31   -30.821   -30.762    0.059    (0)
P              4.046e-05
     MgPO4-           3.426e-05   2.339e-05    -4.465    -4.631   -0.166    (0)
     CaPO4-           3.873e-06   2.644e-06    -5.412    -5.578   -0.166    (0)
     MgHPO4           1.270e-06   1.454e-06    -5.896    -5.837
```

0.059      (0)
   HPO4-2              7.620e-07    1.545e-07    -6.118     -6.811     -
0.693      (0)
    CaHPO4               1.432e-07     1.640e-07     -6.844      -6.785
0.059      (0)
    NaHPO4-            1.299e-07    8.869e-08    -6.886     -7.052     -
0.166      (0)
    PO4-3               1.719e-08    4.746e-10    -7.765     -9.324     -
1.559      (0)
    KHPO4-             2.234e-09    1.525e-09    -8.651     -8.817     -
0.166      (0)
    H2PO4-             5.352e-10    3.653e-10    -9.271     -9.437     -
0.166      (0)
    MgH2PO4+         2.214e-10     1.511e-10     -9.655     -9.821     -
0.166      (0)
    CaH2PO4+         2.651e-11     1.809e-11    -10.577    -10.742     -
0.166      (0)
S(6)            2.528e-02
    SO4-2               1.271e-02    2.441e-03    -1.896     -2.613     -
0.717      (0)
    MgSO4               6.341e-03     7.263e-03     -2.198      -2.139
0.059      (0)
    NaSO4-             5.274e-03    3.600e-03    -2.278     -2.444     -
0.166      (0)
    CaSO4               8.229e-04     9.426e-04     -3.085      -3.026
0.059      (0)
    KSO4-               1.271e-04    8.678e-05    -3.896     -4.062     -
0.166      (0)
    SrSO4               7.556e-06     8.656e-06     -5.122      -5.063
0.059      (0)
    HSO4-               5.314e-11    3.483e-11   -10.275    -10.458     -
0.183      (0)
    CaHSO4+          1.097e-12    8.105e-13   -11.960    -12.091     -
0.131      (0)
Si              1.113e-04
    H3SiO4-            7.151e-05    4.564e-05    -4.146     -4.341     -
0.195      (0)
    H4SiO4               3.966e-05     4.544e-05     -4.402      -4.343
0.059      (0)
    H2SiO4-2        9.742e-08    2.115e-08    -7.011     -7.675     -
0.663      (0)

SiF6-2          0.000e+00    0.000e+00    -40.593    -41.286    -0.693    (0)

Sr          8.092e-05
  Sr+2          7.274e-05    1.819e-05    -4.138    -4.740    -0.602    (0)
  SrSO4          7.556e-06    8.656e-06    -5.122    -5.063    0.059    (0)
  SrCO3          5.517e-07    5.517e-07    -6.258    -6.258    0.000    (0)
  SrHCO3+          6.062e-08    4.138e-08    -7.217    -7.383    -0.166    (0)
  SrOH+          9.310e-09    6.248e-09    -8.031    -8.204    -0.173    (0)

----------------------------Saturation indices----------------------------

  Phase                SI** log IAP    log K(298 K,    1 atm)

  Anhydrite          -0.96    -5.33    -4.36    CaSO4
  Aragonite          1.30    -7.04    -8.34    CaCO3
  Artinite          1.91    11.51    9.60    MgCO3:Mg(OH)2:3H2O
  Brucite          0.92    17.76    16.84    Mg(OH)2
  Calcite          1.44    -7.04    -8.48    CaCO3
  Celestite          -0.72    -7.35    -6.63    SrSO4
  CH4(g)          -90.70    -93.56    -2.86    CH4
  Chalcedony          -0.78    -4.33    -3.55    SiO2
  Chrysotile          12.42    44.62    32.20    Mg3Si2O5(OH)4
  Clinoenstatite    2.09    13.44    11.34    MgSiO3
  CO2(g)          -5.83    -7.30    -1.47    CO2
  Cristobalite    -0.74    -4.33    -3.59    SiO2
  Diopside          6.16    26.05    19.89    CaMgSi2O6
  Dolomite          3.83    -13.26    -17.09    CaMg(CO3)2
  Dolomite(d)          3.28    -13.26    -16.54    CaMg(CO3)2
  Epsomite          -2.42    -4.56    -2.14    MgSO4:7H2O
  FCO3Apatite                27.22                -87.18    -114.40
Ca9.316Na0.36Mg0.144(PO4)4.8(CO3)1.2F2.48
  Fluorapatite    8.46    -9.14    -17.60    Ca5(PO4)3F
  Fluorite          -1.39    -11.99    -10.60    CaF2
  Forsterite          2.89    31.20    28.31    Mg2SiO4
  Gypsum          -0.76    -5.34    -4.58    CaSO4:2H2O
  H2(g)          -27.67    -30.82    -3.15    H2

| | | | |
|---|---|---|---|
| H2O(g) | -1.52 | -0.01 | 1.51 | H2O |
| Halite | -2.62 | -1.04 | 1.58 | NaCl |
| Huntite | 4.27 | -25.69 | -29.97 | CaMg3(CO3)4 |
| Hydromagnesite | 1.61 | -7.15 | -8.76 | Mg5(CO3)4(OH)2:4H2O |
| Hydroxyapatite | 8.75 | 5.33 | -3.42 | Ca5(PO4)3OH |
| Magadiite | -6.73 | -21.03 | -14.30 | NaSi7O13(OH)3:3H2O |
| Magnesite | 1.81 | -6.22 | -8.03 | MgCO3 |
| Mirabilite | -2.63 | -3.75 | -1.11 | Na2SO4:10H2O |
| Nahcolite | -3.81 | -4.36 | -0.55 | NaHCO3 |
| Natron | -4.15 | -5.46 | -1.31 | Na2CO3:10H2O |
| Nesquehonite | -0.62 | -6.24 | -5.62 | MgCO3:3H2O |
| O2(g) | -27.87 | -30.76 | -2.89 | O2 |
| Portlandite | -5.86 | 16.94 | 22.80 | Ca(OH)2 |
| Quartz | -0.35 | -4.33 | -3.98 | SiO2 |
| Sepiolite | 6.76 | 22.52 | 15.76 | Mg2Si3O7.5OH:3H2O |
| Sepiolite(d) | 3.86 | 22.52 | 18.66 | Mg2Si3O7.5OH:3H2O |
| Silicagel | -1.31 | -4.33 | -3.02 | SiO2 |
| SiO2(a) | -1.62 | -4.33 | -2.71 | SiO2 |
| SrF2 | -5.47 | -14.01 | -8.54 | SrF2 |
| Strontianite | 0.21 | -9.06 | -9.27 | SrCO3 |
| Talc | 14.57 | 35.97 | 21.40 | Mg3Si4O10(OH)2 |
| Thenardite | -3.50 | -3.67 | -0.18 | Na2SO4 |
| Thermonatrite | -5.52 | -5.39 | 0.13 | Na2CO3:H2O |
| Tremolite | 31.50 | 88.08 | 56.57 | Ca2Mg5Si8O22(OH)2 |
| Trona | -8.96 | -9.76 | -0.80 | NaHCO3:Na2CO3:2H2O |

**For a gas, SI = log10(fugacity). Fugacity = pressure * phi / 1 atm.
   For ideal gases, phi = 1.
* * *
End of simulation.
* * ** * *
Reading input data for simulation 2.
* * ** * *
End of Run after 0.39 Seconds.
* * *
| | |
|---|---|
| Cyanobacteria+ virus

10  Day | Input file: C:\Users\XUHENGCHAO\Desktop\PHreeQC\culture\CP1-10.pqi
Output file: C:\Users\XUHENGCHAO\Desktop\PHreeQC\culture\CP1-10.pqo
Database file: C:\Program Files\USGS\Phreeqc Interactive 3.3.12-12704\database\wateq4f.dat
* * *
Reading data base.
* * *
    SOLUTION_MASTER_SPECIES
    SOLUTION_SPECIES
    PHASES
    EXCHANGE_MASTER_SPECIES
    EXCHANGE_SPECIES
    SURFACE_MASTER_SPECIES
    SURFACE_SPECIES
    RATES
    END
* * *
Reading input data for simulation 1.
* * *
    DATABASE C:\Program Files\USGS\Phreeqc Interactive 3.3.12-12704\database\wateq4f.dat
    SOLUTION 1
        temp     25
        pH       8
        pe       4
        redox    pe
        units    mmol/l
        density  1.02
        Alkalinity 1.57
        B       0.37
        Br      0.72
        C       0.96
        Ca      7.64
        Cl      483.51
        F       0.07
        K       8.04
        Mg     50.05
        N(5)    0.88 |

```
                Na           417.34
                P             0.04
                S(6)         24.99
                Si            0.11
                Sr            0.08
                water       1 # kg
* * *
Beginning of initial solution calculations.
* * *
Initial solution 1.

pH will be adjusted to obtain desired alkalinity.

----------------------------Solution composition----------------------------

      Elements              Molality             Moles

      Alkalinity           1.588e-03      1.588e-03
      B                       3.742e-04      3.742e-04
      Br                      7.282e-04      7.282e-04
      C                       9.709e-04      9.709e-04
      Ca                     7.727e-03      7.727e-03
      Cl                      4.890e-01      4.890e-01
      F                       7.079e-05      7.079e-05
      K                      8.131e-03      8.131e-03
      Mg                     5.062e-02      5.062e-02
      N(5)                   8.900e-04      8.900e-04
      Na                     4.221e-01      4.221e-01
      P                       4.045e-05      4.045e-05
      S(6)                   2.527e-02      2.527e-02
      Si                      1.112e-04      1.112e-04
      Sr                     8.091e-05      8.091e-05

----------------------------Description of solution----------------------------

                                    pH    =     8.857         Adjust
alkalinity

                                    pe    =    4.000
                        Activity of water    =    0.983
```

```
                          Ionic strength (mol/kgw)   =     5.883e-01
                              Mass of water (kg)    =     1.000e+00
                              Total CO2 (mol/kg)   =     9.709e-04
                               Temperature (癈)   =    25.00
                           Electrical balance (eq)   =     4.197e-03
        Percent error, 100*(Cat-|An|)/(Cat+|An|)   =     0.40
                                    Iterations   =     9
                                     Total H    = 1.110145e+02
                                     Total O    = 5.561468e+01

----------------------------Distribution of species----------------------------

                                                     Log          Log
Log      mole V
    Species                    Molality    Activity   Molality   Activity    Gamma
cm?mol

    OH-                        1.142e-05   7.074e-06    -4.942     -5.150     -
0.208       (0)
    H+                         1.833e-09   1.391e-09    -8.737     -8.857     -
0.120        0.00
    H2O                        5.551e+01   9.831e-01     1.744     -0.007
0.000      18.07
B                 3.742e-04
    H3BO3                      2.053e-04   2.350e-04    -3.688     -3.629
0.059       (0)
    H2BO3-                     1.689e-04   9.721e-05    -3.772     -4.012     -
0.240       (0)
    BF(OH)3-                   3.754e-09   2.160e-09    -8.425     -8.666     -
0.240       (0)
    BF2(OH)2-                  1.314e-14   7.563e-15   -13.881    -14.121     -
0.240       (0)
    BF3OH-                     4.708e-22   2.709e-22   -21.327    -21.567     -
0.240       (0)
    BF4-                       6.267e-29   3.606e-29   -28.203    -28.443     -
0.240       (0)
Br                7.282e-04
    Br-                        7.282e-04   5.381e-04    -3.138     -3.269     -
0.131       (0)
C(-4)           0.000e+00
    CH4                        0.000e+00   0.000e+00   -84.456    -84.397
```

```
                0.059     (0)
C(4)          9.709e-04
   HCO3-           5.175e-04   3.533e-04   -3.286   -3.452   -
0.166     (0)
   MgCO3              1.263e-04   1.446e-04   -3.899   -3.840
0.059     (0)
   NaCO3-          9.564e-05   6.530e-05   -4.019   -4.185   -
0.166     (0)
   MgHCO3+          8.236e-05   5.259e-05   -4.084   -4.279   -
0.195     (0)
   CO3-2           5.480e-05   1.191e-05   -4.261   -4.924   -
0.663     (0)
   NaHCO3              5.109e-05   5.850e-05   -4.292   -4.233
0.059     (0)
   CaCO3              3.066e-05   3.510e-05   -4.513   -4.455
0.059     (0)
   CaHCO3+          1.135e-05   7.928e-06   -4.945   -5.101   -
0.156     (0)
   CO2                9.817e-07   1.124e-06   -6.008   -5.949
0.059     (0)
   SrHCO3+          1.446e-07   9.871e-08   -6.840   -7.006   -
0.166     (0)
   SrCO3              1.389e-07   1.389e-07   -6.857   -6.857
0.000     (0)
Ca            7.727e-03
   Ca+2            6.930e-03   1.759e-03   -2.159   -2.755   -
0.595     (0)
   CaSO4              7.503e-04   8.591e-04   -3.125   -3.066
0.059     (0)
   CaCO3              3.066e-05   3.510e-05   -4.513   -4.455
0.059     (0)
   CaHCO3+          1.135e-05   7.928e-06   -4.945   -5.101   -
0.156     (0)
   CaPO4-          2.385e-06   1.628e-06   -5.623   -5.788   -
0.166     (0)
   CaHPO4              8.362e-07   9.575e-07   -6.078   -6.019
0.059     (0)
   CaF+            5.161e-07   3.538e-07   -6.287   -6.451   -
0.164     (0)
   CaOH+           2.953e-07   2.063e-07   -6.530   -6.685   -
0.156     (0)
```

|  | CaH2PO4+ | 1.466e-09 | 1.001e-09 | -8.834 | -8.999 | -0.166 | (0) |
|  | CaHSO4+ | 9.476e-12 | 7.002e-12 | -11.023 | -11.155 | -0.131 | (0) |
| Cl | 4.890e-01 |  |  |  |  |  |  |
|  | Cl- | 4.890e-01 | 3.092e-01 | -0.311 | -0.510 | -0.199 | (0) |
| F | 7.079e-05 |  |  |  |  |  |  |
|  | F- | 3.726e-05 | 2.309e-05 | -4.429 | -4.637 | -0.208 | (0) |
|  | MgF+ | 2.960e-05 | 1.940e-05 | -4.529 | -4.712 | -0.183 | (0) |
|  | NaF | 3.417e-06 | 3.912e-06 | -5.466 | -5.408 | 0.059 | (0) |
|  | CaF+ | 5.161e-07 | 3.538e-07 | -6.287 | -6.451 | -0.164 | (0) |
|  | BF(OH)3- | 3.754e-09 | 2.160e-09 | -8.425 | -8.666 | -0.240 | (0) |
|  | HF | 4.207e-11 | 4.817e-11 | -10.376 | -10.317 | 0.059 | (0) |
|  | BF2(OH)2- | 1.314e-14 | 7.563e-15 | -13.881 | -14.121 | -0.240 | (0) |
|  | HF2- | 6.887e-15 | 4.267e-15 | -14.162 | -14.370 | -0.208 | (0) |
|  | H2F2 | 5.281e-21 | 6.047e-21 | -20.277 | -20.218 | 0.059 | (0) |
|  | BF3OH- | 4.708e-22 | 2.709e-22 | -21.327 | -21.567 | -0.240 | (0) |
|  | BF4- | 6.267e-29 | 3.606e-29 | -28.203 | -28.443 | -0.240 | (0) |
|  | SiF6-2 | 4.847e-37 | 9.840e-38 | -36.315 | -37.007 | -0.692 | (0) |
| H(0) | 2.393e-29 |  |  |  |  |  |  |
|  | H2 | 1.197e-29 | 1.370e-29 | -28.922 | -28.863 | 0.059 | (0) |
| K | 8.131e-03 |  |  |  |  |  |  |
|  | K+ | 8.004e-03 | 5.061e-03 | -2.097 | -2.296 | -0.199 | (0) |
|  | KSO4- | 1.275e-04 | 8.703e-05 | -3.895 | -4.060 | -0.166 | (0) |
|  | KHPO4- | 1.435e-08 | 9.795e-09 | -7.843 | -8.009 | - |  |

| | | | | | |
|---|---|---|---|---|---|
| 0.166 | (0) | | | | |
| Mg | 5.062e-02 | | | | |
| Mg+2 | 4.393e-02 | 1.272e-02 | -1.357 | -1.895 | - |
| 0.538 | (0) | | | | |
| MgSO4 | 6.373e-03 | 7.298e-03 | -2.196 | -2.137 | |
| 0.059 | (0) | | | | |
| MgCO3 | 1.263e-04 | 1.446e-04 | -3.899 | -3.840 | |
| 0.059 | (0) | | | | |
| MgHCO3+ | 8.236e-05 | 5.259e-05 | -4.084 | -4.279 | - |
| 0.195 | (0) | | | | |
| MgOH+ | 4.592e-05 | 3.264e-05 | -4.338 | -4.486 | - |
| 0.148 | (0) | | | | |
| MgF+ | 2.960e-05 | 1.940e-05 | -4.529 | -4.712 | - |
| 0.183 | (0) | | | | |
| MgPO4- | 2.326e-05 | 1.588e-05 | -4.633 | -4.799 | - |
| 0.166 | (0) | | | | |
| MgHPO4 | 8.175e-06 | 9.361e-06 | -5.088 | -5.029 | |
| 0.059 | (0) | | | | |
| MgH2PO4+ | 1.350e-08 | 9.220e-09 | -7.870 | -8.035 | - |
| 0.166 | (0) | | | | |
| N(5) | 8.900e-04 | | | | |
| NO3- | 8.900e-04 | 5.328e-04 | -3.051 | -3.273 | - |
| 0.223 | (0) | | | | |
| Na | 4.221e-01 | | | | |
| Na+ | 4.166e-01 | 2.945e-01 | -0.380 | -0.531 | - |
| 0.151 | (0) | | | | |
| NaSO4- | 5.290e-03 | 3.612e-03 | -2.277 | -2.442 | - |
| 0.166 | (0) | | | | |
| NaCO3- | 9.564e-05 | 6.530e-05 | -4.019 | -4.185 | - |
| 0.166 | (0) | | | | |
| NaHCO3 | 5.109e-05 | 5.850e-05 | -4.292 | -4.233 | |
| 0.059 | (0) | | | | |
| NaF | 3.417e-06 | 3.912e-06 | -5.466 | -5.408 | |
| 0.059 | (0) | | | | |
| NaHPO4- | 8.347e-07 | 5.699e-07 | -6.078 | -6.244 | - |
| 0.166 | (0) | | | | |
| O(0) | 3.748e-35 | | | | |
| O2 | 1.874e-35 | 2.146e-35 | -34.727 | -34.668 | |
| 0.059 | (0) | | | | |
| P | 4.045e-05 | | | | |
| MgPO4- | 2.326e-05 | 1.588e-05 | -4.633 | -4.799 | - |

```
0.166       (0)
   MgHPO4                  8.175e-06    9.361e-06      -5.088      -5.029
0.059       (0)
   HPO4-2                  4.889e-06    9.926e-07     -5.311      -6.003    -
0.692       (0)
   CaPO4-                  2.385e-06    1.628e-06     -5.623      -5.788    -
0.166       (0)
   CaHPO4                  8.362e-07    9.575e-07      -6.078      -6.019
0.059       (0)
   NaHPO4-                 8.347e-07    5.699e-07      -6.078      -6.244    -
0.166       (0)
   H2PO4-                  3.258e-08    2.224e-08     -7.487      -7.653    -
0.166       (0)
   KHPO4-                  1.435e-08    9.795e-09     -7.843      -8.009    -
0.166       (0)
   MgH2PO4+                1.350e-08    9.220e-09     -7.870      -8.035    -
0.166       (0)
   PO4-3                   1.162e-08    3.216e-10     -7.935      -9.493    -
1.558       (0)
   CaH2PO4+                1.466e-09    1.001e-09     -8.834      -8.999    -
0.166       (0)
S(6)            2.527e-02
   SO4-2                   1.272e-02    2.447e-03     -1.895      -2.611    -
0.716       (0)
   MgSO4                   6.373e-03    7.298e-03      -2.196      -2.137
0.059       (0)
   NaSO4-                  5.290e-03    3.612e-03     -2.277      -2.442    -
0.166       (0)
   CaSO4                   7.503e-04    8.591e-04      -3.125      -3.066
0.059       (0)
   KSO4-                   1.275e-04    8.703e-05     -3.895      -4.060    -
0.166       (0)
   SrSO4                   7.612e-06    8.716e-06      -5.119      -5.060
0.059       (0)
   HSO4-                   5.049e-10    3.310e-10     -9.297      -9.480    -
0.183       (0)
   CaHSO4+                 9.476e-12    7.002e-12    -11.023     -11.155    -
0.131       (0)
Si              1.112e-04
   H4SiO4                  9.348e-05    1.070e-04      -4.029      -3.970
0.059       (0)
```

| | | | | | |
|---|---|---|---|---|---|
| H3SiO4- | 1.777e-05 | 1.134e-05 | -4.750 | -4.945 | -0.195 (0) |
| H2SiO4-2 | 2.552e-09 | 5.547e-10 | -8.593 | -9.256 | -0.663 (0) |
| SiF6-2 | 4.847e-37 | 9.840e-38 | -36.315 | -37.007 | -0.692 (0) |

Sr            8.091e-05

| | | | | | |
|---|---|---|---|---|---|
| Sr+2 | 7.301e-05 | 1.827e-05 | -4.137 | -4.738 | -0.602 (0) |
| SrSO4 | 7.612e-06 | 8.716e-06 | -5.119 | -5.060 | 0.059 (0) |
| SrHCO3+ | 1.446e-07 | 9.871e-08 | -6.840 | -7.006 | -0.166 (0) |
| SrCO3 | 1.389e-07 | 1.389e-07 | -6.857 | -6.857 | 0.000 (0) |
| SrOH+ | 9.861e-10 | 6.620e-10 | -9.006 | -9.179 | -0.173 (0) |

-----------------------------Saturation indices-----------------------------

| Phase | SI** | log IAP | log K(298 K, | 1 atm) | |
|---|---|---|---|---|---|
| Anhydrite | -1.01 | -5.37 | -4.36 | CaSO4 | |
| Aragonite | 0.66 | -7.68 | -8.34 | CaCO3 | |
| Artinite | -0.64 | 8.96 | 9.60 | MgCO3:Mg(OH)2:3H2O | |
| Brucite | -1.04 | 15.80 | 16.84 | Mg(OH)2 | |
| Calcite | 0.80 | -7.68 | -8.48 | CaCO3 | |
| Celestite | -0.72 | -7.35 | -6.63 | SrSO4 | |
| CH4(g) | -81.54 | -84.40 | -2.86 | CH4 | |
| Chalcedony | -0.40 | -3.96 | -3.55 | SiO2 | |
| Chrysotile | 7.30 | 39.50 | 32.20 | Mg3Si2O5(OH)4 | |
| Clinoenstatite | 0.51 | 11.85 | 11.34 | MgSiO3 | |
| CO2(g) | -4.48 | -5.95 | -1.47 | CO2 | |
| Cristobalite | -0.37 | -3.96 | -3.59 | SiO2 | |
| Diopside | 2.96 | 22.85 | 19.89 | CaMgSi2O6 | |
| Dolomite | 2.59 | -14.50 | -17.09 | CaMg(CO3)2 | |
| Dolomite(d) | 2.04 | -14.50 | -16.54 | CaMg(CO3)2 | |
| Epsomite | -2.42 | -4.56 | -2.14 | MgSO4:7H2O | |
| FCO3Apatite | | 25.30 | | -89.10 | -114.40 |
| Ca9.316Na0.36Mg0.144(PO4)4.8(CO3)1.2F2.48 | | | | | |
| Fluorapatite | 7.75 | -9.85 | -17.60 | Ca5(PO4)3F | |

```
        Fluorite           -1.43     -12.03   -10.60   CaF2
        Forsterite         -0.64     27.66    28.31    Mg2SiO4
        Gypsum                -0.80     -5.38    -4.58   CaSO4:2H2O
        H2(g)             -25.71     -28.86    -3.15   H2
        H2O(g)             -1.52     -0.01     1.51    H2O
        Halite             -2.62     -1.04     1.58    NaCl
        Huntite             1.83     -28.14   -29.97   CaMg3(CO3)4
        Hydromagnesite    -2.74     -11.51    -8.76    Mg5(CO3)4(OH)2:4H2O
        Hydroxyapatite     7.06      3.64     -3.42    Ca5(PO4)3OH
        Magadiite          -5.10     -19.40   -14.30   NaSi7O13(OH)3:3H2O
        Magnesite           1.21     -6.82    -8.03    MgCO3
        Mirabilite         -2.63     -3.75    -1.11    Na2SO4:10H2O
        Nahcolite          -3.43     -3.98    -0.55    NaHCO3
        Natron             -4.75     -6.06    -1.31    Na2CO3:10H2O
        Nesquehonite       -1.22     -6.84    -5.62    MgCO3:3H2O
        O2(g)             -31.78     -34.67    -2.89   O2
        Portlandite        -7.86     14.94    22.80    Ca(OH)2
        Quartz              0.02     -3.96    -3.98    SiO2
        Sepiolite           3.97     19.73    15.76    Mg2Si3O7.5OH:3H2O
        Sepiolite(d)        1.07     19.73    18.66    Mg2Si3O7.5OH:3H2O
        Silicagel          -0.94     -3.96    -3.02    SiO2
        SiO2(a)            -1.24     -3.96    -2.71    SiO2
        SrF2               -5.47     -14.01    -8.54   SrF2
        Strontianite       -0.39     -9.66    -9.27    SrCO3
        Talc               10.20     31.60    21.40    Mg3Si4O10(OH)2
        Thenardite         -3.49     -3.67    -0.18    Na2SO4
        Thermonatrite      -6.12     -5.99     0.13    Na2CO3:H2O
        Tremolite          20.73     77.30    56.57    Ca2Mg5Si8O22(OH)2
        Trona              -9.19     -9.98    -0.80    NaHCO3:Na2CO3:2H2O

**For a gas, SI = log10(fugacity). Fugacity = pressure * phi / 1 atm.
   For ideal gases, phi = 1.
* * *
End of simulation.
* * ** * *
Reading input data for simulation 2.
* * *
```

| | |
|---|---|
| | ------------------------------
End of Run after 0.172 Seconds.
------------------------------ |
| Cyanobacteria+
virus
12 Day | Input file: C:\Users\XUHENGCHAO\Desktop\PHreeQC\culture\CP1-13.pqi
Output file: C:\Users\XUHENGCHAO\Desktop\PHreeQC\culture\CP1-13.pqo
Database file: C:\Program Files\USGS\Phreeqc Interactive 3.3.12-12704\database\wateq4f.dat
* * *
Reading data base.
* * *
    SOLUTION_MASTER_SPECIES
    SOLUTION_SPECIES
    PHASES
    EXCHANGE_MASTER_SPECIES
    EXCHANGE_SPECIES
    SURFACE_MASTER_SPECIES
    SURFACE_SPECIES
    RATES
    END
* * *
Reading input data for simulation 1.
* * *
    DATABASE C:\Program Files\USGS\Phreeqc Interactive 3.3.12-12704\database\wateq4f.dat
    SOLUTION 1
        temp     25
        pH      8
        pe      4
        redox   pe
        units   mmol/l
        density  1.02
        Alkalinity 2.18
        B       0.37
        Br     0.72
        C      0.64
        Ca     7.88
        Cl     483.51 |

```
                F           0.07
                K           8.04
                Mg          49.72
                N(5)        0.88
                Na          417.34
                P           0.04
                S(6)        24.99
                Si          0.11
                Sr          0.08
                water       1 # kg
* * *
Beginning of initial solution calculations.
* * *
Initial solution 1.

pH will be adjusted to obtain desired alkalinity.

----------------------------Solution composition----------------------------

    Elements              Molality          Moles

    Alkalinity         2.205e-03    2.205e-03
    B                     3.742e-04    3.742e-04
    Br                    7.282e-04    7.282e-04
    C                     6.473e-04    6.473e-04
    Ca                    7.969e-03    7.969e-03
    Cl                    4.890e-01    4.890e-01
    F                     7.079e-05    7.079e-05
    K                     8.131e-03    8.131e-03
    Mg                    5.028e-02    5.028e-02
    N(5)                  8.900e-04    8.900e-04
    Na                    4.221e-01    4.221e-01
    P                     4.045e-05    4.045e-05
    S(6)                  2.527e-02    2.527e-02
    Si                    1.112e-04    1.112e-04
    Sr                    8.091e-05    8.091e-05

--------------------------Description of solution--------------------------
```

pe    =    4.000

Activity of water    =    0.983

Ionic strength (mol/kgw)    =    5.876e-01

Mass of water (kg)    =    1.000e+00

Total CO2 (mol/kg)    =    6.473e-04

Temperature (癈)    =    25.00

Electrical balance (eq)    =    3.398e-03

Percent error, 100*(Cat-|An|)/(Cat+|An|)    =    0.32

Iterations    =    8

Total H    = 1.110143e+02

Total O    = 5.561420e+01

--------------------------Distribution of species--------------------------

|  |  |  | Log | Log | Log | mole V |
| Species | Molality | Activity | Molality | Activity | Gamma | cm?mol |
| OH- | 1.095e-04 | 6.788e-05 | -3.960 | -4.168 | -0.208 | (0) |
| H+ | 1.910e-10 | 1.450e-10 | -9.719 | -9.839 | -0.120 | 0.00 |
| H2O | 5.551e+01 | 9.831e-01 | 1.744 | -0.007 | 0.000 | 18.07 |
| B | 3.742e-04 |  |  |  |  |  |
| H2BO3- | 3.321e-04 | 1.912e-04 | -3.479 | -3.719 | -0.240 | (0) |
| H3BO3 | 4.207e-05 | 4.816e-05 | -4.376 | -4.317 | 0.059 | (0) |
| BF(OH)3- | 7.741e-10 | 4.455e-10 | -9.111 | -9.351 | -0.240 | (0) |
| BF2(OH)2- | 2.842e-16 | 1.636e-16 | -15.546 | -15.786 | -0.240 | (0) |
| BF3OH- | 1.068e-24 | 6.146e-25 | -23.972 | -24.211 | -0.240 | (0) |
| BF4- | 1.491e-32 | 8.579e-33 | -31.827 | -32.067 | -0.240 | (0) |
| Br | 7.282e-04 |  |  |  |  |  |

```
	Br-                    7.282e-04    5.381e-04    -3.138    -3.269    -
0.131      (0)
C(-4)          0.000e+00
	CH4                    0.000e+00    0.000e+00    -94.040   -93.981
0.059      (0)
C(4)           6.473e-04
	MgCO3                  2.144e-04    2.455e-04    -3.669    -3.610
0.059      (0)
	NaCO3-                 1.649e-04    1.126e-04    -3.783    -3.949    -
0.166      (0)
	CO3-2                  9.445e-05    2.053e-05    -4.025    -4.688    -
0.663      (0)
	HCO3-                  9.297e-05    6.349e-05    -4.032    -4.197    -
0.166      (0)
	CaCO3                  5.441e-05    6.229e-05    -4.264    -4.206
0.059      (0)
	MgHCO3+                1.457e-05    9.306e-06    -4.836    -5.031    -
0.195      (0)
	NaHCO3                 9.182e-06    1.051e-05    -5.037    -4.978
0.059      (0)
	CaHCO3+                2.098e-06    1.466e-06    -5.678    -5.834    -
0.156      (0)
	SrCO3                  2.394e-07    2.394e-07    -6.621    -6.621
0.000      (0)
	SrHCO3+                2.598e-08    1.774e-08    -7.585    -7.751    -
0.166      (0)
	CO2                    1.839e-08    2.105e-08    -7.735    -7.677
0.059      (0)
Ca             7.969e-03
	Ca+2                   7.131e-03    1.811e-03    -2.147    -2.742    -
0.595      (0)
	CaSO4                  7.747e-04    8.870e-04    -3.111    -3.052
0.059      (0)
	CaCO3                  5.441e-05    6.229e-05    -4.264    -4.206
0.059      (0)
	CaPO4-                 3.691e-06    2.520e-06    -5.433    -5.599    -
0.166      (0)
	CaOH+                  2.916e-06    2.038e-06    -5.535    -5.691    -
0.156      (0)
	CaHCO3+                2.098e-06    1.466e-06    -5.678    -5.834    -
0.156      (0)
```

| | | | | | | |
|---|---|---|---|---|---|---|
| CaF+ | | 5.344e-07 | 3.664e-07 | -6.272 | -6.436 | -0.164 (0) |
| CaHPO4 | | 1.349e-07 | 1.545e-07 | -6.870 | -6.811 | 0.059 (0) |
| CaH2PO4+ | | 2.465e-11 | 1.683e-11 | -10.608 | -10.774 | -0.166 (0) |
| CaHSO4+ | | 1.020e-12 | 7.534e-13 | -11.992 | -12.123 | -0.131 (0) |
| Cl | 4.890e-01 | | | | | |
| Cl- | | 4.890e-01 | 3.092e-01 | -0.311 | -0.510 | -0.199 (0) |
| F | 7.079e-05 | | | | | |
| F- | | 3.749e-05 | 2.323e-05 | -4.426 | -4.634 | -0.208 (0) |
| MgF+ | | 2.933e-05 | 1.923e-05 | -4.533 | -4.716 | -0.183 (0) |
| NaF | | 3.438e-06 | 3.937e-06 | -5.464 | -5.405 | 0.059 (0) |
| CaF+ | | 5.344e-07 | 3.664e-07 | -6.272 | -6.436 | -0.164 (0) |
| BF(OH)3- | | 7.741e-10 | 4.455e-10 | -9.111 | -9.351 | -0.240 (0) |
| HF | | 4.413e-12 | 5.052e-12 | -11.355 | -11.297 | 0.059 (0) |
| HF2- | | 7.268e-16 | 4.504e-16 | -15.139 | -15.346 | -0.208 (0) |
| BF2(OH)2- | | 2.842e-16 | 1.636e-16 | -15.546 | -15.786 | -0.240 (0) |
| H2F2 | | 5.810e-23 | 6.652e-23 | -22.236 | -22.177 | 0.059 (0) |
| BF3OH- | | 1.068e-24 | 6.146e-25 | -23.972 | -24.211 | -0.240 (0) |
| BF4- | | 1.491e-32 | 8.579e-33 | -31.827 | -32.067 | -0.240 (0) |
| SiF6-2 | | 0.000e+00 | 0.000e+00 | -40.602 | -41.294 | -0.692 (0) |
| H(0) | 2.600e-31 | | | | | |
| H2 | | 1.300e-31 | 1.488e-31 | -30.886 | -30.827 | 0.059 (0) |
| K | 8.131e-03 | | | | | |
| K+ | | 8.003e-03 | 5.061e-03 | -2.097 | -2.296 | - |

```
                    0.199      (0)
   KSO4-                    1.279e-04   8.731e-05   -3.893    -4.059    -
0.166      (0)
   KHPO4-                   2.249e-09   1.535e-09   -8.648    -8.814    -
0.166      (0)
Mg              5.028e-02
   Mg+2                     4.326e-02   1.253e-02   -1.364    -1.902    -
0.538      (0)
   MgSO4                       6.298e-03   7.210e-03   -2.201    -2.142
0.059      (0)
   MgOH+                    4.339e-04   3.084e-04   -3.363    -3.511    -
0.148      (0)
   MgCO3                       2.144e-04   2.455e-04   -3.669    -3.610
0.059      (0)
   MgPO4-                   3.445e-05   2.352e-05   -4.463    -4.629    -
0.166      (0)
   MgF+                     2.933e-05   1.923e-05   -4.533    -4.716    -
0.183      (0)
   MgHCO3+                  1.457e-05   9.306e-06   -4.836    -5.031    -
0.195      (0)
   MgHPO4                      1.262e-06   1.445e-06   -5.899    -5.840
0.059      (0)
   MgH2PO4+                 2.172e-10   1.483e-10   -9.663    -9.829    -
0.166      (0)
N(5)            8.900e-04
   NO3-                     8.900e-04   5.329e-04   -3.051    -3.273    -
0.223      (0)
Na              4.221e-01
   Na+                      4.166e-01   2.944e-01   -0.380    -0.531    -
0.151      (0)
   NaSO4-                   5.306e-03   3.623e-03   -2.275    -2.441    -
0.166      (0)
   NaCO3-                   1.649e-04   1.126e-04   -3.783    -3.949    -
0.166      (0)
   NaHCO3                      9.182e-06   1.051e-05   -5.037    -4.978
0.059      (0)
   NaF                         3.438e-06   3.937e-06   -5.464    -5.405
0.059      (0)
   NaHPO4-                  1.308e-07   8.932e-08   -6.883    -7.049    -
0.166      (0)
O(0)            3.178e-31
```

| Species | | Molality | Activity | Log Molality | Log Activity | Log Gamma | |
|---|---|---|---|---|---|---|---|
| O2 | | 1.589e-31 | 1.819e-31 | -30.799 | -30.740 | 0.059 | (0) |
| P | 4.045e-05 | | | | | | |
| MgPO4- | | 3.445e-05 | 2.352e-05 | -4.463 | -4.629 | -0.166 | (0) |
| CaPO4- | | 3.691e-06 | 2.520e-06 | -5.433 | -5.599 | -0.166 | (0) |
| MgHPO4 | | 1.262e-06 | 1.445e-06 | -5.899 | -5.840 | 0.059 | (0) |
| HPO4-2 | | 7.660e-07 | 1.556e-07 | -6.116 | -6.808 | -0.692 | (0) |
| CaHPO4 | | 1.349e-07 | 1.545e-07 | -6.870 | -6.811 | 0.059 | (0) |
| NaHPO4- | | 1.308e-07 | 8.932e-08 | -6.883 | -7.049 | -0.166 | (0) |
| PO4-3 | | 1.747e-08 | 4.837e-10 | -7.758 | -9.315 | -1.558 | (0) |
| KHPO4- | | 2.249e-09 | 1.535e-09 | -8.648 | -8.814 | -0.166 | (0) |
| H2PO4- | | 5.321e-10 | 3.633e-10 | -9.274 | -9.440 | -0.166 | (0) |
| MgH2PO4+ | | 2.172e-10 | 1.483e-10 | -9.663 | -9.829 | -0.166 | (0) |
| CaH2PO4+ | | 2.465e-11 | 1.683e-11 | -10.608 | -10.774 | -0.166 | (0) |
| S(6) | 2.527e-02 | | | | | | |
| SO4-2 | | 1.276e-02 | 2.455e-03 | -1.894 | -2.610 | -0.716 | (0) |
| MgSO4 | | 6.298e-03 | 7.210e-03 | -2.201 | -2.142 | 0.059 | (0) |
| NaSO4- | | 5.306e-03 | 3.623e-03 | -2.275 | -2.441 | -0.166 | (0) |
| CaSO4 | | 7.747e-04 | 8.870e-04 | -3.111 | -3.052 | 0.059 | (0) |
| KSO4- | | 1.279e-04 | 8.731e-05 | -3.893 | -4.059 | -0.166 | (0) |
| SrSO4 | | 7.637e-06 | 8.744e-06 | -5.117 | -5.058 | 0.059 | (0) |
| HSO4- | | 5.278e-11 | 3.461e-11 | -10.278 | -10.461 | -0.183 | (0) |
| CaHSO4+ | | 1.020e-12 | 7.534e-13 | -11.992 | -12.123 | - | |

0.131     (0)
Si               1.112e-04
    H3SiO4-            7.178e-05     4.584e-05     -4.144     -4.339     -
0.195     (0)
    H4SiO4            3.937e-05     4.507e-05     -4.405     -4.346
0.059     (0)
    H2SiO4-2          9.891e-08     2.150e-08     -7.005     -7.667     -
0.663     (0)
    SiF6-2            0.000e+00     0.000e+00     -40.602     -41.294     -
0.692     (0)
Sr               8.091e-05
    Sr+2              7.300e-05     1.827e-05     -4.137     -4.738     -
0.602     (0)
    SrSO4             7.637e-06     8.744e-06     -5.117     -5.058
0.059     (0)
    SrCO3             2.394e-07     2.394e-07     -6.621     -6.621
0.000     (0)
    SrHCO3+           2.598e-08     1.774e-08     -7.585     -7.751     -
0.166     (0)
    SrOH+             9.461e-09     6.352e-09     -8.024     -8.197     -
0.173     (0)

----------------------------Saturation indices----------------------------

    Phase                SI** log IAP     log K(298 K,     1 atm)

    Anhydrite         -0.99     -5.35     -4.36     CaSO4
    Aragonite         0.91     -7.43     -8.34     CaCO3
    Artinite          1.55     11.15     9.60     MgCO3:Mg(OH)2:3H2O
    Brucite           0.92     17.76     16.84     Mg(OH)2
    Calcite           1.05     -7.43     -8.48     CaCO3
    Celestite         -0.72     -7.35     -6.63     SrSO4
    CH4(g)            -91.12     -93.98     -2.86     CH4
    Chalcedony        -0.78     -4.33     -3.55     SiO2
    Chrysotile        12.42     44.63     32.20     Mg3Si2O5(OH)4
    Clinoenstatite    2.09     13.44     11.34     MgSiO3
    CO2(g)            -6.21     -7.68     -1.47     CO2
    Cristobalite      -0.74     -4.33     -3.59     SiO2
    Diopside          6.14     26.03     19.89     CaMgSi2O6
    Dolomite          3.07     -14.02     -17.09     CaMg(CO3)2
    Dolomite(d)       2.52     -14.02     -16.54     CaMg(CO3)2

```
    Epsomite          -2.42       -4.56     -2.14   MgSO4:7H2O
    FCO3Apatite                   26.56               -87.84    -114.40
Ca9.316Na0.36Mg0.144(PO4)4.8(CO3)1.2F2.48
    Fluorapatite       8.35       -9.25    -17.60   Ca5(PO4)3F
    Fluorite          -1.41      -12.01    -10.60   CaF2
    Forsterite         2.90       31.20     28.31   Mg2SiO4
    Gypsum            -0.79       -5.37     -4.58   CaSO4:2H2O
    H2(g)            -27.68      -30.83     -3.15   H2
    H2O(g)            -1.52       -0.01      1.51   H2O
    Halite            -2.62       -1.04      1.58   NaCl
    Huntite            2.77      -27.20    -29.97   CaMg3(CO3)4
    Hydromagnesite     0.13       -8.63     -8.76   Mg5(CO3)4(OH)2:4H2O
    Hydroxyapatite     8.63        5.21     -3.42   Ca5(PO4)3OH
    Magadiite         -6.75      -21.05    -14.30   NaSi7O13(OH)3:3H2O
    Magnesite          1.44       -6.59     -8.03   MgCO3
    Mirabilite        -2.63       -3.75     -1.11   Na2SO4:10H2O
    Nahcolite         -4.18       -4.73     -0.55   NaHCO3
    Natron            -4.51       -5.82     -1.31   Na2CO3:10H2O
    Nesquehonite      -0.99       -6.61     -5.62   MgCO3:3H2O
    O2(g)            -27.85      -30.74     -2.89   O2
    Portlandite       -5.88       16.92     22.80   Ca(OH)2
    Quartz            -0.35       -4.33     -3.98   SiO2
    Sepiolite          6.76       22.52     15.76   Mg2Si3O7.5OH:3H2O
    Sepiolite(d)       3.86       22.52     18.66   Mg2Si3O7.5OH:3H2O
    Silicagel         -1.31       -4.33     -3.02   SiO2
    SiO2(a)           -1.62       -4.33     -2.71   SiO2
    SrF2              -5.47      -14.01     -8.54   SrF2
    Strontianite      -0.16       -9.43     -9.27   SrCO3
    Talc              14.57       35.97     21.40   Mg3Si4O10(OH)2
    Thenardite        -3.49       -3.67     -0.18   Na2SO4
    Thermonatrite     -5.88       -5.76      0.13   Na2CO3:H2O
    Tremolite         31.46       88.04     56.57   Ca2Mg5Si8O22(OH)2
    Trona             -9.70      -10.49     -0.80   NaHCO3:Na2CO3:2H2O

**For a gas, SI = log10(fugacity). Fugacity = pressure * phi / 1 atm.
   For ideal gases, phi = 1.
* * *
End of simulation.
* * *
```

| | |
|---|---|
| | ----------------------------------
Reading input data for simulation 2.
* * ** * *
End of Run after 0.156 Seconds.
------------------------------ |
| Cyanobacteria+
virus
14  Day | Input file: C:\Users\XUHENGCHAO\Desktop\PHreeQC\culture\CP1-14.pqi
Output file: C:\Users\XUHENGCHAO\Desktop\PHreeQC\culture\CP1-14.pqo
Database  file:  C:\Program  Files\USGS\Phreeqc  Interactive  3.3.12-
12704\database\wateq4f.dat
* * *
Reading data base.
* * *
    SOLUTION_MASTER_SPECIES
    SOLUTION_SPECIES
    PHASES
    EXCHANGE_MASTER_SPECIES
    EXCHANGE_SPECIES
    SURFACE_MASTER_SPECIES
    SURFACE_SPECIES
    RATES
    END
* * *
Reading input data for simulation 1.
* * *
    DATABASE  C:\Program  Files\USGS\Phreeqc  Interactive  3.3.12-
12704\database\wateq4f.dat
    SOLUTION 1
        temp      25
        pH        8
        pe        4
        redox     pe
        units     mmol/l
        density   1.02
        Alkalinity 3
        B         0.37 |

```
              Br          0.72
              C           1.03
              Ca          7.54
              Cl          483.51
              F           0.07
              K           8.04
              Mg          49.6
              N(5)        0.88
              Na          417.34
              P           0.04
              S(6)        24.99
              Si          0.11
              Sr          0.08
              water       1 # kg
* * *
Beginning of initial solution calculations.
* * *
Initial solution 1.

pH will be adjusted to obtain desired alkalinity.

---------------------------Solution composition---------------------------

    Elements            Molality            Moles

    Alkalinity          3.034e-03     3.034e-03
    B                     3.742e-04     3.742e-04
    Br                    7.282e-04     7.282e-04
    C                     1.042e-03     1.042e-03
    Ca                    7.626e-03     7.626e-03
    Cl                    4.890e-01     4.890e-01
    F                     7.080e-05     7.080e-05
    K                     8.132e-03     8.132e-03
    Mg                    5.016e-02     5.016e-02
    N(5)                  8.900e-04     8.900e-04
    Na                    4.221e-01     4.221e-01
    P                     4.046e-05     4.046e-05
    S(6)                  2.527e-02     2.527e-02
    Si                    1.113e-04     1.113e-04
```

Sr                    8.091e-05     8.091e-05

---------------------------Description of solution---------------------------

                                           pH    =    9.903          Adjust
alkalinity

                                           pe    =    4.000
                            Activity of water    =    0.983
                      Ionic strength (mol/kgw)   =    5.866e-01
                          Mass of water (kg)    =    1.000e+00
                          Total CO2 (mol/kg)    =    1.042e-03
                           Temperature (癈)    =    25.00
                        Electrical balance (eq)   =    1.638e-03
  Percent error, 100*(Cat-|An|)/(Cat+|An|)    =    0.16
                                   Iterations   =    8
                              Total H   = 1.110144e+02
                              Total O   = 5.561547e+01

---------------------------Distribution of species---------------------------

                                                      Log            Log
Log       mole V
     Species              Molality    Activity  Molality  Activity      Gamma
cm?mol

     OH-                  1.269e-04    7.863e-05    -3.897      -4.104          -
0.208       (0)
     H+                   1.649e-10    1.252e-10    -9.783      -9.903          -
0.120          0.00
     H2O                  5.551e+01    9.831e-01     1.744      -0.007
0.000       18.07
B              3.742e-04
     H2BO3-               3.373e-04    1.942e-04    -3.472      -3.712          -
0.240       (0)
     H3BO3                3.690e-05    4.224e-05    -4.433      -4.374
0.059       (0)
     BF(OH)3-             6.810e-10    3.921e-10    -9.167      -9.407          -
0.240       (0)
     BF2(OH)2-            2.166e-16    1.247e-16   -15.664     -15.904          -
0.240       (0)
     BF3OH-               7.051e-25    4.059e-25   -24.152     -24.392          -

0.240     (0)
    BF4-                8.527e-33    4.909e-33    -32.069    -32.309    -
0.240     (0)
Br            7.282e-04
    Br-                 7.282e-04    5.380e-04    -3.138    -3.269    -
0.131     (0)
C(-4)          0.000e+00
    CH4                 0.000e+00    0.000e+00    -94.457    -94.399
0.059     (0)
C(4)          1.042e-03
    MgCO3               3.540e-04    4.052e-04    -3.451    -3.392
0.059     (0)
    NaCO3-              2.740e-04    1.871e-04    -3.562    -3.728    -
0.166     (0)
    CO3-2               1.569e-04    3.414e-05    -3.804    -4.467    -
0.662     (0)
    HCO3-               1.334e-04    9.111e-05    -3.875    -4.040    -
0.166     (0)
    CaCO3               8.615e-05    9.860e-05    -4.065    -4.006
0.059     (0)
    MgHCO3+             2.076e-05    1.326e-05    -4.683    -4.877    -
0.195     (0)
    NaHCO3              1.318e-05    1.508e-05    -4.880    -4.822
0.059     (0)
    CaHCO3+             2.867e-06    2.004e-06    -5.543    -5.698    -
0.156     (0)
    SrCO3               3.972e-07    3.972e-07    -6.401    -6.401
0.000     (0)
    SrHCO3+             3.719e-08    2.540e-08    -7.430    -7.595    -
0.166     (0)
    CO2                 2.278e-08    2.608e-08    -7.642    -7.584
0.059     (0)
Ca            7.626e-03
    Ca+2                6.789e-03    1.724e-03    -2.168    -2.763    -
0.595     (0)
    CaSO4               7.406e-04    8.477e-04    -3.130    -3.072
0.059     (0)
    CaCO3               8.615e-05    9.860e-05    -4.065    -4.006
0.059     (0)
    CaPO4-              3.581e-06    2.446e-06    -5.446    -5.612    -
0.166     (0)

| Species | Molality | Activity | Log Molality | Log Activity | Log Gamma | |
|---|---|---|---|---|---|---|
| CaOH+ | 3.216e-06 | 2.248e-06 | -5.493 | -5.648 | -0.156 | (0) |
| CaHCO3+ | 2.867e-06 | 2.004e-06 | -5.543 | -5.698 | -0.156 | (0) |
| CaF+ | 5.106e-07 | 3.501e-07 | -6.292 | -6.456 | -0.164 | (0) |
| CaHPO4 | 1.130e-07 | 1.294e-07 | -6.947 | -6.888 | 0.059 | (0) |
| CaH2PO4+ | 1.782e-11 | 1.217e-11 | -10.749 | -10.915 | -0.166 | (0) |
| CaHSO4+ | 8.413e-13 | 6.216e-13 | -12.075 | -12.207 | -0.131 | (0) |
| Cl | 4.890e-01 | | | | | |
| Cl- | 4.890e-01 | 3.093e-01 | -0.311 | -0.510 | -0.199 | (0) |
| F | 7.080e-05 | | | | | |
| F- | 3.762e-05 | 2.332e-05 | -4.425 | -4.632 | -0.208 | (0) |
| MgF+ | 2.922e-05 | 1.916e-05 | -4.534 | -4.718 | -0.183 | (0) |
| NaF | 3.451e-06 | 3.950e-06 | -5.462 | -5.403 | 0.059 | (0) |
| CaF+ | 5.106e-07 | 3.501e-07 | -6.292 | -6.456 | -0.164 | (0) |
| BF(OH)3- | 6.810e-10 | 3.921e-10 | -9.167 | -9.407 | -0.240 | (0) |
| HF | 3.824e-12 | 4.377e-12 | -11.418 | -11.359 | 0.059 | (0) |
| HF2- | 6.317e-16 | 3.916e-16 | -15.199 | -15.407 | -0.208 | (0) |
| BF2(OH)2- | 2.166e-16 | 1.247e-16 | -15.664 | -15.904 | -0.240 | (0) |
| H2F2 | 4.361e-23 | 4.992e-23 | -22.360 | -22.302 | 0.059 | (0) |
| BF3OH- | 7.051e-25 | 4.059e-25 | -24.152 | -24.392 | -0.240 | (0) |
| BF4- | 8.527e-33 | 4.909e-33 | -32.069 | -32.309 | -0.240 | (0) |
| SiF6-2 | 0.000e+00 | 0.000e+00 | -40.891 | -41.583 | -0.692 | (0) |
| H(0) | 1.938e-31 | | | | | |

| Species | Total | Molality | Activity | Log Molality | Log Activity | Log Gamma |
|---|---|---|---|---|---|---|
| H2 | | 9.689e-32 | 1.109e-31 | -31.014 | -30.955 | 0.059 (0) |
| K | 8.132e-03 | | | | | |
| K+ | | 8.003e-03 | 5.062e-03 | -2.097 | -2.296 | -0.199 (0) |
| KSO4- | | 1.283e-04 | 8.765e-05 | -3.892 | -4.057 | -0.166 (0) |
| KHPO4- | | 1.978e-09 | 1.351e-09 | -8.704 | -8.869 | -0.166 (0) |
| Mg | 5.016e-02 | | | | | |
| Mg+2 | | 4.295e-02 | 1.244e-02 | -1.367 | -1.905 | -0.538 (0) |
| MgSO4 | | 6.277e-03 | 7.185e-03 | -2.202 | -2.144 | 0.059 (0) |
| MgOH+ | | 4.990e-04 | 3.547e-04 | -3.302 | -3.450 | -0.148 (0) |
| MgCO3 | | 3.540e-04 | 4.052e-04 | -3.451 | -3.392 | 0.059 (0) |
| MgPO4- | | 3.485e-05 | 2.380e-05 | -4.458 | -4.623 | -0.166 (0) |
| MgF+ | | 2.922e-05 | 1.916e-05 | -4.534 | -4.718 | -0.183 (0) |
| MgHCO3+ | | 2.076e-05 | 1.326e-05 | -4.683 | -4.877 | -0.195 (0) |
| MgHPO4 | | 1.103e-06 | 1.262e-06 | -5.958 | -5.899 | 0.059 (0) |
| MgH2PO4+ | | 1.637e-10 | 1.118e-10 | -9.786 | -9.951 | -0.166 (0) |
| N(5) | 8.900e-04 | | | | | |
| NO3- | | 8.900e-04 | 5.331e-04 | -3.051 | -3.273 | -0.223 (0) |
| Na | 4.221e-01 | | | | | |
| Na+ | | 4.165e-01 | 2.944e-01 | -0.380 | -0.531 | -0.151 (0) |
| NaSO4- | | 5.323e-03 | 3.635e-03 | -2.274 | -2.439 | -0.166 (0) |
| NaCO3- | | 2.740e-04 | 1.871e-04 | -3.562 | -3.728 | -0.166 (0) |
| NaHCO3 | | 1.318e-05 | 1.508e-05 | -4.880 | -4.822 | 0.059 (0) |
| NaF | | 3.451e-06 | 3.950e-06 | -5.462 | -5.403 | |

0.059      (0)

    NaHPO4-              1.150e-07    7.856e-08    -6.939    -7.105    -0.166      (0)

O(0)          5.724e-31

    O2                   2.862e-31    3.276e-31    -30.543    -30.485    0.059      (0)

P            4.046e-05

    MgPO4-              3.485e-05    2.380e-05    -4.458    -4.623    -0.166      (0)

    CaPO4-              3.581e-06    2.446e-06    -5.446    -5.612    -0.166      (0)

    MgHPO4              1.103e-06    1.262e-06    -5.958    -5.899    0.059      (0)

    HPO4-2              6.734e-07    1.369e-07    -6.172    -6.864    -0.692      (0)

    NaHPO4-              1.150e-07    7.856e-08    -6.939    -7.105    -0.166      (0)

    CaHPO4              1.130e-07    1.294e-07    -6.947    -6.888    0.059      (0)

    PO4-3              1.778e-08    4.929e-10    -7.750    -9.307    -1.557      (0)

    KHPO4-              1.978e-09    1.351e-09    -8.704    -8.869    -0.166      (0)

    H2PO4-              4.040e-10    2.759e-10    -9.394    -9.559    -0.166      (0)

    MgH2PO4+              1.637e-10    1.118e-10    -9.786    -9.951    -0.166      (0)

    CaH2PO4+              1.782e-11    1.217e-11    -10.749    -10.915    -0.166      (0)

S(6)          2.527e-02

    SO4-2              1.280e-02    2.464e-03    -1.893    -2.608    -0.715      (0)

    MgSO4              6.277e-03    7.185e-03    -2.202    -2.144    0.059      (0)

    NaSO4-              5.323e-03    3.635e-03    -2.274    -2.439    -0.166      (0)

    CaSO4              7.406e-04    8.477e-04    -3.130    -3.072    0.059      (0)

    KSO4-              1.283e-04    8.765e-05    -3.892    -4.057    -0.166      (0)

    SrSO4              7.650e-06    8.757e-06    -5.116    -5.058

0.059      (0)

   HSO4-           4.572e-11    2.999e-11    -10.340    -10.523    -
0.183      (0)

   CaHSO4+        8.413e-13    6.216e-13    -12.075    -12.207    -
0.131      (0)

Si          1.113e-04

   H3SiO4-        7.541e-05    4.817e-05    -4.123    -4.317    -
0.195      (0)

   H4SiO4         3.572e-05    4.089e-05    -4.447    -4.388
0.059      (0)

   H2SiO4-2     1.203e-07    2.618e-08    -6.920    -7.582    -
0.662      (0)

   SiF6-2        0.000e+00    0.000e+00    -40.891    -41.583    -
0.692      (0)

Sr          8.091e-05

   Sr+2          7.282e-05    1.823e-05    -4.138    -4.739    -
0.602      (0)

   SrSO4         7.650e-06    8.757e-06    -5.116    -5.058
0.059      (0)

   SrCO3         3.972e-07    3.972e-07    -6.401    -6.401
0.000      (0)

   SrHCO3+       3.719e-08    2.540e-08    -7.430    -7.595    -
0.166      (0)

   SrOH+         1.093e-08    7.342e-09    -7.961    -8.134    -
0.173      (0)

----------------------------Saturation indices----------------------------

   Phase               SI** log IAP     log K(298 K,    1 atm)

| Phase | SI** | log IAP | log K | |
|---|---|---|---|---|
| Anhydrite | -1.01 | -5.37 | -4.36 | CaSO4 |
| Aragonite | 1.11 | -7.23 | -8.34 | CaCO3 |
| Artinite | 1.89 | 11.49 | 9.60 | MgCO3:Mg(OH)2:3H2O |
| Brucite | 1.05 | 17.89 | 16.84 | Mg(OH)2 |
| Calcite | 1.25 | -7.23 | -8.48 | CaCO3 |
| Celestite | -0.72 | -7.35 | -6.63 | SrSO4 |
| CH4(g) | -91.54 | -94.40 | -2.86 | CH4 |
| Chalcedony | -0.82 | -4.37 | -3.55 | SiO2 |
| Chrysotile | 12.71 | 44.92 | 32.20 | Mg3Si2O5(OH)4 |
| Clinoenstatite | 2.18 | 13.52 | 11.34 | MgSiO3 |
| CO2(g) | -6.12 | -7.58 | -1.47 | CO2 |

| Mineral | SI | | | Formula |
|---|---|---|---|---|
| Cristobalite | -0.79 | -4.37 | -3.59 | SiO2 |
| Diopside | 6.29 | 26.18 | 19.89 | CaMgSi2O6 |
| Dolomite | 3.49 | -13.60 | -17.09 | CaMg(CO3)2 |
| Dolomite(d) | 2.94 | -13.60 | -16.54 | CaMg(CO3)2 |
| Epsomite | -2.43 | -4.57 | -2.14 | MgSO4:7H2O |
| FCO3Apatite | | 26.67 | | -87.73 | -114.40 |
| Ca9.316Na0.36Mg0.144(PO4)4.8(CO3)1.2F2.48 | | | | |
| Fluorapatite | 8.27 | -9.33 | -17.60 | Ca5(PO4)3F |
| Fluorite | -1.43 | -12.03 | -10.60 | CaF2 |
| Forsterite | 3.11 | 31.41 | 28.31 | Mg2SiO4 |
| Gypsum | -0.81 | -5.39 | -4.58 | CaSO4:2H2O |
| H2(g) | -27.81 | -30.96 | -3.15 | H2 |
| H2O(g) | -1.52 | -0.01 | 1.51 | H2O |
| Halite | -2.62 | -1.04 | 1.58 | NaCl |
| Huntite | 3.62 | -26.35 | -29.97 | CaMg3(CO3)4 |
| Hydromagnesite | 1.13 | -7.63 | -8.76 | Mg5(CO3)4(OH)2:4H2O |
| Hydroxyapatite | 8.62 | 5.19 | -3.42 | Ca5(PO4)3OH |
| Magadiite | -6.98 | -21.28 | -14.30 | NaSi7O13(OH)3:3H2O |
| Magnesite | 1.66 | -6.37 | -8.03 | MgCO3 |
| Mirabilite | -2.63 | -3.74 | -1.11 | Na2SO4:10H2O |
| Nahcolite | -4.02 | -4.57 | -0.55 | NaHCO3 |
| Natron | -4.29 | -5.60 | -1.31 | Na2CO3:10H2O |
| Nesquehonite | -0.77 | -6.39 | -5.62 | MgCO3:3H2O |
| O2(g) | -27.59 | -30.48 | -2.89 | O2 |
| Portlandite | -5.77 | 17.03 | 22.80 | Ca(OH)2 |
| Quartz | -0.39 | -4.37 | -3.98 | SiO2 |
| Sepiolite | 6.88 | 22.64 | 15.76 | Mg2Si3O7.5OH:3H2O |
| Sepiolite(d) | 3.98 | 22.64 | 18.66 | Mg2Si3O7.5OH:3H2O |
| Silicagel | -1.36 | -4.37 | -3.02 | SiO2 |
| SiO2(a) | -1.66 | -4.37 | -2.71 | SiO2 |
| SrF2 | -5.46 | -14.00 | -8.54 | SrF2 |
| Strontianite | 0.06 | -9.21 | -9.27 | SrCO3 |
| Talc | 14.78 | 36.18 | 21.40 | Mg3Si4O10(OH)2 |
| Thenardite | -3.49 | -3.67 | -0.18 | Na2SO4 |
| Thermonatrite | -5.66 | -5.54 | 0.13 | Na2CO3:H2O |
| Tremolite | 31.96 | 88.53 | 56.57 | Ca2Mg5Si8O22(OH)2 |
| Trona | -9.32 | -10.12 | -0.80 | NaHCO3:Na2CO3:2H2O |

**For a gas, SI = log10(fugacity). Fugacity = pressure * phi / 1 atm.
For ideal gases, phi = 1.

| | |
|---|---|
| | ------------------
End of simulation.
* * ** * *
Reading input data for simulation 2.
* * ** * *
End of Run after 0.172 Seconds.
------------------------------ |
| Cyanobacteria+
virus
16 Day | Input file: C:\Users\XUHENGCHAO\Desktop\PHreeQC\culture\CP1-15.pqi
Output file: C:\Users\XUHENGCHAO\Desktop\PHreeQC\culture\CP1-15.pqo
Database file: C:\Program Files\USGS\Phreeqc Interactive 3.3.12-12704\database\wateq4f.dat
* * *
Reading data base.
* * *
    SOLUTION_MASTER_SPECIES
    SOLUTION_SPECIES
    PHASES
    EXCHANGE_MASTER_SPECIES
    EXCHANGE_SPECIES
    SURFACE_MASTER_SPECIES
    SURFACE_SPECIES
    RATES
    END
* * *
Reading input data for simulation 1.
* * *
    DATABASE C:\Program Files\USGS\Phreeqc Interactive 3.3.12-12704\database\wateq4f.dat
    SOLUTION 1
        temp      25
        pH        8
        pe        4
        redox    pe |

```
                    units       mmol/l
                    density     1.02
                    Alkalinity  2.86
                    B              0.37
                    Br             0.72
                    C              1.43
                    Ca             7.62
                    Cl           483.51
                    F              0.07
                    K              8.04
                    Mg            49.76
                    N(5)           0.88
                    Na           417.34
                    P              0.04
                    S(6)          24.99
                    Si             0.11
                    Sr             0.08
                    water       1 # kg
* * *
Beginning of initial solution calculations.
* * *
Initial solution 1.

pH will be adjusted to obtain desired alkalinity.

----------------------------Solution composition----------------------------

     Elements              Molality           Moles

     Alkalinity          2.893e-03     2.893e-03
     B                      3.742e-04     3.742e-04
     Br                     7.282e-04     7.282e-04
     C                      1.446e-03     1.446e-03
     Ca                     7.707e-03     7.707e-03
     Cl                     4.890e-01     4.890e-01
     F                      7.080e-05     7.080e-05
     K                      8.132e-03     8.132e-03
     Mg                     5.033e-02     5.033e-02
     N(5)                   8.900e-04     8.900e-04
```

```
                Na                     4.221e-01    4.221e-01
                P                      4.046e-05    4.046e-05
                S(6)                   2.528e-02    2.528e-02
                Si                     1.113e-04    1.113e-04
                Sr                     8.091e-05    8.091e-05

--------------------------Description of solution--------------------------

                                       pH   =    9.371            Adjust
alkalinity

                                       pe   =    4.000
                        Activity of water   =    0.983
                  Ionic strength (mol/kgw)  =    5.874e-01
                     Mass of water (kg)  =    1.000e+00
                    Total CO2 (mol/kg)   =    1.446e-03
                      Temperature (癈)    =   25.00
                  Electrical balance (eq)  =    2.266e-03
   Percent error, 100*(Cat-|An|)/(Cat+|An|)   =    0.22
                            Iterations   =    9
                              Total H   = 1.110145e+02
                              Total O   = 5.561624e+01

--------------------------Distribution of species--------------------------

                                              Log             Log
Log        mole V
    Species               Molality    Activity  Molality  Activity      Gamma
cm?mol

    OH-                   3.730e-05    2.311e-05    -4.428    -4.636      -
0.208      (0)
    H+                    5.609e-10    4.258e-10    -9.251    -9.371      -
0.120          0.00
    H2O                   5.551e+01    9.831e-01     1.744    -0.007
0.000       18.07
B                3.742e-04
    H2BO3-                2.728e-04    1.570e-04    -3.564    -3.804      -
0.240      (0)
    H3BO3                 1.015e-04    1.162e-04    -3.994    -3.935
0.059      (0)
    BF(OH)3-              1.865e-09    1.074e-09    -8.729    -8.969      -
```

0.240      (0)
    BF2(OH)2-           2.010e-15    1.157e-15    -14.697    -14.937    -
0.240      (0)
    BF3OH-              2.216e-23    1.276e-23    -22.654    -22.894    -
0.240      (0)
    BF4-                9.078e-31    5.225e-31    -30.042    -30.282    -
0.240      (0)
Br              7.282e-04
    Br-                 7.282e-04    5.381e-04    -3.138     -3.269     -
0.131      (0)
C(-4)          0.000e+00
    CH4                 0.000e+00    0.000e+00    -89.143    -89.085
0.059      (0)
C(4)           1.446e-03
    HCO3-               4.508e-04    3.079e-04    -3.346     -3.512     -
0.166      (0)
    MgCO3               3.550e-04    4.064e-04    -3.450     -3.391
0.059      (0)
    NaCO3-              2.722e-04    1.859e-04    -3.565     -3.731     -
0.166      (0)
    CO3-2               1.559e-04    3.391e-05    -3.807     -4.470     -
0.663      (0)
    CaCO3               8.643e-05    9.895e-05    -4.063     -4.005
0.059      (0)
    MgHCO3+             7.085e-05    4.525e-05    -4.150     -4.344     -
0.195      (0)
    NaHCO3              4.451e-05    5.096e-05    -4.352     -4.293
0.059      (0)
    CaHCO3+             9.789e-06    6.840e-06    -5.009     -5.165     -
0.156      (0)
    SrCO3               3.942e-07    3.942e-07    -6.404     -6.404
0.000      (0)
    CO2                 2.618e-07    2.998e-07    -6.582     -6.523
0.059      (0)
    SrHCO3+             1.256e-07    8.575e-08    -6.901     -7.067     -
0.166      (0)
Ca              7.707e-03
    Ca+2                6.860e-03    1.742e-03    -2.164     -2.759     -
0.595      (0)
    CaSO4               7.458e-04    8.539e-04    -3.127     -3.069
0.059      (0)

|  | CaCO3 | 8.643e-05 | 9.895e-05 | -4.063 | -4.005 | 0.059 | (0) |
|  | CaHCO3+ | 9.789e-06 | 6.840e-06 | -5.009 | -5.165 | -0.156 | (0) |
|  | CaPO4- | 3.203e-06 | 2.187e-06 | -5.494 | -5.660 | -0.166 | (0) |
|  | CaOH+ | 9.553e-07 | 6.675e-07 | -6.020 | -6.176 | -0.156 | (0) |
|  | CaF+ | 5.138e-07 | 3.522e-07 | -6.289 | -6.453 | -0.164 | (0) |
|  | CaHPO4 | 3.438e-07 | 3.936e-07 | -6.464 | -6.405 | 0.059 | (0) |
|  | CaH2PO4+ | 1.845e-10 | 1.260e-10 | -9.734 | -9.900 | -0.166 | (0) |
|  | CaHSO4+ | 2.883e-12 | 2.130e-12 | -11.540 | -11.672 | -0.131 | (0) |
| Cl | 4.890e-01 |  |  |  |  |  |  |
|  | Cl- | 4.890e-01 | 3.093e-01 | -0.311 | -0.510 | -0.199 | (0) |
| F | 7.080e-05 |  |  |  |  |  |  |
|  | F- | 3.746e-05 | 2.322e-05 | -4.426 | -4.634 | -0.208 | (0) |
|  | MgF+ | 2.938e-05 | 1.927e-05 | -4.532 | -4.715 | -0.183 | (0) |
|  | NaF | 3.435e-06 | 3.933e-06 | -5.464 | -5.405 | 0.059 | (0) |
|  | CaF+ | 5.138e-07 | 3.522e-07 | -6.289 | -6.453 | -0.164 | (0) |
|  | BF(OH)3- | 1.865e-09 | 1.074e-09 | -8.729 | -8.969 | -0.240 | (0) |
|  | HF | 1.295e-11 | 1.483e-11 | -10.888 | -10.829 | 0.059 | (0) |
|  | HF2- | 2.131e-15 | 1.321e-15 | -14.671 | -14.879 | -0.208 | (0) |
|  | BF2(OH)2- | 2.010e-15 | 1.157e-15 | -14.697 | -14.937 | -0.240 | (0) |
|  | H2F2 | 5.004e-22 | 5.728e-22 | -21.301 | -21.242 | 0.059 | (0) |
|  | BF3OH- | 2.216e-23 | 1.276e-23 | -22.654 | -22.894 | -0.240 | (0) |
|  | BF4- | 9.078e-31 | 5.225e-31 | -30.042 | -30.282 | - |  |

```
                            0.240     (0)
    SiF6-2          3.227e-39   6.554e-40    -38.491    -39.183     -0.692
(0)
H(0)          2.242e-30
    H2              1.121e-30   1.283e-30    -29.950    -29.892
0.059     (0)
K          8.132e-03
    K+              8.004e-03   5.062e-03     -2.097     -2.296     -
0.199     (0)
    KSO4-           1.280e-04   8.738e-05     -3.893     -4.059     -
0.166     (0)
    KHPO4-          5.957e-09   4.068e-09     -8.225     -8.391     -
0.166     (0)
Mg          5.033e-02
    Mg+2            4.337e-02   1.256e-02     -1.363     -1.901     -
0.538     (0)
    MgSO4           6.319e-03   7.234e-03     -2.199     -2.141
0.059     (0)
    MgCO3           3.550e-04   4.064e-04     -3.450     -3.391
0.059     (0)
    MgOH+           1.481e-04   1.053e-04     -3.829     -3.978     -
0.148     (0)
    MgHCO3+         7.085e-05   4.525e-05     -4.150     -4.344     -
0.195     (0)
    MgPO4-          3.115e-05   2.127e-05     -4.506     -4.672     -
0.166     (0)
    MgF+            2.938e-05   1.927e-05     -4.532     -4.715     -
0.183     (0)
    MgHPO4          3.352e-06   3.837e-06     -5.475     -5.416
0.059     (0)
    MgH2PO4+        1.694e-09   1.157e-09     -8.771     -8.937     -
0.166     (0)
N(5)          8.900e-04
    NO3-            8.900e-04   5.330e-04     -3.051     -3.273     -
0.223     (0)
Na          4.221e-01
    Na+             4.165e-01   2.944e-01     -0.380     -0.531     -
0.151     (0)
    NaSO4-          5.308e-03   3.625e-03     -2.275     -2.441     -
0.166     (0)
    NaCO3-          2.722e-04   1.859e-04     -3.565     -3.731     -
```

0.166          (0)
   NaHCO3                    4.451e-05     5.096e-05       -4.352        -4.293
0.059        (0)
   NaF                       3.435e-06     3.933e-06       -5.464        -5.405
0.059        (0)
   NaHPO4-              3.464e-07     2.365e-07      -6.460        -6.626        -
0.166        (0)
O(0)          4.273e-33
   O2                        2.137e-33     2.446e-33      -32.670       -32.612
0.059        (0)
P             4.046e-05
   MgPO4-              3.115e-05     2.127e-05      -4.506        -4.672        -
0.166        (0)
   MgHPO4                    3.352e-06     3.837e-06      -5.475        -5.416
0.059        (0)
   CaPO4-              3.203e-06     2.187e-06      -5.494        -5.660        -
0.166        (0)
   HPO4-2              2.029e-06     4.121e-07      -5.693        -6.385        -
0.692        (0)
   NaHPO4-             3.464e-07     2.365e-07      -6.460        -6.626        -
0.166        (0)
   CaHPO4                    3.438e-07     3.936e-07      -6.464        -6.405
0.059        (0)
   PO4-3               1.575e-08     4.364e-10      -7.803        -9.360        -
1.558        (0)
   KHPO4-              5.957e-09     4.068e-09      -8.225        -8.391        -
0.166        (0)
   H2PO4-              4.139e-09     2.826e-09      -8.383        -8.549        -
0.166        (0)
   MgH2PO4+            1.694e-09     1.157e-09      -8.771        -8.937        -
0.166        (0)
   CaH2PO4+            1.845e-10     1.260e-10      -9.734        -9.900        -
0.166        (0)
S(6)          2.528e-02
   SO4-2               1.277e-02     2.457e-03      -1.894        -2.610        -
0.716        (0)
   MgSO4                     6.319e-03     7.234e-03      -2.199        -2.141
0.059        (0)
   NaSO4-              5.308e-03     3.625e-03      -2.275        -2.441        -
0.166        (0)
   CaSO4                     7.458e-04     8.539e-04      -3.127        -3.069

0.059       (0)
   KSO4-            1.280e-04    8.738e-05    -3.893     -4.059     -
0.166       (0)
   SrSO4              7.620e-06    8.723e-06     -5.118     -5.059
0.059       (0)
   HSO4-            1.551e-10    1.017e-10    -9.809     -9.993     -
0.183       (0)
   CaHSO4+         2.883e-12    2.130e-12   -11.540   -11.672     -
0.131       (0)
Si          1.113e-04
   H4SiO4           6.863e-05    7.857e-05     -4.163     -4.105
0.059       (0)
   H3SiO4-         4.261e-05    2.721e-05    -4.371     -4.565     -
0.195       (0)
   H2SiO4-2       1.999e-08    4.347e-09    -7.699     -8.362     -
0.663       (0)
   SiF6-2         3.227e-39   6.554e-40   -38.491   -39.183   -0.692
(0)
Sr          8.091e-05
   Sr+2             7.277e-05    1.821e-05    -4.138     -4.740     -
0.602       (0)
   SrSO4              7.620e-06    8.723e-06     -5.118     -5.059
0.059       (0)
   SrCO3              3.942e-07    3.942e-07     -6.404     -6.404
0.000       (0)
   SrHCO3+         1.256e-07    8.575e-08     -6.901     -7.067     -
0.166       (0)
   SrOH+             3.212e-09    2.156e-09     -8.493     -8.666     -
0.173       (0)

----------------------------Saturation indices----------------------------

   Phase                SI** log IAP     log K(298 K,    1 atm)

   Anhydrite         -1.01     -5.37    -4.36   CaSO4
   Aragonite         1.11     -7.23    -8.34   CaCO3
   Artinite          0.83     10.43    9.60   MgCO3:Mg(OH)2:3H2O
   Brucite          -0.01     16.83    16.84   Mg(OH)2
   Calcite          1.25     -7.23    -8.48   CaCO3
   Celestite         -0.72     -7.35    -6.63   SrSO4
   CH4(g)          -86.22    -89.08    -2.86   CH4

| Chalcedony | -0.54 | -4.09 | -3.55 | SiO2 |
| Chrysotile | 10.10 | 42.31 | 32.20 | Mg3Si2O5(OH)4 |
| Clinoenstatite | 1.40 | 12.74 | 11.34 | MgSiO3 |
| CO2(g) | -5.06 | -6.52 | -1.47 | CO2 |
| Cristobalite | -0.50 | -4.09 | -3.59 | SiO2 |
| Diopside | 4.73 | 24.63 | 19.89 | CaMgSi2O6 |
| Dolomite | 3.49 | -13.60 | -17.09 | CaMg(CO3)2 |
| Dolomite(d) | 2.94 | -13.60 | -16.54 | CaMg(CO3)2 |
| Epsomite | -2.42 | -4.56 | -2.14 | MgSO4:7H2O |
| FCO3Apatite | | 26.45 | -87.95 | -114.40 | Ca9.316Na0.36Mg0.144(PO4)4.8(CO3)1.2F2.48 |
| Fluorapatite | 8.13 | -9.47 | -17.60 | Ca5(PO4)3F |
| Fluorite | -1.43 | -12.03 | -10.60 | CaF2 |
| Forsterite | 1.27 | 29.58 | 28.31 | Mg2SiO4 |
| Gypsum | -0.80 | -5.38 | -4.58 | CaSO4:2H2O |
| H2(g) | -26.74 | -29.89 | -3.15 | H2 |
| H2O(g) | -1.52 | -0.01 | 1.51 | H2O |
| Halite | -2.62 | -1.04 | 1.58 | NaCl |
| Huntite | 3.63 | -26.34 | -29.97 | CaMg3(CO3)4 |
| Hydromagnesite | 0.08 | -8.69 | -8.76 | Mg5(CO3)4(OH)2:4H2O |
| Hydroxyapatite | 7.95 | 4.53 | -3.42 | Ca5(PO4)3OH |
| Magadiite | -5.53 | -19.83 | -14.30 | NaSi7O13(OH)3:3H2O |
| Magnesite | 1.66 | -6.37 | -8.03 | MgCO3 |
| Mirabilite | -2.63 | -3.75 | -1.11 | Na2SO4:10H2O |
| Nahcolite | -3.49 | -4.04 | -0.55 | NaHCO3 |
| Natron | -4.29 | -5.61 | -1.31 | Na2CO3:10H2O |
| Nesquehonite | -0.77 | -6.39 | -5.62 | MgCO3:3H2O |
| O2(g) | -29.72 | -32.61 | -2.89 | O2 |
| Portlandite | -6.83 | 15.97 | 22.80 | Ca(OH)2 |
| Quartz | -0.11 | -4.09 | -3.98 | SiO2 |
| Sepiolite | 5.61 | 21.37 | 15.76 | Mg2Si3O7.5OH:3H2O |
| Sepiolite(d) | 2.71 | 21.37 | 18.66 | Mg2Si3O7.5OH:3H2O |
| Silicagel | -1.07 | -4.09 | -3.02 | SiO2 |
| SiO2(a) | -1.38 | -4.09 | -2.71 | SiO2 |
| SrF2 | -5.47 | -14.01 | -8.54 | SrF2 |
| Strontianite | 0.06 | -9.21 | -9.27 | SrCO3 |
| Talc | 12.73 | 34.13 | 21.40 | Mg3Si4O10(OH)2 |
| Thenardite | -3.49 | -3.67 | -0.18 | Na2SO4 |
| Thermonatrite | -5.66 | -5.54 | 0.13 | Na2CO3:H2O |
| Tremolite | 26.82 | 83.39 | 56.57 | Ca2Mg5Si8O22(OH)2 |
| Trona | -8.79 | -9.59 | -0.80 | NaHCO3:Na2CO3:2H2O |

**For a gas, SI = log10(fugacity). Fugacity = pressure * phi / 1 atm.
   For ideal gases, phi = 1.
* * *
End of simulation.
* * ** * *
Reading input data for simulation 2.
* * ** * *
End of Run after 0.162 Seconds.
* * *

[revised manuscript text omitted]

---

## Author Response (AR3)

Dear editor:

We are very glad to receive your decision about our paper "Precipitation of Calcium Carbonate Mineral Induced by Viral Lysis of Cyanobacteria: Evidence from Laboratory Experiments". Thank you very much for your hard work and timely reply. In revised version of manuscript:

- Title of the manuscript has been changed to "Precipitation of Calcium Carbonate Mineral Induced by Viral Lysis of Cyanobacteria: Evidence from Laboratory Experiments". This version of title has been reviewed by the editor in our response (BG-2018-194-manuscript-version 3.pdf, Date: 14 Dec 2018 and BG-2018-194-response-version 2.pdf, Date: 14 Dec 2018).
- The caption of Fig. 5 is corrected.
- "Phage" used in the text, tables and figures are changed to "virus".
-  New Foundation names are added. During this work, we has been helped by right of these foundations, but we forgot to write them.

Thank you very much for your consideration.

Sincerely Yours

Xiaotong Peng, on behalf of the co-authors.

[revised manuscript text omitted]

*Data availability*. Data used to generate Figs. 2, 3 are listed in the appendices. Genomic sequences are archived in an open-access database: https://submit.ncbi.nlm.nih.gov/subs/?search=SUB4436587

30

*Author contributions*: This work was designed by HX, XP and SY. Experimental work was performed by HX, SB, KT, SL and ZG. SB and HJ analysed genomic sequences. HX and SB led the writing of the papers and prepared the figures with contributions from all co-authors.

5    *Competing interests*. The authors declare that they have no conflict of interest.

**Acknowledgments**

Authors would like to acknowledge Dr. Dezhang Ren at Shanghai Jiaotong University for his help with the XRD analysis. The authors are also indebted to Ms Ling Li and Dr. Weijia Zhang at the Institute of Deep-sea Science and engineering for
10   their assistance with cyanobacteria identification. Financial support for this research came from the "National Key R&D Program of China" (Grant NO. 2018YFC0309802), the "National Natural Science Foundation of China" (Grant NO. 41506139 and 41506097), "
[revised manuscript text omitted]

| | Aragonite | 1.36 | 1.3 | 0.68 | 0.92 | 1.12 | 1.1 |
| | Brucite | 0.52 | 0.92 | -1.06 | 0.93 | 1.06 | -0.02 |
| | ACC | -0.70 | -0.76 | -1.38 | -1.14 | -0.94 | -0.96 |
| | vaterite | 0.94 | 0.88 | 0.26 | 0.50 | 0.70 | 0.68 |